# Neural Hamilton–Jacobi Characteristic Flows for Optimal Transport

**Yesom Park[1], Shu Liu[2], Mo Zhou[1], Stanley Osher[1]**[*]
[1]Department of Mathematics, University of California, Los Angeles
[2]Department of Mathematics, Florida State University
`yeisom@math.ucla.edu, sl25bn@fsu.edu, mozhou366@math.ucla.edu`

## Abstract

We present a novel framework for solving optimal transport (OT) problems based on the Hamilton–Jacobi (HJ) equation, whose viscosity solution uniquely characterizes the OT map. By leveraging the method of characteristics, we derive closed-form, bidirectional transport maps, thereby eliminating the need for numerical integration. The proposed method adopts a pure minimization framework: a single neural network is trained with a loss function derived from the method of characteristics of the HJ equation. This design guarantees convergence to the optimal map while eliminating adversarial training stages, thereby substantially reducing computational complexity. Furthermore, the framework naturally extends to a wide class of cost functions and supports class-conditional transport. Extensive experiments on diverse datasets demonstrate the accuracy, scalability, and efficiency of the proposed method, establishing it as a principled and versatile tool for OT applications with provable optimality.

## 1 Introduction

Optimal transport (OT) is a fundamental problem that seeks the most cost-efficient transform from one probability distribution into another by minimizing a transportation cost function, which quantifies the effort to move mass. With its strong theoretical foundation and broad practical relevance, OT has been widely applied in diverse areas, including traffic control (Carlier et al., 2008; Danila et al., 2006; Barthélemy & Flammini, 2006), biomedical data analysis (Schiebinger et al., 2019; Koshizuka & Sato, 2022; Bunne et al., 2023), generative modeling (Wang et al., 2021; Onken et al., 2021; Zhang & Katsoulakis, 2023; Liu et al., 2019), and domain adaptation (Courty et al., 2016; 2017; Damodaran et al., 2018; Balaji et al., 2020). In recent years, there has been growing interest in deep learning techniques to solve OT problems, leading to the development of methods grounded in various mathematical formulations. Early approaches were primarily built upon the classical Monge formulation (Lu et al., 2020; Xie et al., 2019) and its relaxation into the Kantorovich framework (Makkuva et al., 2020). While theoretically rigorous, these methods often suffer from high computational complexity. The primal–dual formulation, which recasts the OT problem as a saddle-point optimization over the generative map and the Kantorovich potential function, has inspired scalable algorithms (Liu et al., 2019; Taghvaei & Jalali, 2019; Korotin et al., 2021a; Liu et al., 2021; Choi et al., 2024). Similar approaches have also been proposed for the Monge problem with general costs (Asadulaev et al., 2024; Fan et al., 2023). However, these approaches typically rely on adversarial training of two neural networks, which is challenging to manage and often introduces instability and inefficiency into the optimization process. Alternative approaches have investigated dynamical formulations using ordinary differential equations (ODEs) (Yang & Karniadakis, 2020; Onken et al., 2021; Tong et al., 2020; Huguet et al., 2022) and entropic-regularized models involving stochastic differential equations (SDEs) (Genevay et al., 2016; Seguy et al., 2017; Daniels et al., 2021; Gushchin et al., 2023; Zhou et al., 2024). Machine learning algorithms that unify Lagrangian and Eulerian perspectives of Mean Field Control problems Ruthotto et al. (2020); Lin et al. (2021); Zhao et al. (2025) likewise provide a computational framework for OT. Nevertheless, these methods typically require solving systems of differential equations, resulting in substantial computational overhead during both training and inference. Moreover, many existing methods yield bias maps that deviate from the OT solution due to the incorporation of regularization terms into the formulation.

---

[*]Corresponding author. Email: `sjo@math.ucla.edu`

| Method (representative reference) | Optimization | # Networks | OT direction | Sampling | Optimality of $T$ |
|---|---|---|---|---|---|
| Dual Formulation (Asadulaev et al., 2024) | Min-Max | Two | One-way | Direct | No |
| Dynamical Models (Onken et al., 2021) | Min | Single | Bidirectional | Iterative | No |
| **HJ-based (Proposed)** | Min | Single | Bidirectional | Direct | Yes |

Table 1: Comparison of key features across different OT model approaches. Optimality indicates whether an approach is guaranteed, in principle, to recover the true OT map under sufficient network capacity.

**Contributions.** We propose a novel and efficient framework, termed *neural characteristic flow (NCF)*, for solving OT problems via the Hamilton–Jacobi (HJ) equation, whose viscosity solution characterizes the OT map. Despite its strong theoretical foundation for OT, the HJ formulation poses two major challenges: non-uniqueness of solutions and the need to solve ODEs in dynamical formulations. We overcome both by leveraging the method of characteristics and an implicit solution formula (Park & Osher, 2025) to obtain closed-form, bidirectional transport maps without numerical integration of ODEs. NCF uses a single neural network and avoids adversarial training or dual-network architectures, reducing complexity while improving efficiency. Our framework guarantees theoretical consistency with OT optimality conditions and supports a broad class of cost functions, including class-conditional transport. We also provide convergence analysis for Gaussian settings and demonstrate strong empirical performance across datasets of varying dimensions. A comparison of key features across different OT model approaches is summarized in Table 1.

## 2 PRELIMINARY

### 2.1 MONGE'S OPTIMAL TRANSPORT PROBLEM

For a domain $\Omega \subset \mathbb{R}^d$, we denote $\mathscr{P}(\Omega)$ as the space of probability measures on $\Omega$. Let $c : \Omega \times \Omega \to [0, \infty]$ be a cost function that measures the cost of transporting one unit of mass. For $\mu, \nu \in \mathscr{P}(\Omega)$, the classical Monge problem formulates OT as finding a measurable map $T : \Omega \to \Omega$ that pushes forward $\mu$ to $\nu$, i.e., $T_\sharp \mu = \nu$, while minimizing the transportation cost:

$$W_c(\mu, \nu) \coloneqq \inf_{T_\sharp \mu = \nu} \int_\Omega c(\mathbf{x}, T(\mathbf{x})) \, \mathrm{d}\mu(\mathbf{x}). \tag{1}$$

We call a solution $T^*$ to (1) an OT map between $\mu$ and $\nu$. In the case where the cost $c$ is expressed as a function of the difference between the two variables, $T^*$ is characterized as follows:

**Theorem 2.1** (Santambrogio (2015)). *When $c(\mathbf{x}, \mathbf{y}) = \ell(\mathbf{x} - \mathbf{y})$ for a lower semi-continuous (l.s.c.), sub-differentiable, and strictly convex function $\ell : \Omega \to \mathbb{R}$, the optimal map is expressed in terms of the Kantorovich dual potential function $\varphi^* : \Omega \to \mathbb{R}$ as*

$$T^*(\mathbf{x}) = \mathbf{x} + \nabla h(\nabla \varphi^*(\mathbf{x})), \tag{2}$$

*where $h(\mathbf{z}) = \sup_{\mathbf{y} \in \mathbb{R}^d} \{\mathbf{z}^\top \mathbf{y} - \ell(\mathbf{y})\}$ is the Legendre transform of $\ell$.*

### 2.2 DYNAMICAL FORMULATION

Benamou & Brenier (2000) formulate the OT (1) in a continuous-time dynamical formulation:

$$\inf_v \mathbb{E}_\mu \left[ \int_0^{t_f} \ell(v(\mathbf{x}(t), t)) \, \mathrm{d}t \right] \tag{3}$$

$$\text{s.t. } \dot{\mathbf{x}} = v, \ \mathbf{x}(0) \sim \mu, \ \mathbf{x}(t_f) \sim \nu, \tag{4}$$

where the terminal time $t_f > 0$ is typically set to 1. Within this dynamical framework, the associated optimality condition is governed by the *Hamilton–Jacobi (HJ) equation* (Evans, 2022, chapter 10):

$$\begin{cases} \frac{\partial u}{\partial t} + h(\nabla u) = 0 & \text{in } \Omega \times (0, t_f) \\ u = g & \text{on } \Omega \times \{t = 0\}, \end{cases} \tag{5}$$

coupled with the continuity equation $\partial_t \rho + \nabla \cdot (\rho \nabla h(\nabla u)) = 0$ that governs the evolution of the probability distribution. Here, $\nabla u$ denotes the gradient of $u$ with respect to the spatial variable $\mathbf{x}$, and $g$ represents the initial condition, whose explicit analytic form is typically intractable. The optimal velocity field is then determined by $v^* = \nabla h(\nabla u)$, where $u$ is the *viscosity solution* to HJ equation (5).

## 3 RELATED WORKS

Deep learning methods for OT have gained traction following the development of scalable OT solvers (Genevay et al., 2016; Seguy et al., 2017) and WGANs (Arjovsky et al., 2017). Many approaches utilize GAN-based models to approximate OT plans, although they often suffer from training instability and extensive hyperparameter tuning. Another major line of work is based on the Kantorovich dual formulation (Kantorovich, 2006), where the OT map is recovered via optimization of dual potentials, typically parameterized by input convex neural networks (ICNNs) (Amos et al., 2017). While theoretically sound, these methods involve unstable min-max optimization. To address these issues, natural gradient methods have been proposed to improve computational efficacy (Shen et al., 2020; Liu et al., 2024). Regularization techniques such as $L^2$ penalties (Genevay et al., 2016; Sanjabi et al., 2018) and cycle-consistency constraints (Korotin et al., 2019; 2021b) have been proposed, though unconstrained alternatives have shown stronger empirical performance (Korotin et al., 2021a; Fan et al., 2022).

To address the settings where deterministic OT maps may not exist, recent work has considered weak OT formulations (Backhoff-Veraguas et al., 2019). Neural approaches for weak OT and class-conditional transport have been proposed (Korotin et al., 2023; Asadulaev et al., 2024), but may yield spurious solutions under weak quadratic costs. Kernalized costs (Korotin et al., 2022) have been introduced to mitigate this.

OT has also been modeled as a dynamical system via continuous flows (Yang & Karniadakis, 2020; Tong et al., 2020; Onken et al., 2021; Huguet et al., 2022). While expressive, these methods require solving ODEs during training and inference, making them computationally expensive. Entropic and $f$-divergence regularized stochastic models (Daniels et al., 2021; Gushchin et al., 2023) improve smoothness but often rely on Langevin dynamics, which can be biased in high dimensions (Korotin et al., 2019). The HJ equation has been used to improve OT models, with physics-informed neural network (PINN) (Raissi et al., 2019) approaches applying $L^2$ penalties on HJ residuals to improve continuous normalizing flows, ODE-based formulations (Yang & Karniadakis, 2020; Onken et al., 2021), and stochastic variants (Zhang & Katsoulakis, 2023). However, due to the ill-posed nature of the HJ equation, this approach lacks guarantees for recovering the viscosity solution.

## 4 HJ CHARACTERISTIC FLOWS FOR OT

In this section, we represent the OT map through the characteristics of the HJ equation, offering a principled and efficient framework for OT. Note that solving the HJ equation directly is challenging due to its inherent ill-posedness, non-smoothness of solutions, and gradient discontinuities, all of which complicate both theoretical analysis and numerical approximation.

**Method of Characteristics.** The viscosity solution to (5) is theoretically characterized by the following system of *characteristic ordinary differential equations (CODEs)* Evans (2022); Park & Osher (2025):

$$\begin{cases} \dot{\mathbf{x}} = \nabla h(\mathbf{p}) & \text{(6a)} \\ \dot{u} = -h(\mathbf{p}) + \mathbf{p}^\top \nabla h(\mathbf{p}) & \text{(6b)} \\ \dot{\mathbf{p}} = 0, & \text{(6c)} \end{cases}$$

where $\mathbf{p}$ denotes the shorthand for $\nabla u$. CODE for $\mathbf{p}$ (6c) implies that $\mathbf{p}$ remains constant along each characteristic trajectory. Consequently, the characteristics are straight lines of the form $\mathbf{x}(t) = t\nabla h(\mathbf{p}) + \mathbf{x}(0)$, which coincide with the OT map in (2) at terminal time $t = t_f$. From a dynamical perspective, the ODE (4) can be interpreted as the characteristic equations (6a) of the HJ equation that determine the OT map (2). In other words, the transported point $T^*(\mathbf{x})$ of a sample $\mathbf{x} \sim \mu$ corresponds to the terminal position of the characteristic line that originates from $\mathbf{x}$.

Our CODE formulation not only provides a principled construction of the forward transport map but also naturally characterizes the backward map. We denote by $T_\mu^{\nu*}$ the forward OT map transporting $\mu$ to $\nu$, and by $T_\nu^{\mu*}$ the backward map transporting $\nu$ to $\mu$.

**Proposition 4.1** (Bidirectional OT Map). *There exists a viscosity solution $u^*$ to the HJ equation (5) that characterizes both the forward and backward OT maps through its forward and backward characteristic flows:*

$$T_\mu^{\nu*}(\mathbf{x}) = \mathbf{x} + t_f \nabla h\left(\nabla u^*\left(\mathbf{x}, 0\right)\right), \quad \mathbf{x} \sim \mu, \tag{7}$$

$$T_\nu^{\mu*}(\mathbf{y}) = \mathbf{y} - t_f \nabla h\left(\nabla u^*\left(\mathbf{y}, t_f\right)\right), \quad \mathbf{y} \sim \nu. \tag{8}$$

Accordingly, the viscosity solution of the HJ equation enables a bidirectional characterization of the OT map via forward and backward characteristic flows. Notably, since the characteristics are straight lines, both the forward and inverse transport maps admit explicit closed-form expressions. This obviates the need for numerical integration of ODEs typically required in conventional dynamical formulations. Consequently, the CODE-based formulation addresses a key computational bottleneck, enabling efficient and direct computation of bidirectional transport maps.

**Implicit Solution Formula.** Recently, a novel mathematical formulation for the viscosity solution of HJ equations has been developed using the system of CODEs (Park & Osher, 2025). Within this formulation, the viscosity solution admits the following implicit formula:

$$u\left(\mathbf{x}, t\right) = -th\left(\nabla u\right) + t\nabla u^\top \nabla h\left(\nabla u\right) + g\left(\mathbf{x} - t\nabla h\left(\nabla u\right)\right). \tag{9}$$

**Proposition 4.2.** *For OT problems* (1) *where $\ell$ satisfies the conditions in Theorem 2.1, the implicit solution formula* (9) *characterizes the viscosity solution of the HJ equation* (5) *almost everywhere.*

*Proof.* Detailed proof is provided in Appendix A.1. □

## 5 METHODS

### 5.1 OT WITH GENERAL COSTS

We propose a novel deep learning method, termed *neural characteristic flow (NCF)*, for learning bidirectional OT maps under general cost $\ell$ by solving the HJ equation (5) vis its implicit solution formula (9). The HJ equation characterizes the OT map as the gradient of the viscosity solution, ensuring that the resulting map minimizes the given cost functional. When coupled with the continuity equation, it also describes the evolution of probability distributions, thus guaranteeing correct mass transport from source to target. However, jointly solving this coupled system of PDEs is computationally expensive. To address this, the proposed NCF computes the OT map solely through the HJ equation, avoiding the need to solve the continuity equation explicitly.

**Implicit Neural Representation.** We represent the solution $u$ of the HJ equation using an implicit neural representation (INR) $u_\theta : \mathbb{R}^d \times \mathbb{R} \to \mathbb{R}$ parameterized by $\theta$. The network takes the spatial variable $\mathbf{x}$ and temporal variable $t$ as input. By the universal approximation theorem (Hornik et al., 1989; Leshno et al., 1993), the INR can approximate the viscosity solution to the HJ equation. We denote by $T_\mu^\nu[u_\theta]$ as the transport map that aims to map $\mu$ to $\nu$ defined by (7) through $u_\theta$:

$$T_\mu^\nu[u_\theta](\mathbf{x}) = \mathbf{x} + t_f \nabla h\left(\nabla u_\theta\left(\mathbf{x}, 0\right)\right). \tag{10}$$

The backward map $T_\nu^\mu[u_\theta]$ is analogously defined according to (8) via $u_\theta$ evaluated at $t = t_f$.

**HJ-based Training Loss.** While the HJ equation does not directly encode distributional information, it can recover the desired OT map, provided that an appropriate initial function $g$ reflects the relationship between the source and target distributions. However, in practice, where only finite samples from these distributions are available, deriving an analytic form for $g$ is generally intractable. To address this challenge, we introduce a loss term to ensure that the initial condition is appropriately learned during training, thereby steering the HJ solution toward accurately solving the desired OT problem. Specifically, this term enforces alignment between the generated samples obtained via $T[u_\theta]$ and the given target data. This alignment can be effectively quantified using discrepancy measures such as the maximum mean discrepancy (MMD) (Smola et al., 2006), whose value between two distributions $\mu$ and $\nu$ are defined as follows:

$$\mathrm{MMD}(\mu, \nu)^2 = \iint_{\Omega \times \Omega} k(\mathbf{x}, \mathbf{y})\, \mathrm{d}(\mu(\mathbf{x}) - \nu(\mathbf{x}))\, \mathrm{d}(\mu(\mathbf{y}) - \nu(\mathbf{y})), \tag{11}$$

where $k(\cdot, \cdot) : \Omega \times \Omega \to \mathbb{R}$ is a kernel function. The population loss for the MMD is

$$\mathcal{L}_{\mathrm{MMD}}(u_\theta) = \mathrm{MMD}(T_\mu^\nu[u_\theta]_\sharp \mu, \nu)^2. \tag{12}$$

We adopt the negative distance kernel $k(\mathbf{x}, \mathbf{y}) = -\|\mathbf{x} - \mathbf{y}\|_2$, which has proved to handle high-dimensional problems efficiently (Hertrich et al., 2024). With this kernel, the MMD loss becomes the squared energy distance (Rizzo & Székely, 2016).

In our implementation of the implicit solution formula, we replace the initial function $g$ with $u_\theta$ evaluated at $t = 0$, and train the model using the following $\varrho$-weighted loss function

$$\mathcal{L}_{\mathrm{HJ}}(u_\theta) = \iint_{\Omega \times [0, t_f]} \Big( u_\theta + th(\nabla u_\theta) - t\nabla u_\theta^\top \nabla h(\nabla u_\theta) - u_\theta(\mathbf{x} - t\nabla h(\nabla u_\theta), 0) \Big)^2 \mathrm{d}\varrho(\mathbf{x})\, \mathrm{d}t,$$
(13)

where $\varrho$ is a probability measure o $\Omega$ used to weight the residual so that the implicit solution formula (9) is enforced across the entire spatial domain while allowing $\mathcal{L}_{\mathrm{HJ}}$ to be efficiently approximated via Monte Carlo sampling. When $\Omega$ is bounded, a natural choice is the uniform distribution on $\Omega$. In practice, $\varrho$ may also be chosen as a uniform distribution supported on the region covered by the available samples, concentrating computational effort on the portions of the domain most relevant to the transport dynamics.

The overall loss combines the implicit HJ loss and the MMD loss with a weight $\lambda > 0$:

$$\mathcal{L}_{\mathrm{HJ}}(u_\theta) + \lambda \mathcal{L}_{\mathrm{MMD}}(u_\theta).$$
(14)

We refer to Appendix B for practical choices of $\varrho$ and the Monte Carlo estimation of the loss.

**Advantages of the Proposed Approach.** Our method offers several key advantages over existing OT frameworks, as summarized in Table 1. First, it jointly learns both forward and backward OT maps using a single neural network in one training phase. This contrasts with prior methods that require multiple networks, either due to the lack of invertibility or the use of adversarial dual formulations—leading to increased model complexity and training cost. Our method also avoids the instability of min-max optimization common in dual approaches, resulting in more stable training. Second, unlike dynamical OT models that require solving ODEs or SDEs, we use the method of characteristics to obtain OT maps in closed form. This removes the need for iterative solvers and improves sampling efficiency at both training and inference time. Third, our model directly incorporates the HJ equation via an implicit solution formula that reliably recovers the viscosity solution, as supported by the numerical results in Section 6. This not only aligns with the theoretical optimality conditions of OT but also helps identify and correct deviations from the target solution during training. Finally, our framework supports a broad class of cost functions beyond the quadratic case, offering greater flexibility and wider applicability across OT tasks.

### 5.2 THEORETICAL ANALYSES

In this section, we present theoretical analyses of our method, focusing on the OT problem with $\Omega = \mathbb{R}^d$ and the quadratic cost $\ell(\cdot) = \frac{1}{2}\|\cdot\|^2$, for which the corresponding Hamiltonian is given by $h(\cdot) = \frac{1}{2}\|\cdot\|^2$ as well. We prove that the minimizer of the loss (14) exactly recovers the true OT maps. Moreover, in the Gaussian setting, we establish stability analysis by showing that a small loss guarantees convergence to the true solution.

**Consistency Analysis** With some mild convexity assumption, we establish that the minimizer of (14) leads precisely to the optimal transport map.

**Theorem 5.1** (Consistency of loss). *Suppose the probability distributions $\mu, \nu$ have finite second moments and $\varrho \in \mathscr{P}(\mathbb{R}^d)$ is strictly positive. Assume $u \in C^1_{loc}(\mathbb{R}^d \times [0, t_f])$, and define $u_1(\cdot) := u(\cdot, t_f) \in C^2_{loc}(\mathbb{R}^d)$ with $\nabla u_1 \in L^2(\mathbb{R}^d, \mathbb{R}^d; \nu)$. If $u$ minimizes the loss functional (14), i.e.,*

$$\mathcal{L}_{HJ}(u) + \lambda \mathcal{L}_{MMD}(u) = 0,$$

*and the map $T_\nu^\mu[u]$ is bijective with its Jacobian $D_x T_\nu^\mu[u](x)$ is positive definite for any $x \in \mathbb{R}^d$, then $T_\nu^\mu[u]$ and $T_\mu^\nu[u]$ are the optimal transport maps from $\nu$ to $\mu$, and vice versa.*

The proof is provided in Appendix A.2. See also Remark A.5 for further discussion on the monotonicity condition for $D_x T_\nu^\mu[u]$.

*Remark* 5.2 (On regularity assumption of $u$). It is worth noting that the transport curves associated with the Wasserstein-2 OT problem do not intersect for $t \in [0, t_f]$ (cf. Chap. 8 of (Villani et al., 2008)). Since these curves constitute the characteristics of the HJ equation associated with the OT problem, we can expect classical solutions to the HJ equation, provided that $\mu$ and $\nu$ admit

sufficiently regular density functions. This observation motivates the regularity assumption on $u$ in Theorem 5.1. Moreover, $u$ is parametrized with neural networks in practice, which naturally preserve the regularity.

**Stability Analysis** The loss (14) also exhibits favorable stability properties, which we illustrate in the Gaussian setting. Let $\mu = N(\mathbf{b}_\mu, \Sigma_\mu)$, $\nu = N(\mathbf{b}_\nu, \Sigma_\nu)$, then the OT map is

$$T_\mu^{\nu*}(\mathbf{x}) = A(\mathbf{x} - \mathbf{b}_\mu) + \mathbf{b}_\nu, \tag{15}$$

where $A := \Sigma_\mu^{-\frac{1}{2}} (\Sigma_\mu^{\frac{1}{2}} \Sigma_\nu \Sigma_\mu^{\frac{1}{2}})^{\frac{1}{2}} \Sigma_\mu^{-\frac{1}{2}}$. For analytical tractability, we consider a simplified quadratic parameterization $u_\theta(\mathbf{x}, t) = -(\frac{1}{2}\mathbf{x}^\top \theta_2(t)\mathbf{x} + \theta_1(t)^\top \mathbf{x} + \theta_0(t))$, where $\theta = [\theta_2(\cdot), \theta_1(\cdot), \theta_0(\cdot)]$ : $[0, t_f] \to \mathbb{R}_{\text{sym}}^{d \times d} \times \mathbb{R}^d \times \mathbb{R}$. Although this represents a restricted subclass of neural networks, it permits rigorous analysis and yields insights relevant to more general architectures.

**Assumption 5.3.** $\theta(t)$ is bounded by $K$ and $K$-Lipschitz. $\|\mathbf{b}_\mu\|, \|\mathbf{b}_\nu\|, \|\Sigma_\mu\|_F, \|\Sigma_\nu\|_F \le K$. $A$ is strictly positive definite with smallest eigenvalue $\lambda_A > 0$.

**Theorem 5.4** (Stability of loss). *Under Assumption 5.3, the errors for $u_\theta$ and $T_\mu^\nu[u_\theta]$ satisfy*

$$\|u_\theta - u^*\|_{L^\infty([-1,1]^d)} + \|T_\mu^\nu[u_\theta] - T_\mu^{\nu*}\|_{L^\infty([-1,1]^d)} \le C \left( \mathcal{L}_{HJ}^{\frac{1}{3}} + \mathcal{L}_{MMD}^{\frac{1}{4}} \right), \tag{16}$$

*where $u^*$ and $T_\mu^{\nu*}$ are the true solution and OT map. $C$ only depends on $d$, $K$ and $\lambda_A$.*

The theorem implies that sufficiently small loss guarantees convergence of the approximate solution $u_\theta$—and consequently the resulting transport map $T_\mu^\nu[u_\theta]$—to their true counterparts. Furthermore, the proof shows that while multiple transport maps may minimize the MMD loss, the implicit HJ loss ensures that the OT map is uniquely recovered. The detailed description and proof for the theorem are deferred to Appendix A.3.

## 5.3 CLASS-CONDITIONAL OT

We extend our HJ-based framework to class-conditional OT, transporting source to target independently within each of the $K$ labeled classes so as to preserve label consistency and class-specific structure. This formulation is particularly well-suited for domain adaptation and class-conditional generative modeling, where preserving class-specific features is crucial.

The OT map between samples of the $k$-th class must satisfy the HJ equation within the support of the corresponding class-specific distribution, as dictated by the optimality condition. Consequently, the global transport map $T_\mu^{\nu*}$ satisfies the HJ equation (5) across the entire domain. Although non-differentiable regions may arise due to intersections between transport maps of different classes, such discontinuities occur primarily in the boundaries between class supports. Since the gradient of the HJ solution is computed only within the support of each class-specific distribution, the transport map remains expressible in these regions. Accordingly, we retain the implicit HJ loss function (13) and modify the MMD loss to account for class conditioning as follows:

$$\mathcal{E}_{\text{class}}\left((T_\mu^\nu[u_\theta])_\sharp \mu, \nu\right) = \frac{1}{K} \sum_{k=1}^{K} \mathcal{E}((T_\mu^\nu[u_\theta])_\sharp \mu_k, \nu_k). \tag{17}$$

A similar approach was proposed by Asadulaev et al. (2024).

## 6 EXPERIMENTAL RESULTS

We evaluate the effectiveness of the proposed *neural characteristic flow (NCF)* across diverse OT tasks. All experiments in this section employ the quadratic cost function $\ell = \frac{1}{2} \|\cdot\|_2^2$, which is the canonical cost associated with the Wasserstein-2 distance. Computations were performed on a single NVIDIA GV100 (TITAN V) GPU. Further implementation details are provided in Appendix B.

## 6.1 UNCONDITIONAL OT

### 6.1.1 2D TOY EXAMPLES

We test the proposed NCF on a 2D toy dataset. We also compare our model with the neural optimal transport (NOT) framework (Korotin et al., 2023), including both the strong (deterministic) and

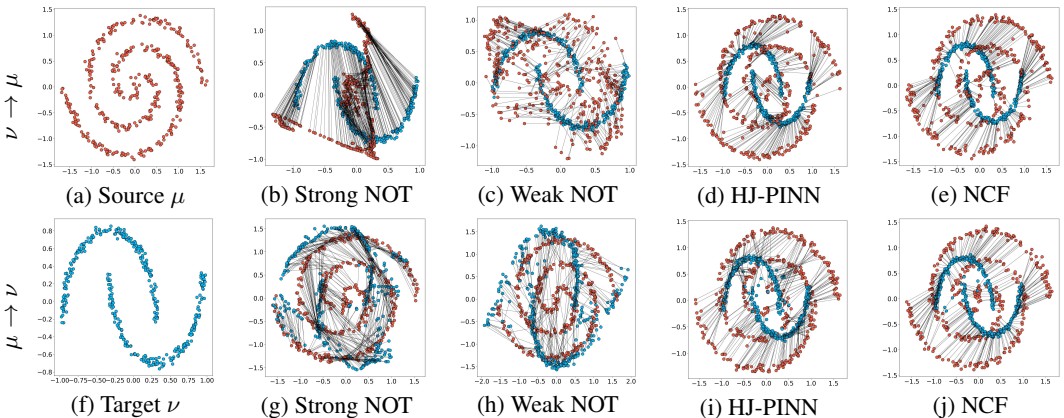

Figure 1: *Swiss roll ($\mu$) $\rightleftarrows$ Double moons ($\nu$)*: The top row shows transport in the direction $\nu \to \mu$, and the bottom row shows $\mu \to \nu$. The leftmost column displays $\mu$ and $\nu$ for reference.

weak (stochastic) variants. Since NOT directly parameterizes the transport map, it requires separate training for each transport direction. Additionally, we include an ablation study replacing our implicit solution formula loss (13) with a PINN loss on the HJ equation, referred to as HJ-PINN.

Figure 1 shows bidirectional transport results on 2D distributions. In addition to visualizing the transported distributions, we overlay the learned transport maps as black solid lines to assess whether each model has captured an OT plan. For weak NOT, the map is the average over noise inputs, as in the original work. Compared to all baselines, our method captures source and target distributions more accurately and learns transport maps closely aligned with the optimal solution. Strong NOT produces noisy, incoherent transport. Weak NOT performs better but still shows overlapping trajectories, indicating an incomplete OT representation. HJ-PINN yields noisy, intersecting transport paths, suggesting failure to learn OT dynamics. In contrast, our model learns accurate OT maps without trajectory crossings. Moreover, unlike NOT, which requires four separate networks for bidirectional training, our method achieves more accurate bidirectional transport with a single network. These results highlight the superior accuracy and efficiency of our approach. For further experimental results on the 2D example, please refer to Appendix C.1.

### 6.1.2 EVALUATION ON HIGH-DIMENSIONAL GAUSSIANS

For general distributions, the ground truth OT solution is unknown, making quantitative evaluation challenging. To enable precise assessment, we consider the Gaussian case: $\mu = \mathcal{N}(\mathbf{0}, \Sigma_\mu)$ and $\nu = \mathcal{N}(\mathbf{0}, \Sigma_\nu)$, where a closed-form solution is available via (15). Following Korotin et al. (2021a), we vary the dimension $d$ from 2 to 64, with $\Sigma_\mu$ and $\Sigma_\nu$ generated using random eigenvectors uniformly sampled on the unit sphere and logarithms of eigenvalues drawn uniformly from $[-2, 2]$. In addition to strong NOT and HJ-PINN, we evaluate several established OT methods: MM-v1 (Taghvaei & Jalali, 2019; Korotin et al., 2021a), which solves a min-max dual problem using input-convex neural networks (ICNNs), alternating between optimizing the potential and its convex conjugate; MM:R (Korotin et al., 2021a) also employs a min-max framework but does not enforce convexity, instead learning separate networks for forward and backward maps via a negative Wasserstein loss combined with a conjugacy loss; LS (Seguy et al., 2017), which addresses the dual problem via entropic regularization; and WGAN-QC (Liu et al., 2019), which employs a WGAN architecture with quadratic cost. Except for NOT—which directly parameterizes transport maps—all models use a shared architecture for potential functions.

Performance is measured using the unexplained variance percentage (UVP) (Korotin et al., 2019), which quantifies the $L^2$ error of the estimated transport map, normalized by $\text{Var}(\nu)$. Computational efficiency is also evaluated in terms of training and inference time, peak memory usage, and memory required to store bidirectional OT maps. Table 2 reports UVP across models and dimensions, while Figure 2 summarizes computational metrics.

Our method consistently yields accurate OT maps with favorable scaling behavior, outperforming NOT, WGAN-QC, and LS, which exhibit greater deviation from the ground-truth transport. In higher dimensions, the performance of NCF is slightly reduced, which we attribute to the fact that, unlike baseline methods that parameterize separate networks for forward and backward maps, our approach represents both directions using a single network. Specifically, NCF approximates both

Table 2: *Quantitative evaluation on Gaussian distributions.* UVP ($\downarrow$) is measured across different OT methods as the data dimension $d$ increases.

| Method | $d = 2$ | $d = 4$ | $d = 8$ | $d = 16$ | $d = 32$ | $d = 64$ |
|---|---|---|---|---|---|---|
| NOT | 77.248 | 125.419 | 114.056 | 176.086 | 182.287 | 196.831 |
| WGAN-QC | 1.596 | 5.897 | 31.0367 | 59.314 | 113.237 | 141.407 |
| LS | 5.806 | 9.781 | 15.963 | 25.232 | 41.445 | 55.360 |
| MM-v1 | 0.161 | 0.172 | 0.173 | 0.210 | 0.374 | 0.415 |
| MM:R | 0.012 | 0.048 | 0.0117 | 0.202 | 0.354 | 0.604 |
| HJ-PINN | 0.080 | 0.069 | 0.163 | 0.458 | 0.576 | 1.683 |
| **NCF** | 0.010 | 0.021 | 0.086 | 0.146 | 0.436 | 0.858 |
| **NCF-Adaptive** | 0.010 | 0.022 | 0.090 | 0.155 | 0.307 | 0.407 |

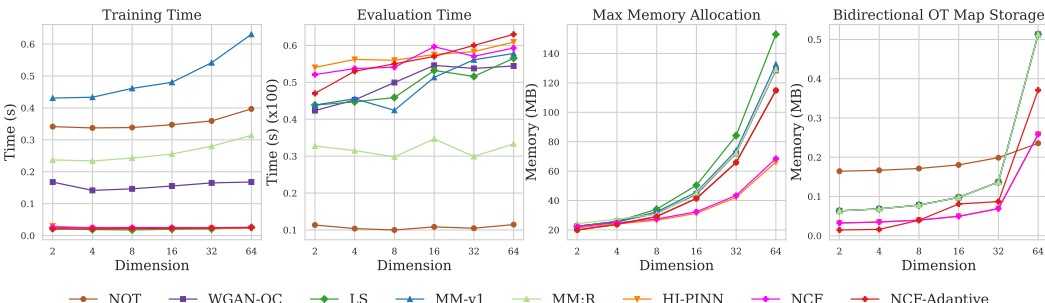

Figure 2: *Computational comparison.* Training time (s/epoch), evaluation time (s/epoch), peak memory (MB) during training, and memory (MB) for storing bidirectional OT maps are reported.

maps using exactly half the network capacity required by methods such as MM-v1. This parameter reduction is advantageous in low dimensions; however, in higher dimensions (e.g., $d = 32, 64$), the limited capacity constrains the simultaneous learning of both directional maps. To address this, we conducted experiments with a slightly enlarged network for high-dimensional cases—without modifying any training hyperparameters—denoted as NCF-Adaptive in the results. With this adjustment, our method recovers performance in high dimensions while still using fewer parameters than MM-v1 or MM:R.

Moreover, baseline methods such as MM-v1 and MM:R incur substantially longer training times and significantly higher memory usage. In addition, MM:R, which relies on min–max optimization, exhibits noticeable training instability (see Appendix C.2.) In contrast, our approach avoids expensive nested min–max optimization and leverages a single network, resulting in faster, stable, and more memory-efficient training. At inference, NOT achieves the lowest latency due to its direct map parameterization, whereas other methods, including ours, require gradient-based evaluation, introducing additional overhead. This overhead, however, decreases with increasing dimension. Finally, comparison with HJ-PINN highlights the superior effectiveness of our implicit loss in approximating the viscosity solution to the underlying HJ equation.

### 6.1.3 APPLICATION TO COLOR TRANSFER

We employ the dataset provided by CycleGAN (Zhu et al., 2017) for image color transfer experiments. From each of the three available groups of image pairs, we selected 10 representative pairs. For each pair, we perform both forward and backward color transfer. To evaluate the effectiveness of our model, we include comparisons with two widely used classical color transfer methods: a standard per-channel histogram matching technique and the approach of Reinhard et al. (2001), which aligns the mean and standard deviation of color channels. These baselines represent statistical methods that do not rely on OT, providing a complementary perspective on performance. We include NOT and MM-v1 as deep learning OT baselines.

To quantitatively evaluate color fidelity and distributional consistency, we employ two widely used histogram-based metrics: Earth Mover's distance (EMD) and histogram intersection (HI), summarized in Table 3. Across all three domains, our method consistently achieves superior performance compared to all baselines in both metrics. In particular, our proposed method exhibits superior robustness in handling more complex and multimodal color distributions compared to MM-v1, especially in contrast to the simpler Gaussian settings examined in the previous section. Qualitative results are provided in Appendix C.3.

Table 3: *Quantitative evaluation of color transfer.* Earth mover distance (EMD) and histogram intersection (HI) between color distributions of target and transported images are reported.

| Method | Winter-Summer | | Monet-Photograph | | Gogh-Photograph | |
|---|---|---|---|---|---|---|
| | EMD ($\downarrow$) | HI ($\uparrow$) | EMD ($\downarrow$) | HI ($\uparrow$) | EMD ($\downarrow$) | HI ($\uparrow$) |
| HisMatching | 0.0012 | 0.7296 | 0.0013 | 0.7532 | 0.0010 | 0.7668 |
| Reinhard | 0.0013 | 0.6255 | 0.0012 | 0.7255 | 0.0009 | 0.7406 |
| NOT | 0.0008 | 0.8002 | 0.0008 | 0.8210 | 0.0008 | 0.8247 |
| MM-v1 | 0.0014 | 0.7295 | 0.0011 | 0.7722 | 0.0007 | 0.8265 |
| MM:R | 0.0015 | 0.6404 | 0.0013 | 0.6810 | 0.0018 | 0.6260 |
| NCF | **0.0005** | **0.8914** | **0.0004** | **0.9174** | **0.0003** | **0.9117** |

(a) Distributions    (b) Unconditional NCF    (c) GNOT    (d) Class-conditional NCF

Figure 3: *2D class-conditional OT.* The leftmost column displays $\mu$ (red) and $\nu$ (blue), with class labels indicated by distinct markers. In the remaining columns, blue dots denote transported samples, while solid black and dotted gray lines represent the learned transport maps for each class.

## 6.2 CLASS-CONDITIONAL OT

### 6.2.1 2D TOY EXAMPLES

We present experimental results on a 2D synthetic dataset consisting of class-labeled samples, designed to evaluate class-conditional OT. To assess the ability of the proposed class-conditional NCF variant to model class-guided transport, we compare it against an unconditional NCF, which does not utilize label information. Furthermore, to benchmark our method against existing approaches, we include NOT with general cost functionals (GNOT) (Asadulaev et al., 2024), a recent model designed to perform class-conditional OT.

Figure 3 presents results on a 2D Gaussian mixture dataset, where each data point is associated with a class label. The unconditional NCF, lacking access to label information, learns a global transport map that ignores class structure, aligning source and target points purely based on $W^2$ distance. In contrast, both GNOT and the proposed class-conditional NCF learn separate transport maps per class. However, GNOT exhibits intersecting transport paths between classes, suggesting suboptimality with respect to the transport cost. The class-conditional NCF effectively disentangles transport across classes and yields maps that closely approximate the optimal solutions. These results highlight the accuracy and effectiveness of our approach, grounded in a CODE-based formulation of the HJ equation, for learning class-conditional transport in structured settings.

### 6.2.2 MNIST & FASHION MNIST

We apply our model to the MNIST (LeCun, 1998) and Fashion MNIST (Xiao et al., 2017) datasets, each comprising 10 classes. Given their substantially lower intrinsic dimensionality relative to the ambient space (Pope et al., 2021), we solve class-conditional OT problems in latent spaces obtained via $\beta$-VAEs (Higgins et al., 2017); see Appendix B.4 for details.

We consider transport from each Fashion MNIST class to its corresponding MNIST class; additional class-conditional OT tasks on MNIST are provided in Appendix C.4. We compare against baselines from Asadulaev et al. (2024), including NOT and GNOT, as well as a domain adaptation OT method

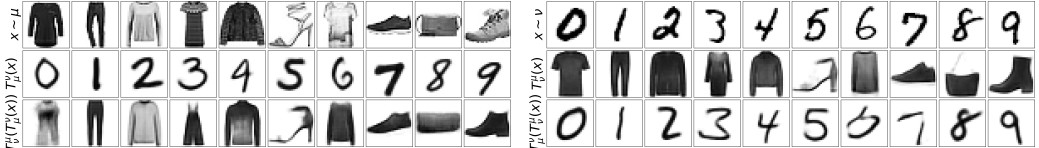

Figure 4: *Class-conditional OT between MNIST and Fashion MNIST*. **Left**: Transport Fashion MNIST data $x$ forward $T_\mu^\nu[u_\theta](x)$ then backward $T_\nu^\mu[u_\theta](T_\mu^\nu[u_\theta](x))$. **Right**: Transport MNIST data $x$ backward $T_\nu^\mu[u_\theta](x)$ then forward $T_\mu^\nu[u_\theta](T_\nu^\mu[u_\theta](x))$. The first row shows the source data, the second & third rows present the data being transported back and forth using the learned map.

Table 4: Comparison of the accuracy and FID scores for the forward class-conditioned maps (Fashion MNIST → MNIST) learned using different methods. The accuracy and FID scores for the baseline methods are adopted from (Asadulaev et al., 2024).

| Metric | NOT $L^2$ cost | GNOT Stochastic map | Discrete OT SinkhornLpL1 | AugCycleGAN | MUNIT | NCF [Ours] |
|---|---|---|---|---|---|---|
| Accuracy(%) ↑ | 10.96 | 83.22 | 10.67 | 12.03 | 8.93 | **83.42** |
| FID ↓ | 7.51 | **5.26** | >100 | 26.35 | 7.91 | 18.27 |

(Courty et al., 2016; Flamary et al., 2021) that uses discrete OT with label-supervised regularization. Additionally, we evaluate unsupervised image translation methods AugCycleGAN (Almahairi et al., 2018) and MUNIT (Huang et al., 2018).

Figure 4 shows bidirectional transported samples by NCF; uncurated results are in Appendix C.4. These results qualitatively demonstrate NCF's ability to perform bidirectional, class-conditional OT on real images. For quantitative evaluation, we report Fréchet Inception Distance (FID) (Heusel et al., 2017) and class-wise accuracy, which measures how well the class identity is preserved during transport, in Table 4. Our method achieves the highest accuracy, indicating its strong class-aware transport performance. Although the FID score is relatively high, this is largely due to the discrepancy introduced by the VAE decoder. To isolate this effect, we compute the FID between the NCF outputs and the VAE-decoded images. The resulting low score 2.73 indicates that the transport map in the latent space faithfully reproduces the target distribution. This is further supported by the KDE plots in Figure 15, showing close alignment between the transported and target latent distributions along principal components.

## 7 CONCLUSION

We introduced a theoretically grounded OT framework that recovers forward and backward maps in closed form via HJ characteristics. The resulting single-network, integration-free algorithm gives accurate, bidirectional maps, supports a broad class of costs, and extends to class-conditional transport with pairwise MMD alignment. We establish consistency and stability. Several tasks including synthetic, color-transfer, and MNIST demonstrate accuracy and efficiency of our algorithm.

Future directions include improving high-dimensional performance beyond latent-space implementations by developing more efficient gradient evaluations and scalable network designs. Extending the stability analysis to general neural architectures would provide a deeper theoretical understanding of our method and its convergence behavior. Moreover, as demonstrated by our numerical experiments, the proposed NCF framework offers an accurate and computationally efficient approach for estimating class-conditioned transport maps. Exploring real-world applications in this direction—such as domain adaptation Nguyen et al. (2024), cross-domain retrieval Chuang et al. (2023), and biomedical conditional modeling Manupriya et al. (2024)—with a particular emphasis on pursuing improved semantic correctness, represents an important avenue for future research.

## 8 ACKNOWLEDGMENTS

Y. Park and S. Osher were supported in part by DARPA under grant HR00112590074. S. Liu, M. Zhou, and S. Osher were supported in part by NSF under grant 2208272. The work of S. Liu and M. Zhou was supported in part by the AFOSR YIP award No. FA9550-23-1-0087. M. Zhou and S. Osher were additionally supported by AFOSR under MURI grant N00014-20-1-2787. S. Osher was further supported by ARO under grant W911NF-24-1-0157 and by NSF under grant 1554564.

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

## A PROOF

### A.1 PROOF OF PROPOSITION 4.2

Since $\ell$ is l.s.c., sub-differentiable, and strictly convex, its Legendre transform $h$ is also l.s.c., sub-differentiable, and strictly convex. For such a Hamiltonian $h$, it has been proven in (Park & Osher, 2025) that the viscosity solution satisfies the implicit solution formula (9) almost everywhere. Consequently, the optimal solution to these OT problem is characterized by the implicit solution formula.

### A.2 PROOF OF THEOREM 5.1

**Lemma A.1.** *Suppose that $\mu, \nu$ are probability distributions on $\mathbb{R}^d$. Suppose $\mu$ has finite second order moment, i.e., $\int_{\mathbb{R}^d} \|x\|^2 d\mu(x) < \infty$. Assume that $\psi : \mathbb{R}^d \to \mathbb{R} \bigcup \{+\infty\}$ is convex and differentiable $\mu$-a.e. Set $T = \nabla\psi$ and suppose $\int_{\mathbb{R}^d} \|T(x)\|^2 d\mu(x) < +\infty$. Then $T$ is optimal for the transport cost $\frac{1}{2}\|x - y\|^2$ between $\mu$ and $\nu$.*

This lemma is proved in Theorem 1.48 in Santambrogio (2015).

**Lemma A.2.** *Suppose $T : \mathbb{R}^d \to \mathbb{R}^d$ is a bijective map on $\mathbb{R}^d$. Assume that $T$ is strictly monotone, that is, $\langle T(x) - T(y), x - y \rangle > 0$ for arbitrary $x, y \in \mathbb{R}^d$, $x \neq y$. Here, we denote $\langle \cdot, \cdot \rangle$ as the $\ell^2$ inner product on $\mathbb{R}^d$. Suppose $S : \mathbb{R}^d \to \mathbb{R}^d$ satisfies $S \circ T = \mathrm{Id}$, then $S$ is also bijective, and strictly monotone.*

*Proof.* It is straightforward to verify that $S$ is surjective and injective, and it is thus bijective. Now, for arbitrary $x, y \in \mathbb{R}^d$, $x \neq y$, there exists unique $x', y' \in \mathbb{R}^d$, $x' \neq y'$, such that $x = T(x'), y = T(y')$. Thus, we have $S(x) = S(T(x')) = x', S(y) = S(T(y')) = y'$, and $\langle S(x) - S(y), x - y \rangle = \langle x' - y', T(x') - T(y') \rangle > 0$. $\square$

For brevity, in the following discussion, we denote $C^k_{\mathrm{loc}}(\mathbb{R}^d)$ by $C^k(\mathbb{R}^d)$ for any $k \in \mathbb{N}$. We denote $\mathbf{O}_d$ as the $d \times d$ zero matrix. For symmetric matrices $A, B \in \mathbb{R}^{d \times d}$, we denote $A \succ B$ if $A - B$ is positive definite.

**Theorem A.3.** *Given the probability distributions $\mu, \nu \in \mathscr{P}(\mathbb{R}^d)$ with $\int_{\mathbb{R}^d} \|x\|^2 \, d\mu, \int_{\mathbb{R}^d} \|x\|^2 \, d\nu < +\infty$, suppose $u_0 \in C^1_{loc}(\mathbb{R}^d), u_1 \in C^2_{loc}(\mathbb{R}^d), \nabla u_1 \in L^2(\mathbb{R}^d, \mathbb{R}^d; \nu)$ satisfy*

$$u_0(x - t_f \nabla u_1(x)) = u_1(x) - \frac{t_f}{2}\|\nabla u_1(x)\|^2, \quad \forall\, x \in \mathbb{R}^d. \tag{18}$$

*Assume further that $(\mathrm{Id} + t_f \nabla u_0(\cdot))_\sharp \mu = \nu$. If the mapping $\mathrm{Id} - t_f \nabla u_1(\cdot) : \mathbb{R}^d \to \mathbb{R}^d$ is bijective and $\mathbf{I}_d - t_f \nabla^2 u_1(x) \succ \mathbf{O}_d$, then $\mathrm{Id} + t_f \nabla u_0(\cdot)$ is the optimal transport from $\mu$ to $\nu$, and $\mathrm{Id} - t_f \nabla u_1(\cdot)$ is the optimal transport map from $\nu$ to $\mu$.*

*Proof.* We split the proof into several steps:

**Step 1.** We first prove the fact that $\nabla u_0(x - t_f \nabla u_1(x, t)) = \nabla u_1(x)$ for arbitrary $x \in \mathbb{R}^d$. This can be shown by taking gradient with respect to $x$ on both sides of (18):

$$(\mathbf{I}_d - t_f \nabla^2 u_1(x))\nabla u_0(x - t_f \nabla u_1(x)) = \nabla u_1(x) - t_f \nabla^2 u_1(x)\nabla u_1(x).$$

Re-arrange this equation yields

$$(\mathbf{I}_d - t_f \nabla^2 u_1(x))(\nabla u_0(x - t_f \nabla u_1(x)) - \nabla u_1(x)) = 0.$$

As $\mathbf{I}_d - t_f \nabla^2 u_1(x) \succ \mathbf{O}_d$, we deduce that $\nabla u_0(x - t_f \nabla u_1(x)) = \nabla u_1(x)$ for arbitrary $x \in \mathbb{R}^d$.

**Step 2.** For the sake of brevity, we denote $T_0(\cdot) := \mathrm{Id} + t_f \nabla u_0(\cdot)$, and $T_1(\cdot) := \mathrm{Id} - t_f \nabla u_1(\cdot)$. We prove that $T_0 \circ T_1 = \mathrm{Id}$. This can be derived by straightforward calculation:

$$\begin{aligned}
T_0(T_1(x)) = T_1(x) + t_f \nabla u_0(T_1(x)) &= x - t_f \nabla u_1(x) + t_f \nabla u_0(x - t_f \nabla u_1(x)) \\
&= x + t_f(\nabla u_0(x - t_f \nabla u_1(x)) - \nabla u_1(x)) = x,
\end{aligned}$$

for any $x \in \mathbb{R}^d$.

**Step 3.** Before we prove the assertion, we show that $\int_{\mathbb{R}^d} \|T_0(x)\|^2 \, \mathrm{d}\mu < +\infty$, $\int_{\mathbb{R}^d} \|T_1(x)\|^2 \, \mathrm{d}\nu < +\infty$. The latter inequality can be shown using

$$\int_{\mathbb{R}^d} \|T_1(x)\|^2 \, \mathrm{d}\nu \leq 2 \left( \int_{\mathbb{R}^d} \|T_1(x) - x\|^2 \, \mathrm{d}\nu + \int_{\mathbb{R}^d} \|x\|^2 \, \mathrm{d}\nu \right)$$

$$= 2(t_f^2 \|\nabla u_1\|_{L^2(\nu)}^2 + \int_{\mathbb{R}^d} \|x\|^2 \, \mathrm{d}\nu) < +\infty.$$

For the former inequality, we have

$$\int_{\mathbb{R}^d} \|T_0(x)\|^2 \, \mathrm{d}\mu \leq 2t_f^2 \int_{\mathbb{R}^d} \|\nabla u_0(x)\|^2 \, \mathrm{d}\mu + 2 \int_{\mathbb{R}^d} \|x\|^2 \, \mathrm{d}\mu.$$

Using the fact that $T_{1\sharp}\nu = \mu$, The first term above equals $\int_{\mathbb{R}^d} \|\nabla u_0(T_1(x))\|^2 \, \mathrm{d}\nu = \int_{\mathbb{R}^d} \|\nabla u_1(x)\|^2 \, \mathrm{d}\nu = \|\nabla u_1\|_{L^2(\nu)} < +\infty$, where we use the fact that $\nabla u_0(T_1(x)) = \nabla u_1(x)$ established in step 1. This accomplishes the proof for $\int_{\mathbb{R}^d} \|T_0(x)\|^2 \, \mathrm{d}\mu < +\infty$.

**Step 4.** We now prove the conclusion. Firstly, recall that $DT_1(x) = \mathbf{I}_d - t_f \nabla^2 u_1(x) \succ \mathbf{O}_d$, this leads to the fact that $T_1(\cdot)$ is strictly monotone. We now apply Lemma A.2 to show that $T_0$ is also bijective, and strictly monotone.

$T_0$ is bijective suggests that $T_1(\cdot)$ is the inverse mapping of $T_0(\cdot)$, this leads to $T_{1\sharp}\nu = \mu$. As $T_1(\cdot) = \nabla\varphi(\cdot)$ with $\varphi(\cdot) = \frac{\|\cdot\|^2}{2} - t_f u_1(\cdot)$ being convex, combining the fact established in step 3, Lemma A.1 proves that $T_1$ is the optimal transport map from $\nu$ to $\mu$.

Furthermore, $T_0$ is strictly monotone yields that $DT_0(x) = \mathbf{I}_d + t_f \nabla^2 u_0(x) \succ \mathbf{O}_d$, this indicates that $T_0(\cdot) = \nabla\left( \frac{\|\cdot\|^2}{2} + t_f u_0(\cdot) \right)$ is the gradient of a convex function. Combining with the fact that $T_{0\sharp}\mu = \nu$ and finite $L^2(\mu)$ cost for $T_0$, we deduce that $T_0$ is the optimal transport map form $\mu$ to $\nu$. $\square$

Recall $\varrho \in \mathscr{P}(\mathbb{R}^d)$, and the implicit HJ loss $\mathcal{L}_{\mathrm{HJ}}(u)$ defined in (13). The following Theorem is a natural corollary of Theorem A.3.

**Theorem A.4** (Consistency result). *Suppose the probability distributions $\mu, \nu$ possess finite second-order moments. Assume $u \in C^1_{loc}(\mathbb{R}^d \times [0, t_f])$, and define $u_1(\cdot) := u(\cdot, t_f) \in C^2(\mathbb{R}^d)$ with $\nabla u_1 \in L^2(\mathbb{R}^d, \mathbb{R}^d; \nu)$. Denote hyperparameter $\lambda > 0$. Assume that $\varrho$ is a strictly positive probability measure on $\mathbb{R}^d$. Suppose that $u$ minimizes the loss functional, i.e.,*

$$\mathcal{L}_{HJ}(u) + \lambda \mathcal{L}_{MMD}(u) = 0.$$

*Assume further that the map $T_\nu^\mu[u] : \mathbb{R}^d \to \mathbb{R}^d$ is bijective and its Jacobian $D_x T_\nu^\mu[u](x) \succ \mathbf{O}_d$ for any $x \in \mathbb{R}^d$. Then the maps $T_\nu^\mu[u]$ and $T_\mu^\nu[u]$ are the optimal transport maps from $\nu$ to $\mu$ and from $\mu$ to $\nu$, respectively.*

*Proof.* Denote $E(x, t) = (u(x - t_f \nabla u(x, t_f), 0) - u(x, t_f) + \frac{t_f}{2}\|\nabla u(x, t_f)\|^2)^2$, we have $E \in C(\mathbb{R}^d \times [0, t_f])$, and $E(x, t) \geq 0$ on $\mathbb{R}^d \times [0, t_f]$. Denote $m$ as the Lebesgue measure on $[0, t_f]$. Then we have

$$\mathcal{L}_{\mathrm{HJ}}(u) = \iint_{\mathbb{R}^d \times [0, t_f]} E(x, t) \, \mathrm{d}\varrho \, \mathrm{d}m = 0.$$

Tonelli's Theorem Fremlin (2000) leads to

$$\int_{\mathbb{R}^d} \left( \int_0^{t_f} E(x, t) \, \mathrm{d}t \right) \mathrm{d}\varrho(x) = 0.$$

Now, Lemma A.6 indicates that $\phi(x) := \int_0^{t_f} E(x, t) \, \mathrm{d}t$ is continuous on $\mathbb{R}^d$. And we have $\int_{\mathbb{R}^d} \phi(x) \, \mathrm{d}\varrho(x) = 0$. Lemma A.7 suggests that $\phi(x) = 0$, which is $\int_0^{t_f} E(x, t) \, \mathrm{d}t = 0, \forall x \in \mathbb{R}^d$. Using a similar argument as presented in the proof of Lemma A.7 shows $E(x, t) = 0$ for all $x \in \mathbb{R}^d, t \in [0, t_f]$.

Now applying Theorem A.3 with $u_0 = u(\cdot, 0)$ and $u_1 = u(\cdot, t_f)$ proves the assertion. $\square$

*Remark* A.5 (On the Monotonicity Condition). The monotonicity condition $D_x T^\mu_\nu[u] \succ \mathbf{O}_d$ is closely related to the $c$-concavity of $u_1$, which provides a sufficient condition in OT theory (Villani et al., 2008; Santambrogio, 2015). In practice, our proposed method successfully computes the OT map for the benchmark problems even without explicitly enforcing this condition. Nonetheless, it remains an interesting direction for future research to investigate efficient strategies for enforcing monotonicity and to assess the potential benefits of doing so.

**Lemma A.6.** *Suppose $E \subset \mathbb{R}$ is compact. Assume that $f \in C(\mathbb{R}^d \times E)$, then $y(x) := \int_E f(x,t)\,\mathrm{d}t$ is continuous with respect to variable $x$.*

*Proof.* Fix arbitrary $x \in \mathbb{R}^d$, pick an $r > 0$, we denote $B^r_x = \{y \mid \|y - x\| \le r\}$. Since $f$ is continuous on the compact set $B^r_x \times E$, we know that $|f|$ is bounded from above. We can then apply the dominated convergence theorem to show that $\lim_{z \to x} y(z) = \lim_{z \to x} \int_E f(z,t)\,\mathrm{d}t = \int_E f(x,t)\,\mathrm{d}t = y(x)$. Thus, $y$ is continuous on $\mathbb{R}^d$. $\qquad\square$

Recall that a Borel measure $\sigma$ on $\mathbb{R}^d$ is strictly positive if $\sigma(B) > 0$ for any open set $B \subset \mathbb{R}^d$.

**Lemma A.7.** *Suppose $\sigma \in \mathscr{P}(\mathbb{R}^d)$ is strictly positive, assume $g : \mathbb{R}^d \to [0, +\infty)$ is continuous function. If $\int_{\mathbb{R}^d} g(x)\,\mathrm{d}\sigma(x) = 0$, one has $g(x) = 0$ for any $x \in \mathbb{R}^d$.*

*Proof.* Suppose not true, we have a specific $x_0 \in \mathbb{R}^d$ such that $g(x_0) > 0$. Since $g$ is continuous, one can find $\delta > 0$, such that $|g(x) - g(x_0)| < \frac{1}{2}g(x_0)$ for arbitrary $x \in B^\delta_{x_0} := \{x \mid \|x - x_0\| < \delta_0\}$. Consider the indicator function $\chi_{B^\delta_{x_0}}(x) = \begin{cases} 1 & x \in B^\delta_{x_0}; \\ 0 & \text{otherwise,} \end{cases}$ one has $g(\cdot) \ge \frac{g(x_0)}{2}\chi_{B^\delta_{x_0}}(\cdot)$ on $\mathbb{R}^d$. This yields $\int_{\mathbb{R}^d} g(x)\,\mathrm{d}\sigma(x) \ge \int_{\mathbb{R}^d} \frac{g(x_0)}{2}\chi_{B^\delta_{x_0}}(x)\,\mathrm{d}\sigma(x) = \frac{g(x_0)\sigma(B^\delta_{x_0})}{2} > 0$, where the last inequality is due to $\sigma$ is strictly positive and the ball $B^\delta_{x_0}$ is an open set. This is a contradiction. $\quad\square$

## A.3    PROOF OF THEOREM 5.4

Without loss of generality, we we prove the theorem with $t_f = 1$. The proof can be generalized to arbitrary time horizon with no difficulty. We first state and prove two auxiliary lemmas.

**Lemma A.8.** *Let $f(t)$ be a bounded and Lipschitz function on $[0,1]$. Then*

$$\|f\|_{L^\infty} \le C \|f\|_{L^2}^{2/3}.$$

*This $C$ only depends on the bound and Lipschitz constant of $f$.*

*Proof.* Let $K = \|f\|_{L^\infty}$, and $L$ be the Lipschitz constant of $f$. We split into two cases.

*Case 1.* If $K \ge L$, then by the Lipschitz condition of $f$, we have

$$\int_0^1 f(t)^2\,\mathrm{d}t \ge \int_0^1 (K - Lt)^2\,\mathrm{d}t = K^2 - LK + \frac{1}{3}L^2 \ge \frac{1}{3}K^2.$$

*Case 2.* If $K < L$, since $f$ is $L$-Lipschitz, $f(t)$ is non-zero for an interval of length at least $K/L$, and

$$\int_0^1 f(t)^2\,\mathrm{d}t \ge \int_0^{K/L} (K - Lt)^2\,\mathrm{d}t = \frac{K^3}{3L}.$$

In both cases, we can conclude that

$$\|f\|_{L^\infty} \le C \|f\|_{L^2}^{2/3}.$$

$\square$

Next, we present a lemma that shows the stability of the MMD with respect to the mean and covariance of Gaussian distributions. The MMD with kernel $k(\mathbf{x}, \mathbf{y}) = -\|x - y\|$ is known as the energy distance, which is extensively studied in Székely & Rizzo (2013); Rizzo & Székely (2016).

**Lemma A.9** (Stability of MMD on Gaussian). *Let $\mu \sim N(\mathbf{b}_\mu, \Sigma_\mu)$ and $\nu \sim N(\mathbf{b}_\nu, \Sigma_\nu)$ be two Gaussian distributions in $\mathbb{R}^d$, with $\|\mathbf{b}_\mu\|, \|\mathbf{b}_\nu\|, \|\Sigma_\mu\|_2, \|\Sigma_\nu\|_2 \leq K$. Then, there exists a constant $C$ that only depends on $d$ and $K$ s.t.*

$$\|\mathbf{b}_\mu - \mathbf{b}_\nu\|^2 + \|\Sigma_\mu - \Sigma_\nu\|_2^2 \leq C \, \mathrm{MMD}(\mu, \nu)^2. \tag{19}$$

*Remark* A.10. The boundedness assumption is necessary. As a counterexample, let $\mu \sim N(0, \sigma^2)$ and $\nu \sim N(1, \sigma^2)$. Let $\sigma^2 \to \infty$, $|b_\mu - b_\nu| = |0 - 1| = 1$ remains unchanged, while $\mathrm{MMD}(\mu, \nu)$ converges to 0 (see (20)).

*Proof.* Throughout the proof, we will use the notations $c$ and $C$ to denote positive constants that only depends on $d$ and $K$. These constants may change from line to line. We start by recalling an important result. Let $\phi_\mu(\mathbf{t}), \phi_\nu(\mathbf{t}) : \mathbb{R}^d \to \mathbb{R}$ be the characteristic functions of $\mu$, $\nu$

$$\phi_j(\mathbf{t}) = \exp\left(i\mathbf{b}_j^\top \mathbf{t} - \frac{1}{2}\mathbf{t}^\top \Sigma_j \mathbf{t}\right) \quad (j = \mu, \nu).$$

Then (Székely & Rizzo, 2013, Proposition 1) establishes that

$$\mathrm{MMD}(\mu, \nu)^2 = \Gamma\left(\frac{d+1}{2}\right) \pi^{-\frac{d+1}{2}} \int_{\mathbb{R}^d} \frac{|\phi_\mu(\mathbf{t}) - \phi_\nu(\mathbf{t})|^2}{\|\mathbf{t}\|^{d+1}} \, d\mathbf{t}. \tag{20}$$

Next, we split into two steps and give bounds for the mean and covariance separately.

**Step 1.** We give bounds for the mean. For any given $\mathbf{t} \in \mathbb{R}^d$, the four complex numbers $\exp\left(i\mathbf{b}_j^\top \mathbf{t} - \frac{1}{2}\mathbf{t}\Sigma_k \mathbf{t}\right)$ $(j, k = \mu, \nu)$ forms a isosceles trapezoid in the complex domain. In an isosceles trapezoid, the length of each diagonal is greater than or equal to the arithmetic mean of the lengths of the two parallel sides. Therefore,

$$\begin{aligned}
|\phi_\mu(\mathbf{t}) - \phi_\nu(\mathbf{t})| &= \left|\exp\left(i\mathbf{b}_\mu^\top \mathbf{t} - \frac{1}{2}\mathbf{t}^\top \Sigma_\mu \mathbf{t}\right) - \exp\left(i\mathbf{b}_\nu^\top \mathbf{t} - \frac{1}{2}\mathbf{t}^\top \Sigma_\nu \mathbf{t}\right)\right| \\
&\geq \frac{1}{2}\left[\exp\left(-\frac{1}{2}\mathbf{t}^\top \Sigma_\mu \mathbf{t}\right) + \exp\left(-\frac{1}{2}\mathbf{t}^\top \Sigma_\nu \mathbf{t}\right)\right] \left|1 - \exp\left(i(\mathbf{b}_\mu - \mathbf{b}_\nu)^\top \mathbf{t}\right)\right| \\
&\geq \exp(-C\|\mathbf{t}\|^2)\left|1 - \cos((\mathbf{b}_\mu - \mathbf{b}_\nu)^\top \mathbf{t}) - i\sin((\mathbf{b}_\mu - \mathbf{b}_\nu)^\top \mathbf{t})\right|.
\end{aligned}$$

Substitute this estimation into (20), we obtain

$$\begin{aligned}
\mathrm{MMD}(\mu, \nu)^2 &\geq c \int_{\mathbb{R}^d} \frac{\exp(-C\|\mathbf{t}\|^2)}{\|\mathbf{t}\|^{d+1}} \left|1 - \cos((\mathbf{b}_\mu - \mathbf{b}_\nu)^\top \mathbf{t}) - i\sin((\mathbf{b}_\mu - \mathbf{b}_\nu)^\top \mathbf{t})\right|^2 d\mathbf{t} \\
&= 4c \int_{\mathbb{R}^d} \frac{\exp(-C\|\mathbf{t}\|^2)}{\|\mathbf{t}\|^{d+1}} \sin^2\left(\frac{1}{2}(\mathbf{b}_\mu - \mathbf{b}_\nu)^\top \mathbf{t}\right) d\mathbf{t}.
\end{aligned}$$

Since $\|\mathbf{b}_\mu - \mathbf{b}_\nu\| \leq 2K$ is bounded, we can find $r > 0$ that only depends on $K$ such that $\sin(x) \geq \frac{1}{2}x$ for all $x \in [0, \frac{1}{2}\|\mathbf{b}_\mu - \mathbf{b}_\nu\|r]$. Denote $B_r$ the ball in $\mathbb{R}^d$ centered at the origin with radius $r$. we have

$$\begin{aligned}
\mathrm{MMD}(\mu, \nu)^2 &\geq c \int_{B_r} \frac{\exp(-C\|\mathbf{t}\|^2)}{\|\mathbf{t}\|^{d+1}} \sin^2\left(\frac{1}{2}(\mathbf{b}_\mu - \mathbf{b}_\nu)^\top \mathbf{t}\right) d\mathbf{t} \\
&\geq c \int_{B_r} \frac{\exp(-Cr^2)}{\|\mathbf{t}\|^{d+1}} \left(\frac{1}{4}(\mathbf{b}_\mu - \mathbf{b}_\nu)^\top \mathbf{t}\right)^2 d\mathbf{t} \\
&\geq c \int_{B_r} \frac{1}{\|\mathbf{t}\|^{d+1}} \left|(\mathbf{b}_\mu - \mathbf{b}_\nu)^\top \mathbf{t}\right|^2 d\mathbf{t}.
\end{aligned}$$

If we further restrict the $3D$ angle of $\mathbf{t}$ in the set $\widetilde{B}_r = \{\mathbf{t} \in B_r : |(\mathbf{b}_\mu - \mathbf{b}_\nu)^\top \mathbf{t}| \geq \frac{1}{2}\|\mathbf{b}_\mu - \mathbf{b}_\nu\|\|\mathbf{t}\|\}$, i.e., the angle between $\mathbf{t}$ and $\mathbf{b}_\mu - \mathbf{b}_\nu$ is close to 0 or $\pi$. Then,

$$\begin{aligned}
\mathrm{MMD}(\mu, \nu)^2 &\geq c \int_{\widetilde{B}_r} \frac{1}{\|\mathbf{t}\|^{d+1}} \left|(\mathbf{b}_\mu - \mathbf{b}_\nu)^\top \mathbf{t}\right|^2 d\mathbf{t} \\
&\geq \frac{1}{4}c \int_{\widetilde{B}_r} \frac{1}{\|\mathbf{t}\|^{d-1}} \|\mathbf{b}_\mu - \mathbf{b}_\nu\|^2 d\mathbf{t} = c\|\mathbf{b}_\mu - \mathbf{b}_\nu\|^2.
\end{aligned}$$

**Step 2.** We give bounds for the covariance. This time, we use the fact that the length of each diagonal in a isosceles trapezoid is greater than or equal to the length of either of the non-parallel (equal) sides. This gives

$$
|\phi_\mu(\mathbf{t}) - \phi_\nu(\mathbf{t})| = \left| \exp\left( i\mathbf{b}_\mu^\top \mathbf{t} - \frac{1}{2}\mathbf{t}^\top \Sigma_\mu \mathbf{t} \right) - \exp\left( i\mathbf{b}_\nu^\top \mathbf{t} - \frac{1}{2}\mathbf{t}^\top \Sigma_\nu \mathbf{t} \right) \right|
$$

$$
\geq \left| \exp\left( -\frac{1}{2}\mathbf{t}^\top \Sigma_\mu \mathbf{t} \right) - \exp\left( -\frac{1}{2}\mathbf{t}^\top \Sigma_\nu \mathbf{t} \right) \right|
$$

$$
= \exp\left( -\frac{1}{2}\mathbf{t}^\top \Sigma_\mu \mathbf{t} \right) \left| 1 - \exp\left( -\frac{1}{2}\mathbf{t}^\top (\Sigma_\nu - \Sigma_\mu)\mathbf{t} \right) \right|
$$

$$
\geq \exp(-C \|\mathbf{t}\|^2) \left| -\frac{1}{2}\mathbf{t}^\top (\Sigma_\nu - \Sigma_\mu)\mathbf{t} \right|,
$$

where we used $|1 - e^x| \geq |x|$ in the last inequality. Next, we diagonalize the symmetric matrix $\Sigma_\nu - \Sigma_\mu$ as $Q(\Sigma_\nu - \Sigma_\mu)Q^\top = \Lambda = \mathrm{diag}(\lambda_j)_{j=1}^d$. Without loss of generality, we assume that $|\lambda_1| = \|\Sigma_\nu - \Sigma_\mu\|_2$. We denote $\mathbb{S}^{d-1} \subset \mathbb{R}^d$ the unit sphere in $\mathbb{R}^d$. Substituting the estimation above into (20), we get

$$
\mathrm{MMD}(\mu, \nu)^2
$$

$$
\geq c \int_{\mathbb{R}^d} \frac{\exp(-C \|\mathbf{t}\|^2)}{\|\mathbf{t}\|^{d+1}} \left( \mathbf{t}^\top (\Sigma_\nu - \Sigma_\mu)\mathbf{t} \right)^2 \mathrm{d}\mathbf{t}
$$

$$
\geq c \int_{B_1} \frac{1}{\|\mathbf{t}\|^{d+1}} \left( \mathbf{t}^\top (\Sigma_\nu - \Sigma_\mu)\mathbf{t} \right)^2 \mathrm{d}\mathbf{t}
$$

$$
= c \int_0^1 \int_{\mathbb{S}^{d-1}} R^{-d-1} R^4 \left( \mathbf{s}^\top (\Sigma_\nu - \Sigma_\mu)\mathbf{s} \right)^2 R^{d-1} \mathrm{d}\mathbf{s}\, \mathrm{d}R
$$

$$
= c \int_{\mathbb{S}^{d-1}} \left( \mathbf{s}^\top (\Sigma_\nu - \Sigma_\mu)\mathbf{s} \right)^2 \mathrm{d}\mathbf{s} = c \int_{\mathbb{S}^{d-1}} \left( \mathbf{s}^\top \Lambda \mathbf{s} \right)^2 \mathrm{d}\mathbf{s}.
$$

We further pick a subset $\widetilde{\mathbb{S}}^{d-1} = \{ \mathbf{s} \in \mathbb{S}^{d-1} : |\mathbf{s}_1|^2 \geq \frac{2}{3} \}$ (i.e., points on the unit sphere with the first coordinate $\geq \frac{2}{3}$). For all $\mathbf{s} \in \widetilde{\mathbb{S}}^{d-1}$

$$
\left| \mathbf{s}^\top \Lambda \mathbf{s} \right| = \left| \sum_{j=1}^d \lambda_j \mathbf{s}_j^2 \right| \geq |\lambda_1| \mathbf{s}_1^2 - \sum_{j=2}^d |\lambda_j| \, \mathbf{s}_j^2 \geq \frac{1}{3} |\lambda_1|.
$$

Therefore,

$$
\mathrm{MMD}(\mu, \nu)^2 \geq c \int_{\widetilde{\mathbb{S}}^{d-1}} \left( \mathbf{s}^\top \Lambda \mathbf{s} \right)^2 \mathrm{d}\mathbf{s} \geq \frac{1}{9} c \int_{\widetilde{\mathbb{S}}^{d-1}} \lambda_1^2 \, \mathrm{d}\mathbf{s} = c \|\Sigma_\mu - \Sigma_\nu\|_2^2.
$$

Finally, combining **Step 1** and **Step 2**, we reach the conclusion that

$$
\|\mathbf{b}_\mu - \mathbf{b}_\nu\|^2 + \|\Sigma_\mu - \Sigma_\nu\|_2^2 \leq C \, \mathrm{MMD}(\mu, \nu)^2.
$$

$\square$

Before proving Theorem 5.4, we clarify the result we need to show. The HJ equation is

$$
\partial_t u(\mathbf{x}, t) + \frac{1}{2} |\nabla_x u(\mathbf{x}, t)|^2 = 0.
$$

The optimal push forward map is $T^*(\mathbf{x}) = \mathbf{x} + \nabla_x u(\mathbf{x}, 0)$, which implies $\nabla_x u(\mathbf{x}, 0) = T^*(\mathbf{x}) - \mathbf{x}$. The optimal trajectory for OT has constant velocity, given by

$$
\mathbf{x}_t = \mathbf{x} + t\nabla_x u(\mathbf{x}, 0) = \mathbf{x} - t(\mathbf{x} - T^*(\mathbf{x})).
$$

Therefore, the optimal push forward map is

$$
f(\mathbf{x}, t) = (1-t)\mathbf{x} + tT^*(\mathbf{x}) = ((1-t)I + At)\,\mathbf{x} + (\mathbf{b}_\nu - A\mathbf{b}_\mu)t.
$$

Taking derivative in $t$, we get the optimal velocity in Lagrange coordinate

$$\partial_t f(\mathbf{x}, t) = -\mathbf{x} + T^*(\mathbf{x}) = \frac{1}{t}(f(\mathbf{x}, t) - \mathbf{x})$$

$$= \frac{1}{t}\left[ f(\mathbf{x}, t) - ((1-t)I + At)^{-1}\left(f(\mathbf{x}, t) - (\mathbf{b}_\nu - A\mathbf{b}_\mu)t\right)\right]$$

$$= (I + t(A - I))^{-1}\left((A - I)f(\mathbf{x}, t) + \mathbf{b}_\nu - A\mathbf{b}_\mu\right).$$

Therefore, the optimal velocity field in Eulerian coordinate is

$$\nabla_x u(\mathbf{x}, t) = v(\mathbf{x}, t) = (I + t(A - I))^{-1}\left((A - I)\mathbf{x} + \mathbf{b}_\nu - A\mathbf{b}_\mu\right). \tag{21}$$

The most important challenge for convergence analysis is the term $u_\theta(\mathbf{x} - t\nabla_x u_\theta(\mathbf{x}, t), 0)$ in the HJ loss, which contains the composition of the $u$. In order to address this issue, we consider the quadratic parametrization

$$u_\theta(\mathbf{x}, t) = -\left(\frac{1}{2}\mathbf{x}^\top \theta_2(t)\mathbf{x} + \theta_1(t)^\top \mathbf{x} + \theta_0(t)\right) \tag{22}$$

where $\theta = [\theta_2(\cdot), \theta_1(\cdot), \theta_0(\cdot)] : [0, t_f] \to \mathbb{R}^{d\times d}_{\text{sym}} \times \mathbb{R}^d \times \mathbb{R}$ is bounded and Lipschitz. According to (21), the optimal $\theta_2^*(t)$ and $\theta_1^*(t)$ are uniquely determined, and the optimal $\theta_0^*(t)$ is uniquely determine up to an additive constant.

$$\begin{aligned} \theta_2^*(t) &= ((1-t)I + At)^{-1}(I - A) \\ \theta_1^*(t) &= ((1-t)I + At)^{-1}(A\mathbf{b}_\mu - \mathbf{b}_\nu) \\ \theta_0^*(t) &= \theta_0^*(0) + \frac{t}{2}(\mathbf{b}_\nu - A\mathbf{b}_\mu)^\top ((1-t)I + At)^{-1}(\mathbf{b}_\nu - A\mathbf{b}_\mu) \end{aligned} \tag{23}$$

Now we are ready to prove Theorem 5.4. We denote $D = [-1, 1]^d$. Throughout the proof, we will set $\varrho = 2^{-d}\mathbb{1}_D$ and the time domain is $[0, t_f] = [0, 1]$, which coincide with our numerical implementation. The proof can be extended to general domain without essential difficulty. Throughout the proof, when we say a function is bounded and Lipschitz continuous, we mean the bound and Lipschitz constant only depends on $d$, $\lambda_A$, and $K$. We will use $C$ to denote an absolute constant that only depends on $d$, $\lambda_A$, and $K$. The value of $C$ may change from line to line.

*Proof for theorem 5.4.* We only need to show (16) when $\mathcal{L}_{\text{HJ}}$ and $\mathcal{L}_{\text{MMD}}$ are sufficiently small. The proof consists of four steps.

**Step 1.** We analyze the MMD loss in this step. Recall the MMD loss is

$$\mathcal{L}_{\text{MMD}} = \int_\Omega k(\mathbf{x}, \mathbf{y})\,\mathrm{d}((\text{Id} + \nabla_x u_\theta(\cdot, 0))_{\#}\mu - \nu)(\mathbf{x})\,\mathrm{d}((\text{Id} + \nabla_x u_\theta(\cdot, 0))_{\#}\mu - \nu)(\mathbf{y}).$$

Under with the parametrization (22), the MMD loss is between

$$(\text{Id} + \nabla_x u(\cdot, 0))_{\#}\mu = N\left((I - \theta_2(0))\mathbf{b}_\mu - \theta_1(0),\ (I - \theta_2(0))\Sigma_\mu(I - \theta_2(0))\right) =: N(\mathbf{b}'_\mu, \Sigma'_\mu)$$

and $\nu = N(\mathbf{b}_\nu, \Sigma_\nu)$. By Lemma A.9, we have

$$\left\|\mathbf{b}'_\mu - \mathbf{b}_\nu\right\|^2 + \left\|\Sigma'_\mu - \Sigma_\nu\right\|_2^2 \le C\mathcal{L}_{\text{MMD}} \tag{24}$$

Multiplying $\Sigma_\mu^{\frac{1}{2}}$ on both sides for the covariance, we get

$$\left\|\left(\Sigma_\mu^{\frac{1}{2}}(I - \theta_2(0))\Sigma_\mu^{\frac{1}{2}}\right)^2 - \Sigma_\mu^{\frac{1}{2}}\Sigma_\nu\Sigma_\mu^{\frac{1}{2}}\right\|_2 \le C\mathcal{L}_{\text{MMD}}^{\frac{1}{2}}.$$

By the $\frac{1}{2}$-Hölder continuity of matrix square root in operator norm (Bhatia, 2013, Theorem X.1.1) ($\left\|A^{\frac{1}{2}} - B^{\frac{1}{2}}\right\|_2 \le \|A - B\|_2^{\frac{1}{2}}$ for any symmetric positive definite matrix $A, B$), we have

$$\left\|\left[\left(\Sigma_\mu^{\frac{1}{2}}(I - \theta_2(0))\Sigma_\mu^{\frac{1}{2}}\right)^2\right]^{\frac{1}{2}} - \left(\Sigma_\mu^{\frac{1}{2}}\Sigma_\nu\Sigma_\mu^{\frac{1}{2}}\right)^{\frac{1}{2}}\right\|_2 \le C\mathcal{L}_{\text{MMD}}^{\frac{1}{4}}. \tag{25}$$

Next, we diagonalize $\theta_2(t)$. Since $\theta_2(t) \in \mathbb{R}^{d \times d}$ is symmetric, we can find unitary matrix $Q(t)$ and diagonal matrix $\Lambda(t) = \mathrm{diag}(\{\lambda_i(t)\}_{i=1}^d)$ s.t.

$$\theta_2(t) = Q(t)\Lambda(t)Q(t)^\top,$$

and $\lambda_1(0) \geq \ldots \geq \lambda_d(0)$. The column vectors of $Q(t)$ are the orthonormal eigenvectors of $\theta_2(t)$. Since $\theta_2(t)$ is bounded and Lipschitz continuous in $t$, its eigenvalues and eigenvectors are also bounded and Lipschitz continuous in $t$ by Weyl's inequality. $Q(t)$ is uniquely defined up to a sign shift for each column. Then $I - \theta_2(0) = Q(0)(I - \Lambda(0))Q(0)^\top$.

Next, we define a notation of "absolute value" for symmetric matrices. Given a symmetric matrix, we diagonalize it through unitary transform, take absolute value of the diagonal element, and then apply the inverse unitary transform back. As a result, $|I - \Lambda(t)| = \mathrm{diag}(\{|1 - \lambda_i(t)|\}_{i=1}^d)$ and $|I - \theta_2(0)| = Q(0)\,|I - \Lambda(t)|\,Q(0)^\top$, then we have

$$\left[\left(\Sigma_\mu^{\frac{1}{2}}(I - \theta_2(0))\Sigma_\mu^{\frac{1}{2}}\right)^2\right]^{\frac{1}{2}} = \Sigma_\mu^{\frac{1}{2}}\,|I - \theta_2(0)|\,\Sigma_\mu^{\frac{1}{2}}.$$

Here, we remark that while $Q(t)$ is not uniquely defined, the "absolute value" $|I - \theta_2(0)|$ is uniquely defined given $\theta_2(0)$. Plugging this expression into (25), we get

$$\left\|\Sigma_\mu^{\frac{1}{2}}\,|I - \theta_2(0)|\,\Sigma_\mu^{\frac{1}{2}} - \left(\Sigma_\mu^{\frac{1}{2}}\Sigma_\nu\Sigma_\mu^{\frac{1}{2}}\right)^{\frac{1}{2}}\right\|_2 \leq C\mathcal{L}_{\mathrm{MMD}}^{\frac{1}{4}}.$$

Multiplying $\Sigma_\mu^{-\frac{1}{2}}$ on both sides, we get

$$\left\||I - \theta_2(0)| - A\right\|_2 \leq C\mathcal{L}_{\mathrm{MMD}}^{\frac{1}{4}}, \tag{26}$$

which implies

$$\left\||I - \Lambda(0)| - Q(0)AQ(0)^\top\right\|_2 \leq C\mathcal{L}_{\mathrm{MMD}}^{\frac{1}{4}}. \tag{27}$$

Therefore, the off diagonal elements of $(Q(0)AQ(0)^\top)_{ij}$ $(i \neq j)$ and diagonal elements $(Q(0)AQ(0)^\top)_{ii}$ satisfy

$$\left|(Q(0)AQ(0)^\top)_{ij}\right|,\ \left|(Q(0)AQ(0)^\top)_{ii} - |1 - \lambda_i(0)|\right| \leq C_1\mathcal{L}_{\mathrm{MMD}}^{\frac{1}{4}}.$$

Here, we add a subscript 1 in the constant $C_1$ in order to keep track of this constant. Later, whenever we use $C_1$, it means this fixed constant that does not change from line to line.

We remark that there are $2^d$ choices of $\theta_2(0)$ such that $(I - \theta_2(0))\Sigma_\mu(I - \theta_2(0)) = \Sigma_\nu$ through letting $1 - \lambda_i(0) = \pm\lambda_i^A$ $(i = 1, \ldots, d)$, where $\lambda_i^A$ is the $i$-th eigenvalue of $A$. All these choices gives a push forward map that transport $\mu$ to $\nu$ (if we set $\theta_1(0) = (I - \theta_2(0))\mathbf{b}_\mu - \mathbf{b}_\nu$). However, only $\theta_2(0) = I - A$ gives the OT map. The MMD loss $\mathcal{L}_{\mathrm{MMD}}$ cannot distinguish these choices, so the HJ loss $\mathcal{L}_{\mathrm{HJ}}$ is necessary.

**Step 2.** We analyze the implicit HJ loss in this step. Under the parametrization (22), the HJ loss is

$$\mathcal{L}_{\mathrm{HJ}} = \int_0^1 \int_\Omega \Big[\frac{1}{2}\mathbf{x}^\top\theta_2(t)\mathbf{x} + \mathbf{x}^\top\theta_1(t) + \theta_0(t) + \frac{t}{2}(\theta_2(t)\mathbf{x} + \theta_1(t))^\top(\theta_2(t)\mathbf{x} + \theta_1(t))$$
$$- \frac{1}{2}(\mathbf{x} + t(\theta_2(t)\mathbf{x} + \theta_1(t)))^\top\theta_2(0)(\mathbf{x} + t(\theta_2(t)\mathbf{x} + \theta_1(t)))$$
$$- \theta_1(0)^\top(\mathbf{x} + t(\theta_2(t)\mathbf{x} + \theta_1(t))) - \theta_0(0)\Big]^2 \varrho(\mathbf{x})\,\mathrm{d}\mathbf{x}\,\mathrm{d}t.$$

Reorganizing the terms, we have

$$\mathcal{L}_{\mathrm{HJ}} = \int_0^1 \int_\Omega \Big[\frac{1}{2}\mathbf{x}^\top\left(\theta_2(t) + t\theta_2(t)^2 - (I + t\theta_2(t))\theta_2(0)(I + t\theta_2(t))\right)\mathbf{x}$$
$$+ \mathbf{x}^\top(I + t\theta_2(t))(\theta_1(t) - t\theta_2(0)\theta_1(t) - \theta_1(0))$$
$$+ \left(\theta_0(t) + \frac{t}{2}\theta_1(t)^\top\theta_1(t) - \frac{t^2}{2}\theta_1(t)^\top\theta_2(0)\theta_1(t) - t\theta_1(t)^\top\theta_1(0) - \theta_0(0)\right)\Big]^2 \varrho(\mathbf{x})\,\mathrm{d}\mathbf{x}\,\mathrm{d}t.$$
$$=: \int_0^1 \int_\Omega \left[\frac{1}{2}\mathbf{x}^\top\Gamma_2(t)\mathbf{x} + \mathbf{x}^\top\Gamma_1(t) + \Gamma_0(t)\right]^2 \varrho(\mathbf{x})\,\mathrm{d}\mathbf{x}\,\mathrm{d}t.$$

We observe that $\Gamma_2(t)$ is symmetric. The integration in $\mathbf{x}$ for the loss can be computed directly. The zero-th to third order integration in $\mathbf{x}$ can be easily obtained by symmetry (recall $\varrho(x) = 2^{-d} \mathbb{1}_D(x)$ and $D = [-1,1]^d$)

$$\int_D (1, x_i, x_i x_j, x_i x_j x_k) \, d\mathbf{x} = \left( 2^d, 0, \frac{2^d \delta_{ij}}{3}, 0 \right).$$

In order to compute the fourth order integration in $\mathbf{x}$, we temporally denote $\Gamma_2(t)$ by $\Gamma$ for notational simplicity and compute the integration $\int_D (\mathbf{x}^\top \Gamma \mathbf{x})^2 \, d\mathbf{x}$. Expanding everything, the integration is

$$\int_D (\mathbf{x}^\top \Gamma \mathbf{x})^2 \, d\mathbf{x} = \sum_{i,j,k,l=1}^d \Gamma_{ij} \Gamma_{kl} \int_D x_i x_j x_k x_l \, d\mathbf{x}.$$

All the non-zero terms in the form $\Gamma_{ij} \Gamma_{kl} \int_D x_i x_j x_k x_l \, d\mathbf{x}$ can be categorized into 4 cases.

1. $i = j = k = l$. The integration is $\Gamma_{ii}^2 \int_D x_i^4 \, d\mathbf{x} = \frac{2^d}{5} \Gamma_{ii}^2$.

2. $i = j \neq k = l$. The integration is $\Gamma_{ii} \Gamma_{kk} \int_D x_i^2 x_k^2 \, d\mathbf{x} = \frac{2^d}{9} \Gamma_{ii} \Gamma_{kk}$.

3. $i = k \neq j = l$. The integration is $\Gamma_{ij}^2 \int_D x_i^2 x_j^2 \, d\mathbf{x} = \frac{2^d}{9} \Gamma_{ij}^2$.

4. $i = l \neq j = k$. The integration is $\Gamma_{ij} \Gamma_{ji} \int_D x_i^2 x_j^2 \, d\mathbf{x} = \frac{2^d}{9} \Gamma_{ij} \Gamma_{ji} = \frac{2^d}{9} \Gamma_{ij}^2$.

Summing them together, we have

$$\int_D (\mathbf{x}^\top \Gamma \mathbf{x})^2 \, d\mathbf{x} = 2^d \left[ \sum_{i=1}^d \frac{1}{5} \Gamma_{ii}^2 + \sum_{i \neq j} \left( \frac{1}{9} \Gamma_{ii} \Gamma_{jj} + \frac{2}{9} \Gamma_{ij}^2 \right) \right].$$

Therefore, after integration in $\mathbf{x}$, the implicit HJ loss becomes

$$\begin{aligned}
\mathcal{L}_{\mathrm{HJ}} &= \int_0^1 \left[ \frac{1}{4} \sum_{i=1}^d \frac{1}{5} \Gamma_2(t)_{ii}^2 + \frac{1}{4} \sum_{i \neq j} \left( \frac{1}{9} \Gamma_2(t)_{ii} \Gamma_2(t)_{jj} + \frac{2}{9} \Gamma_2(t)_{ij}^2 \right) \right. \\
&\quad \left. + \frac{1}{3} \mathrm{Tr}(\Gamma_2(t)) \Gamma_0(t) + \frac{1}{3} \|\Gamma_1(t)\|^2 + \Gamma_0(t)^2 \right] dt \\
&= \int_0^1 \left[ \frac{1}{45} \sum_{i=1}^d \Gamma_2(t)_{ii}^2 + \frac{1}{18} \sum_{i \neq j} \Gamma_2(t)_{ji}^2 + \frac{1}{3} \|\Gamma_1(t)\|^2 + \left( \frac{1}{6} \mathrm{Tr}(\Gamma_2(t)) + \Gamma_0(t) \right)^2 \right] dt \\
&\geq \int_0^1 \left[ \frac{1}{45} \|\Gamma_2(t)\|_F^2 + \frac{1}{3} \|\Gamma_1(t)\|^2 + \left( \frac{1}{6} \mathrm{Tr}(\Gamma_2(t)) + \Gamma_0(t) \right)^2 \right] dt
\end{aligned}$$

Therefore,

$$\int_0^1 \left( \|\Gamma_2(t)\|_F^2 + \|\Gamma_1(t)\|^2 + \Gamma_0(t)^2 \right) dt \leq C \mathcal{L}_{\mathrm{HJ}}.$$

Therefore, by Lemma A.8, we have

$$\max_t \|\Gamma_2(t)\|_F + \max_t \|\Gamma_1(t)\| + \max_t |\Gamma_0(t)| \leq C_2 \mathcal{L}_{\mathrm{HJ}}^{\frac{1}{3}}. \tag{28}$$

Here, $C_2$ also does not change from line to line.

**Step 3.** In this step, we show that $\theta_2(0)$ must be close to $\theta_2^*(0) = I - A$, provided that $\|\Gamma_2(t)\|_F$ is sufficiently small. Since $A$ has minimum eigenvalue $\lambda_A > 0$, $(Q(0)AQ(0)^\top)_{ii} \geq \lambda_A \geq C_1 \mathcal{L}_{\mathrm{MMD}}^{\frac{1}{4}}$, where the last inequality is because $\mathcal{L}_{\mathrm{MMD}}$ is sufficiently small. We recall that in **Step 1**, we showed for any $i = 1, \ldots, d$

$$\left| (Q(0)AQ(0)^\top)_{ii} - |1 - \lambda_i(0)| \right| \leq C_1 \mathcal{L}_{\mathrm{MMD}}^{\frac{1}{4}}.$$

We want to show that,

$$\left| (Q(0)AQ(0)^\top)_{ii} - (1 - \lambda_i(0)) \right| \leq C_1 \mathcal{L}_{\mathrm{MMD}}^{\frac{1}{4}} \tag{29}$$

for all $i$. I.e., we want to show $1 - \lambda_i(0) \geq 0$ and

$$\lambda_i(0) \in \left[1 - (Q(0)AQ(0)^\top)_{ii} - C_1 \mathcal{L}_{\text{MMD}}^{\frac{1}{4}}, 1 - (Q(0)AQ(0)^\top)_{ii} + C_1 \mathcal{L}_{\text{MMD}}^{\frac{1}{4}}\right]$$

for all $i$. In order to show this, we assume to the contrary that $\lambda_1(0) > 1$ (recall $\lambda_1(0) \geq \ldots \geq \lambda_d(0)$) and

$$\lambda_1(0) \in \left[1 + (Q(0)AQ(0)^\top)_{11} - C_1 \mathcal{L}_{\text{MMD}}^{\frac{1}{4}}, 1 + (Q(0)AQ(0)^\top)_{11} + C_1 \mathcal{L}_{\text{MMD}}^{\frac{1}{4}}\right] \tag{30}$$

We will derive a contradiction. We denote $\widetilde{u}_2(t) := Q(t)^\top \theta_2(0) Q(t)$. Since unitary transform does not change Frobenius norm,

$$\begin{aligned} Q(t)^\top \Gamma_2(t) Q(t) &= \Lambda(t) + t\Lambda(t)^2 - (I + t\Lambda(t))(Q(t)^\top \theta_2(0) Q(t))(I + t\Lambda(t)) \\ &= \Lambda(t) + t\Lambda(t)^2 - (I + t\Lambda(t))\widetilde{u}_2(t)(I + t\Lambda(t)) \end{aligned} \tag{31}$$

shares the same estimation as (28). Let $\{\tau_i(t)\}_{i=1}^d$ be the diagonal element of $\widetilde{u}_2(t) = Q(t)^\top \theta_2(0) Q(t)$. Since $Q(t)$ is bounded and Lipschitz continuous, $\widetilde{u}_2(t)$ and $\tau_i(t)$ are also bounded and Lipschitz continuous. We will also use $K$ to denote their bound and Lipschitz constant. Since $\lambda_1(0) > 1$ is an eigenvalue of $\theta_2(0)$ and hence also an eigenvalue of $\widetilde{u}_2(t)$, we have

$$\max_i \tau_i(t) > 1 \quad \forall t \in [0, 1].$$

Next, we focus on the diagonal elements of $Q(t)^\top \Gamma_2(t) Q(t)$. Since

$$\left\| Q(t)^\top \Gamma_2(t) Q(t) \right\|_F \leq C_2 \mathcal{L}_{\text{HJ}}^{\frac{1}{3}}, \tag{32}$$

its $i$-th diagonal element (recall (31))

$$\begin{aligned} &\lambda_i(t) + t\lambda_i(t)^2 - (1 + t\lambda_i(t))^2 \tau_i(t) \\ &= (1 + t\lambda_i(t))\left[\lambda_i(t) - (1 + t\lambda_i(t))\tau_i(t)\right] \\ &= (1 + t\lambda_i(t))\left[(1 - t\tau_i(t))\lambda_i(t) - \tau_i(t)\right] \end{aligned}$$

also satisfies

$$\left|(1 + t\lambda_i(t))\left[(1 - t\tau_i(t))\lambda_i(t) - \tau_i(t)\right]\right| \leq C_2 \mathcal{L}_{\text{HJ}}^{\frac{1}{3}} \tag{33}$$

for all $t \in [0, 1]$, where recall that $\lambda_i(t)$ is the $i$-th diagonal element for $\Lambda(t) = Q(t)^\top \theta_2(t) Q(t)$, and $\tau_i(t)$ is the $i$-th diagonal element for $\widetilde{u}_2(t) = Q(t)^\top \theta_2(0) Q(t)$.

The rest of the proof for deriving a contradiction to (30) is technical, so we explain the main idea first. In order that $\left|(1 + t\lambda_i(t))\left[(1 - t\tau_i(t))\lambda_i(t) - \tau_i(t)\right]\right|$ is small for all $t \in [0, 1]$, either of the following must hold

1. $1 + t\lambda_i(t) \approx 0$, which implies $\lambda_i(t) \approx -\frac{1}{t}$

2. $(1 - t\tau_i(t))\lambda_i(t) - \tau_i(t) \approx 0$, which implies $\lambda_i(t) \approx \frac{\tau_i(t)}{1 - t\tau_i(t)} = \frac{1}{t}\frac{1}{1 - t\tau_i(t)} - \frac{1}{t}$. (At $t = 0$ the function is $\tau_i(t)$.)

When $t \to 0^+$, $-\frac{1}{t}$ blows up and we cannot have $1 + t\lambda_i(t) \approx 0$.

Since $\max_i \tau_i(t) > 1$, we know from intermediate value theorem that there exists at least one index $i$ and $t_i \in (0, 1)$ s.t. $1 - t_i \tau_i(t_i) = 0$. This implies that the function $\frac{\tau_i(t)}{1 - t\tau_i(t)}$ blows up as $t \to t_i$. As a result, in order that $\left|(1 + t\lambda_i(t))\left[(1 - t\tau_i(t))\lambda_i(t) - \tau_i(t)\right]\right|$ is small for all $t \in [0, 1]$, there has to be some "shift" between two functions: $\lambda_i(t)$ is sometimes close to $-\frac{1}{t}$ and sometimes close to $\frac{\tau_i(t)}{1 - t\tau_i(t)} = \frac{1}{t}\frac{1}{1 - t\tau_i(t)} - \frac{1}{t}$. However, note that the difference between the two functions $-\frac{1}{t}$ and $\frac{1}{t}\frac{1}{1 - t\tau_i(t)} - \frac{1}{t}$ has a positive lower bound

$$\left|\frac{1}{t}\frac{1}{1 - t\tau_i(t)}\right| \geq \left|\frac{1}{1 - t\tau_i(t)}\right| \geq \left|\frac{1}{t\tau_i(t)}\right| \geq \left|\frac{1}{\tau_i(t)}\right| \geq \frac{1}{\|A + I\|_2 + C_1 \mathcal{L}_{\text{MMD}}^{\frac{1}{4}}}. \tag{34}$$

Therefore, for some $t$ in the middle, both $\left|\lambda_i(t) + \frac{1}{t}\right|$ and $\left|\lambda_i(t) - \frac{\tau_i(t)}{1 - t\tau_i(t)}\right|$ are larger than $\frac{1}{2(\|A+I\| + C_1 \mathcal{L}_{\text{MMD}}^{\frac{1}{4}})}$ This gives a contradiction to (33).

Next, we give a rigorous proof for this contradiction. Let

$$t^* = \inf \left\{ t \in [0,1] : 1 - t\tau_i(t) = 0 \text{ for some } i \right\}.$$

Note that the set above is non-empty because $\max_i \tau_i(t) > 1$ for all $t \in [0,1]$. If we do not have the assumption $\lambda_1(0) > 1$, then $\tau_i(t) < 1$ may not be well-defined. By definition of $t^*$, we can find an index $j$ such that $1 - t^* \tau_j(t^*) = 0$. Therefore, $\tau_j(t^*) \geq 1$. Let $t_0 = \frac{1}{3K}$, then for $t \in [0, t_0]$, we have

$$|1 + t\lambda_j(t)| \geq 1 - t_0 K = \frac{2}{3}. \tag{35}$$

Let $t_1 = t^* - \Delta t$, where $\Delta t = \frac{1}{2K(1+2K)}$ then for all $t \in [t_1, t^*]$, we have

$$
\begin{aligned}
|(1 - t\tau_j(t))\lambda_j(t) - \tau_j(t)| &\geq |\tau_j(t)| - |1 - t\tau_j(t)| \, |\lambda_j(t)| \\
&\geq |\tau_j(t^*)| - K|t - t^*| - |1 - t\tau_j(t)|K \\
&\geq 1 - K\Delta t - K \left( |1 + t^* \tau_j(t^*)| + |t^* \tau_j(t^*) - t\tau_j(t)| \right) \\
&\geq 1 - K\Delta t - K \left( 0 + 2K|t - t^*| \right) \geq 1 - \Delta t K(2K + 1) = \frac{1}{2}.
\end{aligned}
\tag{36}
$$

If $t_0 \geq t_1$, we pick $t \in [t_1, t_0]$ and then multiply (35) and (36), we reach a contradiction with (33). If $t_0 < t_1$, then we consider the behavior of

$$(1 + t\lambda_j(t)) \left[ (1 - t\tau_j(t))\lambda_j(t) - \tau_j(t) \right].$$

When $t \in [0, t_0]$, $\lambda_j(t)$ is close to $\frac{\tau_j(t)}{1 - t\tau_j(t)}$ because (35) and (33) implies

$$|(1 - t\tau_j(t))\lambda_j(t) - \tau_j(t)| \leq \frac{3}{2} C_2 \mathcal{L}_{\text{HJ}}^{\frac{1}{3}},$$

which gives

$$\left| \lambda_j(t) - \frac{\tau_j(t)}{1 - t\tau_j(t)} \right| \leq \frac{3 C_2 \mathcal{L}_{\text{HJ}}^{\frac{1}{3}}}{2(1 - t|\tau_j(t)|)} \leq \frac{9 C_2 \mathcal{L}_{\text{HJ}}^{\frac{1}{3}}}{4}. \tag{37}$$

When $t \in [t_1, t^*]$, $\lambda_j(t)$ is close to $-\frac{1}{t}$ because (36) and (33) implies

$$|1 + t\lambda_j(t)| \leq 2 C_2 \mathcal{L}_{\text{HJ}}^{\frac{1}{3}}.$$

This implies

$$\left| \lambda_j(t) + \frac{1}{t} \right| \leq \frac{2 C_2 \mathcal{L}_{\text{HJ}}^{\frac{1}{3}}}{t} \leq \frac{2 C_2 \mathcal{L}_{\text{HJ}}^{\frac{1}{3}}}{t_1} \leq 6 K C_2 \mathcal{L}_{\text{HJ}}^{\frac{1}{3}}, \tag{38}$$

where the last inequality is because $t_1 > t_0 = \frac{1}{3K}$. Therefore, as explained before, there has to be a shift between the two approximations (37) and (38) in the middle when $t \in [t_0, t_1]$ because $\lambda_j(t)$ is Lipschitz continuous. However, the difference between the two functions has a positive lower bound (34)

$$\left| -\frac{1}{t} - \frac{\tau_j(t)}{1 - t\tau_j(t)} \right| = \left| \frac{1}{t(1 - t\tau_j(t))} \right| \geq \frac{1}{\|A + I\|_2 + C_1 \mathcal{L}_{\text{MMD}}^{\frac{1}{4}}}.$$

Therefore, there exists $t_2 \in [t_0, t_1]$ s.t.

$$\left| \lambda_j(t_2) - \frac{\tau_j(t_2)}{1 - t_2 \tau_j(t_2)} \right| \geq \frac{1}{2 \left( \|A + I\|_2 + C_1 \mathcal{L}_{\text{MMD}}^{\frac{1}{4}} \right)} \tag{39}$$

and

$$\left| \lambda_j(t_2) + \frac{1}{t_2} \right| \geq \frac{1}{2 \left( \|A + I\|_2 + C_1 \mathcal{L}_{\text{MMD}}^{\frac{1}{4}} \right)}. \tag{40}$$

Finally, we split into two cases.

*Case 1.* If $|1 - t_2\tau_j(t_2)| \le \frac{1}{3K}$, then

$$\tau_j(t_2) \ge \frac{1}{t_2}\left(1 - \frac{1}{3K}\right) \ge 1 - \frac{1}{3K}$$

and

$$
\begin{aligned}
&|(1 - t_2\tau_j(t_2))\lambda_j(t_2) - \tau_j(t_2)| \\
&\ge \tau_j(t_2) - |(1 - t_2\tau_j(t_2))|\,|\lambda_j(t_2)| \\
&\ge 1 - \frac{1}{3K} - \frac{1}{3K}K \ge \frac{1}{3}.
\end{aligned}
$$

This implies

$$
\begin{aligned}
&|(1 + t_2\lambda_j(t_2))\left[(1 - t_2\tau_j(t_2))\lambda_j(t_2) - \tau_j(t_2)\right]| \\
&\ge \frac{1}{3}|1 + t_2\lambda_j(t_2)| = \frac{1}{3}|t_2|\left|\frac{1}{t_2} + \lambda_j(t_2)\right| \\
&\ge \frac{1}{3}\frac{1}{3K}\frac{1}{2\left(\|A + I\|_2 + C_1\mathcal{L}_{\mathrm{MMD}}^{\frac{1}{4}}\right)} = \mathcal{O}(1),
\end{aligned}
$$

which contradicts to (33).

*Case 2.* If $|1 - t_2\tau_j(t_2)| > \frac{1}{3K}$, then

$$
\begin{aligned}
&|(1 + t_2\lambda_j(t_2))\left[(1 - t_2\tau_j(t_2))\lambda_j(t_2) - \tau_j(t_2)\right]| \\
&= |t_2|\left|\frac{1}{t_2} + \lambda_j(t_2)\right||1 - t_2\tau_j(t_2)|\left|\lambda_j(t_2) - \frac{\tau_j(t_2)}{1 - t_2\tau_j(t_2)}\right| \\
&\ge \frac{1}{3K}\frac{1}{2\left(\|A + I\|_2 + C_1\mathcal{L}_{\mathrm{MMD}}^{\frac{1}{4}}\right)}\frac{1}{3K}\frac{1}{2\left(\|A + I\|_2 + C_1\mathcal{L}_{\mathrm{MMD}}^{\frac{1}{4}}\right)} = \mathcal{O}(1),
\end{aligned}
$$

which also contradicts to (33).

Combining *Case 1* and *Case 2*, we conclude that the assumption $\lambda_1(0) > 1$ cannot hold. Therefore, (29) hold. This further implies $|I - \Lambda(0)| = I - \Lambda(0)$. Plugging back into (27) and (26), we get

$$\left\|I - \Lambda(0) - Q(0)AQ(0)^\top\right\|_2 \le C\mathcal{L}_{\mathrm{MMD}}^{\frac{1}{4}}$$

and

$$\|\theta_2(0) - (I - A)\|_2 \le C\mathcal{L}_{\mathrm{MMD}}^{\frac{1}{4}}.$$

Therefore, we obtain

$$\|\theta_2(0) - (I - A)\|_F \le C\mathcal{L}_{\mathrm{MMD}}^{\frac{1}{4}}. \tag{41}$$

**Step 4.** We show that $\theta_2(t)$, $\theta_1(t)$, and $\theta_0(t)$ satisfies the error estimations (16).

**Step 4.1.** We estimate $\theta_2(t)$ first.

We first show that $1 - t\tau_i(t)$ has a positive lower bound. Recall that $\tau_i(t)$ is the diagonal element of $\widetilde{u}_2(t) = Q(t)^\top\theta_2(0)Q(t)$. We first observe that any diagonal element for

$$Q(t)^\top(I - t(I - A))Q(t) = Q(t)^\top((1 - t)I + tA)Q(t)$$

must be larger than or equal to $\min\{1, \lambda_A\}$. By (41), $1 - t\tau_i(t)$, as a diagonal element of

$$I - tQ(t)^\top\theta_2(0)Q(t) = Q(t)^\top\left[(1 - t)I + tA + t\left(I - A - \theta_2(0)\right)\right]Q(t)$$

must satisfies

$$1 - t\tau_i(t) \ge 1 - t + t\lambda_A - tC\mathcal{L}_{\mathrm{MMD}}^{\frac{1}{4}} \ge \min\{1, \lambda_A\} - C\mathcal{L}_{\mathrm{MMD}}^{\frac{1}{4}} \ge \frac{1}{2}\min\{1, \lambda_A\}.$$

Therefore, $1 - t\tau_i(t)$ has a positive lower bound.

Next, we claim that, for any (fixed) $i$, $1 + t\lambda_i(t)$ has a positive lower bound. Similar to *step 3*, (33) can be rewritten as

$$|1 + t\lambda_i(t)| \, |1 - t\tau_i(t)| \left| \lambda_i(t) - \frac{\tau_i(t)}{1 - t\tau_i(t)} \right| \le C\mathcal{L}_{\text{HJ}}^{\frac{1}{3}}.$$

Therefore, the lower bound for $1 - t\tau_i(t)$ implies

$$|1 + t\lambda_i(t)| \left| \lambda_i(t) - \frac{\tau_i(t)}{1 - t\tau_i(t)} \right| \le C\mathcal{L}_{\text{HJ}}^{\frac{1}{3}} \tag{42}$$

for all $t \in [0, 1]$ and $i$. If we further restrict ourself to $t \in [\frac{1}{3K}, 1]$, we have

$$\left| \lambda_i(t) - (-\frac{1}{t}) \right| \left| \lambda_i(t) - \frac{\tau_i(t)}{1 - t\tau_i(t)} \right| \le C\mathcal{L}_{\text{HJ}}^{\frac{1}{3}} \quad \text{for } t \in [\frac{1}{3K}, 1], \tag{43}$$

implying that $\lambda_i(t)$ must be close to either $-\frac{1}{t}$ or $\dfrac{\tau_i(t)}{1 - t\tau_i(t)}$.

Next, we show the lower bound for $1 + t\lambda_i(t)$. For $t \in [0, \frac{1}{3K}]$,

$$1 + t\tau_i(t) \ge 1 - tK \ge \frac{2}{3}.$$

Therefore, we must have $\forall \, t \in [0, \frac{1}{3K}]$

$$\left| \lambda_i(t) - \frac{\tau_i(t)}{1 - t\tau_i(t)} \right| \le C\mathcal{L}_{\text{HJ}}^{\frac{1}{3}}. \tag{44}$$

For $t \in [\frac{1}{3K}, 1]$, similar to the argument in *step 3*, by (43), $\lambda_i(t)$ must be close to either $-\frac{1}{t}$ or $\frac{\tau_i(t)}{1 - t\tau_i(t)}$, but cannot be close to both because the difference between the two functions has a positive lower bound of $\mathcal{O}(1)$:

$$\left| -\frac{1}{t} - \frac{\tau_i(t)}{1 - t\tau_i(t)} \right| = \left| \frac{1}{t} \frac{1}{1 - t\tau_i(t)} \right| \ge \left| \frac{1}{1 - t\tau_i(t)} \right| \ge \frac{1}{\max\{1, |\tau_i(t)|\}}$$
$$\ge \frac{1}{\max\{1, \|I - A\|_2 + C\mathcal{L}_{\text{MMD}}^{\frac{1}{4}}\}} \ge \frac{1}{2\max\{1, \|I - A\|_2\}} =: c_{\text{diff}}, \tag{45}$$

where the second last inequality is because of (41). (43) and (45) imply that $\lambda_i(t)$ cannot "shift" between $-\frac{1}{t}$ and $\frac{1}{t}\frac{1}{1 - t\tau_i(t)} - \frac{1}{t}$ during $t \in [\frac{1}{3K}, 1]$. Since we already have (44) at $t = \frac{1}{3K}$, (43) implies that $\lambda_i(t)$ is close to $\frac{\tau_i(t)}{1 - t\tau_i(t)}$ for all $t \in [\frac{1}{3K}, 1]$ and hence

$$\left| \lambda_i(t) - (-\frac{1}{t}) \right| \ge c_{\text{diff}} - C\mathcal{L}_{\text{HJ}}^{\frac{1}{3}} \ge \frac{1}{2} c_{\text{diff}} = \mathcal{O}(1).$$

Therefore, for all $t \in [\frac{1}{3K}, 1]$

$$|1 + t\lambda_i(t)| \ge \frac{c_{\text{diff}}}{6K} = \mathcal{O}(1).$$

Combining the lower bound for $t \in [0, \frac{1}{3K}]$, we finish proving the claim that $1 + t\lambda_i(t)$ has a positive lower bound of $\mathcal{O}(1)$ that is independent of $i$. This positive lower bound also implies that $I + t\theta_2(t)$ is invertible and has a positive lower bound (recall $\lambda_i(t)$ are eigenvalues of $\theta_2(t)$). Therefore, by definition of $\Gamma_2(t)$ and (28),

$$\|(I - t\theta_2(0))\theta_2(t) - \theta_2(0)\|_F = \left\| (I + t\theta_2(t))^{-1}\Gamma_2(t) \right\|_F \le C\mathcal{L}_{\text{HJ}}^{\frac{1}{3}}. \tag{46}$$

Next, we give a positive lower bound for $I - t\theta_2(0)$. Note that

$$I - t\theta_2(0) = (1 - t)I + tA + t(I - A - \theta_2(0)).$$

By (41), we know that the smallest eigenvalue of $I - t\theta_2(0)$ is larger than or equal to

$$(1 - t) + t\lambda_A - tC\mathcal{L}_{\text{MMD}}^{\frac{1}{4}} \ge \frac{1}{2}\min\{1, \lambda_A\} = \mathcal{O}(1),$$

which gives a lower bound for $I - t\theta_2(0)$. Applying this bound to (46), we obtain

$$\left\| \theta_2(t) - (I - t\theta_2(0))^{-1} \theta_2(0) \right\|_F \le C\mathcal{L}_{\mathrm{HJ}}^{\frac{1}{3}}. \tag{47}$$

We further notice that by (41)

$$\left\| (I - t(I - A))^{-1} (I - A) - (I - t\theta_2(0))^{-1} \theta_2(0) \right\|_F$$
$$\le \left\| (I - t(I - A))^{-1} (I - A - \theta_2(0)) \right\|_F$$
$$+ \left\| (I - t(I - A))^{-1} t (I - A - \theta_2(0)) (I - t\theta_2(0))^{-1} \theta_2(0) \right\|_F$$
$$\le \min\{1, \lambda_A\}^{-1} \cdot C\mathcal{L}_{\mathrm{MMD}}^{\frac{1}{4}} + \min\{1, \lambda_A\}^{-1} \cdot C\mathcal{L}_{\mathrm{MMD}}^{\frac{1}{4}} \cdot 2 \min\{1, \lambda_A\}^{-1} \cdot K$$
$$= C\mathcal{L}_{\mathrm{MMD}}^{\frac{1}{4}}.$$

Therefore, for any $t \in [0, 1]$,

$$\|\theta_2(t) - \theta_2^*(t)\|_F = \left\| \theta_2(t) - (I - t(I - A))^{-1} (I - A) \right\|_F$$
$$\le \left\| \theta_2(t) - (I - t\theta_2(0))^{-1} \theta_2(0) \right\|_F + \left\| (I - t\theta_2(0))^{-1} \theta_2(0) - (I - t(I - A))^{-1} (I - A) \right\|_F$$
$$\le C \left( \mathcal{L}_{\mathrm{HJ}}^{\frac{1}{3}} + \mathcal{L}_{\mathrm{MMD}}^{\frac{1}{4}} \right). \tag{48}$$

**Step 4.2.** We verify that $\theta_1(t)$ has small error. We first give an error estimation for $\theta_1(0)$. Recall the true value is

$$\theta_1^*(0) = A\mathbf{b}_\mu - \mathbf{b}_\nu = (I - \theta_2^*(0))\mathbf{b}_\mu - \mathbf{b}_\nu.$$

Therefore

$$\|\theta_1(0) - \theta_1^*(0)\|$$
$$\le \|\theta_1(0) - ((I - \theta_2(0))\mathbf{b}_\mu - \mathbf{b}_\nu)\| + \|(I - A - \theta_2(0))\mathbf{b}_\mu\| \tag{49}$$
$$\le C\mathcal{L}_{\mathrm{MMD}}^{\frac{1}{2}} + C\mathcal{L}_{\mathrm{MMD}}^{\frac{1}{4}} \le C\mathcal{L}_{\mathrm{MMD}}^{\frac{1}{4}}$$

where we used (24) and (41) in the second inequality. Next, we give error estimate of $\theta_1(t)$ for $t \in [0, 1]$. Recall that

$$\Gamma_1(t) = (I + t\theta_2(t)) (\theta_1(t) - t\theta_2(0)\theta_1(t) - \theta_1(0))$$

satisfies the estimation (28). Since $I + t\theta_2(t)$ has a positive lower bound (shown in *step 4.1*), we have

$$\|\theta_1(t) - t\theta_2(0)\theta_1(t) - \theta_1(0)\| = \left\| (I + t\theta_2(t))^{-1}\Gamma_1(t) \right\| \le C\mathcal{L}_{\mathrm{HJ}}^{\frac{1}{3}}. \tag{50}$$

Therefore, for any $t \in [0, 1]$,

$$\|\theta_1(t) - \theta_1^*(t)\| = \left\| \theta_1(t) - ((1 - t)I + At)^{-1} (A\mathbf{b}_\mu - \mathbf{b}_\nu) \right\|$$
$$= \left\| \theta_1(t) - (I - t\theta_2^*(0))^{-1} \theta_1^*(0) \right\| \le C \left\| (I - t\theta_2^*(0)) \theta_1(t) - \theta_1^*(0) \right\|$$
$$\le C (\|(I - t\theta_2(0)) \theta_1(t) - \theta_1(0)\| + \|t(\theta_2(0) - \theta_2^*(0))\theta_1(t)\| + \|\theta_1(0) - \theta_1^*(0)\|) \tag{51}$$
$$\le C \left( \mathcal{L}_{\mathrm{HJ}}^{\frac{1}{3}} + \mathcal{L}_{\mathrm{MMD}}^{\frac{1}{4}} + \mathcal{L}_{\mathrm{MMD}}^{\frac{1}{4}} \right) \le C \left( \mathcal{L}_{\mathrm{HJ}}^{\frac{1}{3}} + \mathcal{L}_{\mathrm{MMD}}^{\frac{1}{4}} \right).$$

In the third inequality, we used (50), (41), and (49).

**Step 4.3.** Finally, we verify that $\theta_0(t)$ has small error. Recall that $\theta_0^*(t)$ is uniquely defined up to an additive constant and

$$\theta_0^*(t) - \theta_0^*(0) = \frac{t}{2}(\mathbf{b}_\nu - A\mathbf{b}_\mu)^\top ((1 - t)I + At)^{-1} (\mathbf{b}_\nu - A\mathbf{b}_\mu)$$
$$= \frac{t}{2}\theta_1^*(0)^\top (I - t\theta_2^*(0))^{-1} \theta_1^*(0) = \frac{t}{2}\theta_1^*(t)^\top \theta_1^*(0).$$

Also recall that

$$\Gamma_0(t) = \theta_0(t) - \theta_0(0) + \frac{t}{2}\theta_1(t)^\top \theta_1(t) - \frac{t^2}{2}\theta_1(t)^\top \theta_2(0)\theta_1(t) - t\theta_1(t)^\top \theta_1(0)$$

$$= \theta_0(t) - \theta_0(0) + \frac{t}{2}\theta_1(t)^\top \left[(I - t\theta_2(0))\theta_1(t) - \theta_1(0)\right] - \frac{t}{2}\theta_1(t)^\top \theta_1(0).$$

Therefore,

$$\left|(\theta_0(t) - \theta_0(0)) - (\theta_0^*(t) - \theta_0^*(0))\right|$$

$$= \left|\Gamma_0(t) - \frac{t}{2}\theta_1(t)^\top \left[(I - t\theta_2(0))\theta_1(t) - \theta_1(0)\right] + \frac{t}{2}\theta_1(t)^\top \theta_1(0) - \frac{t}{2}\theta_1^*(t)^\top \theta_1^*(0)\right|$$

$$\leq C\left[\mathcal{L}_{\mathrm{HJ}}^{\frac{1}{3}} + \frac{t}{2}K\mathcal{L}_{\mathrm{HJ}}^{\frac{1}{3}} + \frac{t}{2}\left(|\theta_1(t) - \theta_1^*(t)|\,|\theta_1(0)| + |\theta_1^*(t)|\,|\theta_1(0) - \theta_1^*(0)|\right)\right] \tag{52}$$

$$\leq C\left(\mathcal{L}_{\mathrm{HJ}}^{\frac{1}{3}} + \mathcal{L}_{\mathrm{MMD}}^{\frac{1}{4}}\right),$$

where (28) and (50) are used in the first inequality. (51) and (49) are used in the second inequality.

Finally, combining (48), (51), and (52), we conclude

$$\max_{t \in [0,1]} \left(\|\theta_2(t) - \theta_2^*(t)\|_F + \|\theta_1(t) - \theta_1^*(t)\| + |\theta_0(t) - \theta_0^*(t)|\right) \leq C\left(\mathcal{L}_{\mathrm{HJ}}^{\frac{1}{3}} + \mathcal{L}_{\mathrm{MMD}}^{\frac{1}{4}}\right),$$

which implies (16). $\qquad\square$

Finally we make two remarks to this stability analysis. First, we omit the discretization error and generalization error in this analysis in order to obtain a clear environment for studying the loss function. Second, while we prove the stability result in Gaussian setting, we believe the stability result hold for general distributions, as long as they belongs to some class with sufficient regularity condition.

## B    IMPLEMENTATION DETAILS

The implementation and training code is available at https://github.com/Yebbi/NCF. The code for section 6.2.2 is provided in https://github.com/LSLSliushu/NCF_Class_conditional_OT.

**Empirical Loss via Monte Carlo Approximation.**    In practice, the training loss function is approximated via Monte Carlo estimation. For $\mathcal{L}_{\mathrm{HJ}}$ (13), we set $\varrho$ as the uniform distribution on a compact computation domain $D \subset \Omega$. We uniformly sample a batch of collocation points $\{(\mathbf{x}^{(i)}, t_i)\}_{i=1}^N$ from the space-time computational domain $D \times [0, t_f]$ to obtain the empirical loss

$$\mathcal{E}_{\mathrm{HJ}} = \frac{1}{N}\sum_{i=1}^N \left(u_\theta^{(i)} + t_i h\left(\nabla u_\theta^{(i)}\right) - t_i \nabla u_\theta^{(i)\top}\nabla h\left(\nabla u_\theta^{(i)}\right) - u_\theta^{(i)}\left(\mathbf{x}^{(i)} - t_i\nabla h\left(\nabla u_\theta^{(i)}\right), 0\right)\right)^2,$$

where $u_\theta^{(i)} = u(\mathbf{x}^{(i)}, t_i)$. Similarly, the MMD term (12) is estimated empirically through samples $\{(\mathbf{x}^{(i)}, \mathbf{y}^{(i)})\}_{i=1}^N$ from the initial and target distributions

$$\mathcal{E}_{\mathrm{MMD}} = \frac{1}{N^2}\sum_{i,j=1}^N \left(k(\widetilde{\mathbf{x}}^{(i)}, \widetilde{\mathbf{x}}^{(j)}) + k(\mathbf{y}^{(i)}, \mathbf{y}^{(j)}) - 2k(\widetilde{\mathbf{x}}^{(i)}, \mathbf{y}^{(j)})\right),$$

where $\widetilde{\mathbf{x}}^{(i)} = \mathbf{x}^{(i)} + t_f\nabla u_\theta(\mathbf{x}^{(i)}, 0)$. To better learn bidirectional OT, we employ the MMD loss in both forward and backward directions. The total loss is

$$\min_\theta \mathcal{E}_{\mathrm{HJ}}(u_\theta) + \lambda \mathcal{E}_{\mathrm{MMD}}\left((T_\mu^\nu[u_\theta])_\sharp \mu, \nu\right) + \lambda \mathcal{E}_{\mathrm{MMD}}(\mu, (T_\nu^\mu[u_\theta])_\sharp \nu). \tag{53}$$

### B.1    IMPLEMENTATION DETAILS FOR 2D EXPERIMENTS

**Training.**    For the experiments for 2D toy distributions in Sections 6.1.1 and 6.2.1, we use a simple 5-layer MLP with hidden dimension 64 and Softplus activation (with $\beta = 100$). The model is trained using the Adam optimizer with a learning rate of $10^{-3}$. We sampled 50,000 points from each distribution to create the corresponding sample datasets. At each training epoch, we uniformly sample 1,000 collocation points from the computational domain $D = [-1, 1]^2$ to compute the implicit solution formula loss (13). For the MMD loss, we randomly select 750 samples from the given dataset at each epoch.

**Baselines.** For the NOT baseline, we follow the official implementation provided in the public repository[1] without any modification. The HJ-PINN ablation model was trained under the exact same experimental settings as our proposed NCF across all experiments. For GNOT in the class-conditional setting, we use the official code released by the authors[2] without modification.

## B.2 IMPLEMENTATION DETAILS FOR GAUSSIAN EXPERIMENTS

**Training.** For the high-dimensional Gaussian experiments in Section 6.1.2, we employ the DenseICNN architecture, which is a fully connected neural network with additional input-quadratic skip connections, to ensure a fair comparison with the baseline models provided in (Korotin et al., 2021a). Since our method does not require input convexity, we omit the commonly imposed constraints that enforce positivity of certain neural network weights, which are typically used to guarantee convexity. Following Korotin et al. (2021a), we adopt the network architecture DenseICNN[1; max($2d$,64), max($2d$,64), max($d$,32)] for a $d$-dimensional problem. The model is optimized using Adam with a fixed learning rate of $10^{-4}$, regardless of the input dimension. To construct dataset, we randomly sample $10^5$ points from each of the source and target distributions. We set $D$ as the bounding box (i.e., axis-aligned minimum and maximum values) of these samples and define it as the computational domain for solving the HJ equation. At each training epoch, we uniformly sample 1,000 collocation points from $D$ to compute the implicit solution formula loss (13). For the MMD loss, we randomly select 2,000 points from the given source and target datasets at every epoch.

**Baselines.** The baselines LS, WGAN-QC, MM-v1 and MM:R are all used via the official implementations from the public repository of Korotin et al. (2021a)[3]. The implementations of NOT and HJ-PINNs follow the same settings described in Appendix B.1.

**Evaluation Metric.**

- **Unexplained Variance Percentage (UVP):** Given the predicted transport map $\hat{T}$ from $\mu$ to $\nu$, UVP is defined by $\mathcal{L}^2 - \text{UVP}\left(\hat{T}\right) := 100 \left\|\hat{T} - T^*\right\|_{L^2(\mu)} / \text{Var}\left(\nu\right)$ (%). A UVP value approaching 0% indicates that $\hat{T}$ provides a close approximation to the OT map $T^*$, whereas values substantially exceeding 100% imply that the estimated map fails to capture the underlying structure of the OT. We use $10^5$ random samples drawn from $\mu$ to compute UVP.

- **Memory and Time Metrics:** Memory consumption is reported as the peak memory usage during training. Training time is measured as the average runtime per epoch over 100 epochs. Inference time refers to the time required to transport $10^5$ test samples using the learned map. Additionally, we measure the memory required to store the trained networks for the bidirectional OT maps. For our method, this corresponds to the storage size of a single spatio-temporal solution function for the HJ equation. For dual-based baselines, this reflects the memory needed to store both the primal and dual potential functions. For the NOT baseline, which learns the forward and backward OT maps separately, we report the total memory required to store both learned transport maps.

## B.3 IMPLEMENTATION DETAILS FOR COLOR TRANSFER

**Training.** The color transfer experiments in Section 6.1.3 are trained using exactly the same experimental setup as in the high-dimensional Gaussian case described in Appendix B.2, to ensure a fair comparison with the baseline models.

**Baselines.** For the classical methods, we implemented Reinhard color transfer using OpenCV's Bradski & Kaehler (2008) color space conversion and channel-wise mean-std matching. Histogram matching was implemented by computing per-channel histograms and CDFs, then applying the resulting pixel value mapping directly. Since these methods only support one-way transfer, we conducted forward and backward transfers separately. Both methods serve as standard, straightforward baselines.

---

[1] https://github.com/iamalexkorotin/NeuralOptimalTransport
[2] https://github.com/machinestein/GNOT
[3] https://github.com/iamalexkorotin/Wasserstein2Benchmark

**Evaluation Metrics.**

- **Earth-Mover Distance (EMD):** For both the target and transported images, we compute normalized color histograms separately for each BGR channel. The EMD quantifies the minimal cost required to transform one histogram into another, offering a perceptually meaningful measure of distributional difference. We compute the EMD independently for each channel and report the average across all three. Lower EMD values indicate greater similarity.

- **Histogram Intersection (HI):** HI measures the overlap between the normalized color histograms of the target and transported images. For each BGR channel, we compute the intersection as the sum of the minimum values across corresponding bins. The final score is obtained by averaging over all three channels. Higher values (closer to 1) indicate greater similarity.

### B.4 IMPLEMENTATION DETAILS FOR SECTION 6.2.2

This series of experiments focuses on the MNIST dataset (LeCun, 1998), which comprises 10 classes of $28 \times 28$ grayscale images of handwritten digits ranging from 0 to 9; And the Fashion MNIST dataset consisting of 10 classes of $28 \times 28$ grayscale images of clothing items, labeled from 0 to 9.

For both MNIST & Fashion MNIST datasets, the value of each pixel of the grayscale images takes integer value from 1 to 255. We always normalized the pixel values of each data point to $[0, 1]$ by dividing by 255 before calculation.

**VAE Pretraining.** In our study, we employ pretrained $\beta$-VAE models (Kingma et al., 2013; Higgins et al., 2017), which offer satisfactory generative quality and faithful manifold representations for image encoding. Advanced auto-encoder architectures (Berthelot et al., 2018; Feng & Strohmer, 2024) that better preserve the interpolation quality of decoded images will be considered in future work. Although the ambient dimension of the data is 784, prior work has shown that the dataset exhibits a moderately low intrinsic dimension (Pope et al., 2021). Thus, in our implementation, we set the latent dimension to $d_l = 10$ for in-domain transport tasks on MNIST data set, and to $d_l = 35$ for cross-domain transport between Fashion MNIST and MNIST data sets.

To train the VAE, we consider the encoder $E_\phi(\cdot) : \mathbb{R}^d \to \mathbb{R}^{d_l}$, and encoding variance $S_\phi(\cdot) : \mathbb{R}^d \to \mathbb{R}^{d_l}$, which share the same parameter $\phi$, together with the decoder $D_\omega(\cdot) : \mathbb{R}^{d_l} \to \mathbb{R}^d$, with $\phi, \omega$ being the tunable parameters. For arbitrary $\mathbf{x}_i$ from the dataset and the latent variable $\mathbf{z} \in \mathbb{R}^{d_l}$, the ELBO-type loss $\mathcal{L}_\beta(\phi, \omega; \mathbf{x}_i) := \mathbb{E}_{\mathbf{z} \sim q_\phi(\mathbf{z}|\mathbf{x}_i)} \log p_\omega(\mathbf{x}_i|\mathbf{z}) - \beta D_{\mathrm{KL}}(q_\phi(\cdot|\mathbf{x}_i) \| p_{\mathbf{z}}(\cdot))$ is considered, where we set the conditional probability $p_\omega(\cdot|\mathbf{z}) = \mathcal{N}(D_\omega(\mathbf{z}), \sigma_*^2 \mathbf{I}_d)$, the prior $p_{\mathbf{z}}(\cdot) = \mathcal{N}(\mathbf{0}, \mathbf{I}_{d_l})$, and the posterior $q_\phi(\cdot|\mathbf{x}_i) = \mathcal{N}(E_\phi(\mathbf{x}_i), \Sigma_\phi(\mathbf{x}_i))$. Here $\sigma_*^2$ is predetermined variance, and $\Sigma_\phi(\mathbf{x}_i) = \exp(\mathrm{diag}(S_\phi(\mathbf{x}_i)))$. We optimize the following to obtain $E_\phi, D_\omega$:

$$\max_{\phi, \omega} \frac{1}{M} \sum_{i=1}^{M} \mathcal{L}_\beta(\phi, \omega; \mathbf{x}_i) = -\frac{1}{2M} \left( \sum_{i=1}^{M} \frac{1}{\sigma_*^2} \mathbb{E}_{\boldsymbol{\epsilon} \sim \mathcal{N}(\mathbf{0}, \mathbf{I})} \| \mathbf{x}_i - D_\omega(E_\phi(\mathbf{x}_i) + \sqrt{\Sigma_\phi(\mathbf{x}_i)} \odot \boldsymbol{\epsilon}) \|^2 \right.$$
$$\left. + \beta(-S_\phi(\mathbf{x}_i)^\top \mathbf{1} + \| E_\phi(\mathbf{x}_i) \|^2 + \exp(S_\phi(\mathbf{x}_i))^\top \mathbf{1}) \right).$$

Here we denote $\mathbf{1} = (1, \ldots, 1) \in \mathbb{R}^{d_l}$. In our experiment, we pick $\sigma_*^2 = \frac{1}{100}$, and set $\beta = 0.1$ to ensure reconstruction fidelity over regularization.

We train the VAE pairs $(E_\phi^1(\cdot), D_\omega^1(\cdot))$ and $(E_\phi^2(\cdot), D_\omega^2(\cdot))$ on MNIST dataset $\{\mathbf{x}_i^{(1)}\}$ and Fashion MNIST dataset $\{\mathbf{x}_i^{(2)}\}$ respectively. We set batch size as 32, and apply the Adam algorithm (Kinga et al., 2014) with learning rate $10^{-4}$ for 150 epochs. In practice, the trained VAE reproduces MNIST images with an accuracy 98.2%, and reproduces Fashion MNIST images with an accuracy 87.0%.

**Encoding & Normalization.** Denote $\mathbf{y}_i^{(k)} = E_\phi^k(\mathbf{x}_i^{(k)})$, $k = 1, 2$, we normalize the latent samples $\{\mathbf{y}_i^{(k)}\}_{1 \le i \le N}$ by $\widetilde{\mathbf{y}}_i^{(k)} = (\boldsymbol{\sigma}^{(k)})^{-1}(\mathbf{y}_i^{(k)} - \bar{\mathbf{y}}^{(k)})$ for $1 \le i \le N, k = 1, 2$. Here we denote $\bar{\mathbf{y}}^{(k)} = \frac{1}{N} \sum_{i=1}^{N} \mathbf{x}_i^{(k)}$ as the mean of the dataset, and $\boldsymbol{\sigma}^{(k)} = \mathrm{diag}(\boldsymbol{\Sigma}^{(k)})$ as the entrywise variance, where

$\mathrm{diag}(\boldsymbol{\Sigma}^{(k)})$ denotes a diagonal matrix with its diagonal entries taken from the empirical covariance matrix $\boldsymbol{\Sigma}^{(k)} = \frac{1}{N}\sum_{i=1}^{N}(\mathbf{x}_i^{(k)} - \bar{\mathbf{x}}^{(k)})(\mathbf{x}_i^{(k)} - \bar{\mathbf{x}}^{(k)})^{\top}$.

**Loss function & Training.** We denote $\mu, \nu$ as the distribution of the normalized latent samples $\widetilde{\mathbf{y}}_i^{(k)}$, where $k = 1$ or $2$. To compute the OT map between $\mu, \nu$, we set $t_f = 1$, and introduce neural network $u_\theta : \mathbb{R}^{d_l} \times [0, t_f] \to \mathbb{R}$. In practice, we incorporate the loss functional for backward OT into the original loss (14), that is, we consider

$$\min_{\theta}\left\{\mathcal{L}_{\overrightarrow{\mathrm{HJ}}}(u_\theta) + \mathcal{L}_{\overleftarrow{\mathrm{HJ}}}(u_\theta) + \lambda(\mathcal{E}_{\mathrm{class}}((T_\mu^\nu[u_\theta])_\sharp\,\mu, \nu) + \mathcal{E}_{\mathrm{class}}(\mu, (T_\nu^\mu[u_\theta])_\sharp\,\nu))\right\},$$

where we denote $\mathcal{L}_{\overrightarrow{\mathrm{HJ}}}(u_\theta) := \mathcal{L}_{\mathrm{HJ}}(u_\theta)$ as defined in (13), and define the corresponding backward implicit loss as

$$\mathcal{L}_{\overleftarrow{\mathrm{HJ}}}(u_\theta) = \iint_{\Omega \times [0, t_f]}\left(u_\theta - th\,(\nabla u_\theta) + t\nabla u_\theta^\top \nabla h\,(\nabla u_\theta) - u_\theta\,(\mathbf{x} + t\nabla h\,(\nabla u_\theta), t_f)\right)^2 \mathrm{d}\varrho(\mathbf{x})\,\mathrm{d}t.$$

Since the latent samples are normalized as described above, we set $\varrho = \mathcal{N}(\mathbf{0}, \mathbf{I}_{d_l})$ and independently draw $x_i \sim \varrho$ and $t_i$ uniformly from $[0, t_f]$ to form the collocation points $\{(x_i, t_i)\}$ for approximating $\mathcal{L}_{\overrightarrow{\mathrm{HJ}}}$ and $\mathcal{L}_{\overleftarrow{\mathrm{HJ}}}$. In implementation, we set $\lambda = 500$ in order to balance the scales of the implicit HJ loss and the MMD loss. We denote $N_{\mathrm{HJ}}, N_{\mathrm{MMD}}$ as the batch size for evaluating the implicit loss and MMD between distributions of certain classes. In our experiments, we choose $N_{\mathrm{HJ}} = 4000$ and $N_{\mathrm{MMD}} = 400$. We apply the Adam method with learning rate $10^{-4}$ for optimizing $\theta$. The algorithm is conducted for $1000000$ iterations.

**Neural Net Architectures.** The architectures for the $\beta-$VAE encoder and decoder are summarized in Table 5 and Table 6. Regarding the OT map, we parameterize $u_\theta : \mathbb{R}^{d_l+1} \to \mathbb{R}$ using a ResNet architecture He et al. (2016) with depth $L$ and width (hidden dimension) $\widetilde{d} = 128$. Specifically, we define

$$u_\theta(x, t) = f_L \circ f_{L-1} \circ \cdots \circ f_2 \circ f_1(x, t),$$

where each layer $f_k$ is given by

$$f_k(y) = \begin{cases} A_k y + b_k, & k = 1, \quad A_1 \in \mathbb{R}^{\widetilde{d} \times (d_l+1)},\ b_1 \in \mathbb{R}^{\widetilde{d}}, \\ y + \kappa A_k \sigma(y) + b_k, & 2 \le k \le L - 1, \quad A_k \in \mathbb{R}^{\widetilde{d} \times \widetilde{d}},\ b_k \in \mathbb{R}^{\widetilde{d}}, \\ A_k y + b_k, & k = L, \quad A_L \in \mathbb{R}^{1 \times \widetilde{d}},\ b_L \in \mathbb{R}. \end{cases}$$

We use the hyperbolic tangent activation $\sigma(\cdot) = \tanh(\cdot)$ and set the residual scaling parameter $\kappa = 1$. We set $L = 5$ for in-domain transports on MNIST, and use $L = 6$ for cross-domain trasport task between Fashion MNIST and MNIST.

Table 5: Encoder architecture for $\beta-$VAE for image size $(H, W, C) = (28, 28, 1)$, latent dimension $d_l = 10$.

| Layer | Parameters | Output Shape |
|---|---|---|
| Input ($\mathbf{x}$) | $-$ | $(H, W)$ |
| Conv2D | 128 filters, $5 \times 5$, stride 1, ReLU | $(H, W, 128)$ |
| Conv2D | 128 filters, $3 \times 3$, stride 1, ReLU | $(H, W, 128)$ |
| Conv2D | 64 filters, $3 \times 3$, stride 2, ReLU | $(H/2, W/2, 64)$ |
| Conv2D | 64 filters, $3 \times 3$, stride 1, ReLU | $(H/2, W/2, 64)$ |
| Conv2D | 64 filters, $3 \times 3$, stride 1, ReLU | $(H/2, W/2, 64)$ |
| Conv2D | 64 filters, $3 \times 3$, stride 1, ReLU | $(H/2, W/2, 64)$ |
| Conv2D | 64 filters, $3 \times 3$, stride 1, ReLU | $(H/2, W/2, 64)$ |
| Flatten | $-$ | $(H/2 \cdot W/2 \cdot 64)$ |
| Dense | $(16 \cdot H \cdot W, 64)$, ReLU | $(16 \cdot H \cdot W)$ |
| Dense (mean $E_\phi(\mathbf{x})$) | $(64, d_l)$ | $(d_l)$ |
| Dense (log variance $S_\phi(\mathbf{x})$) | $(64, d_l)$ | $(d_l)$ |
| Output (reparam.) | $\mathbf{y} = E_\phi(\mathbf{x}) + \exp(\frac{1}{2}\mathrm{diag}(S_\phi(\mathbf{x}))) \odot \boldsymbol{\epsilon}$ | $(d_l)$ |

**Evaluation Metrics.** All methods are evaluated on the *testing* portions of the MNIST datasets.

Table 6: Decoder architecture for $\beta-$VAE for image size $(H, W, C) = (28, 28, 1)$, latent dimension $d_l = 10$.

| Layer | Parameters | Output Shape |
|---|---|---|
| Input ($\mathbf{y}$) | $-$ | $(d_l)$ |
| Dense | $(d_l, 16 \cdot H \cdot W)$, ReLU | $(16 \cdot H \cdot W)$ |
| Reshape | $-$ | $(H/2, W/2, 64)$ |
| Conv2DTranspose | 64 filters, $3 \times 3$, stride 1, ReLU | $(H/2, W/2, 64)$ |
| Conv2DTranspose | 64 filters, $3 \times 3$, stride 1, ReLU | $(H/2, W/2, 64)$ |
| Conv2DTranspose | 64 filters, $3 \times 3$, stride 1, ReLU | $(H/2, W/2, 64)$ |
| Conv2DTranspose | 64 filters, $3 \times 3$, stride 1, ReLU | $(H/2, W/2, 64)$ |
| Conv2DTranspose | 64 filters, $3 \times 3$, stride 2, ReLU | $(H, W, 64)$ |
| Conv2DTranspose | 128 filters, $3 \times 3$, stride 1, ReLU | $(H, W, 128)$ |
| Conv2DTranspose | 128 filters, $5 \times 5$, stride 1, ReLU | $(H, W, 128)$ |
| Conv2DTranspose | 1 filter, $5 \times 5$, stride 1, ReLU | $(H, W, 1)$ |
| Output | $\mathbf{x} = D_\omega(\mathbf{y})$ | $(H, W)$ |

(a) Source $\mu$    (b) Strong NOT    (c) Weak NOT    (d) HJ-PINN    (e) NCF

(f) Target $\nu$    (g) Strong NOT    (h) Weak NOT    (i) HJ-PINN    (j) NCF

Figure 5: *Checkerboard ($\mu$) $\rightleftarrows$ Eight Gaussians ($\nu$):* The top row shows transport in the direction $\nu \to \mu$, and the bottom row shows $\mu \to \nu$, with $\mu$ and $\nu$ at the leftmost column.

- **Classification Accuracy:** We evaluate the class-wise accuracy of the generated data. Following (Asadulaev et al., 2024), we train ResNet-18 classifiers achieving $98.85\%$ accuracy on the MNIST test set.

- **FID score:** The FID score is evaluated on the entire test set, which consists of approximately 1,000 samples per class.

## C    FURTHER RESULTS

### C.1    ADDITIONAL RESULTS FOR 2D TOY EXAMPLES

"Results on 2D distributions with multiple modes are presented in Figure 5. As in Section 6.1.1, the proposed NCF successfully learns bidirectional OT even in multi-modal settings with a single network. Compared to baselines, it not only transports the distributions more accurately but also produces transport maps with less overlap, indicating that it learns more optimal transport paths.

**Ablation Study on the Regularization Parameter**    We further present an ablation study to investigate the effect of the regularization parameters $\lambda_f$ and $\lambda_b$ in the proposed loss function (14). These parameters control the balance between the implicit HJ loss and the MMD loss. Specifically, the implicit HJ loss promotes the optimality of the transport map by encouraging alignment with the HJ equation, while the MMD loss measures how well the transported distribution matches the target distribution. Since all experiments in the paper are conducted under the setting $\lambda_f = \lambda_b$, we vary $\lambda_f$ to examine how this trade-off influences the learned transport map. We conduct experiments on the two-dimensional examples introduced in Section 6.1.1 and above. The results are summarized in Figure 6. The case $\lambda_f = \infty$ corresponds to training without the implicit HJ loss, using only the MMD loss.

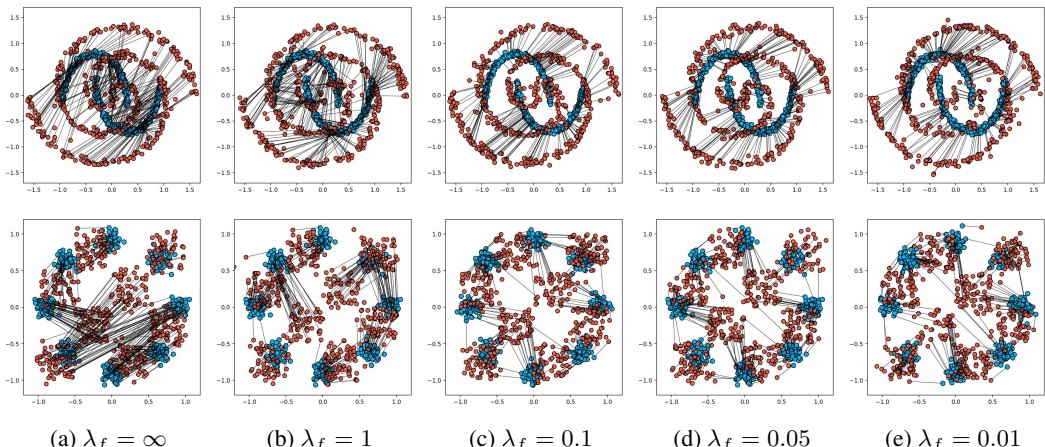

| (a) $\lambda_f = \infty$ | (b) $\lambda_f = 1$ | (c) $\lambda_f = 0.1$ | (d) $\lambda_f = 0.05$ | (e) $\lambda_f = 0.01$ |

Figure 6: Effect of the regularization parameter $\lambda_f$ in the loss function. Each figure visualizes the transport from the source distribution $\mu$ (blue) to the target distribution $\nu$ (red) for varying values of $\lambda_f$. Results are shown for two examples introduced in Section 6.1.1 and Appendix C.1.

As shown in the figure, when $\lambda_f$ is small (i.e., the MMD loss dominates), the transported distribution aligns well with the target, but the resulting transport map becomes highly entangled. This indicates that the model learns a map far from the optimal one, due to the lack of guidance from the HJ constraint. In contrast, increasing $\lambda_f$ enforces stronger adherence to the HJ equation, resulting in a smoother, more structured transport map that closely resembles the optimal solution. However, when $\lambda_f$ becomes too small, the influence of the MMD term diminishes, leading to inaccurate matching of the distributions. These results highlight the complementary roles of the implicit HJ loss and the MMD loss, as also supported by Theorem 5.4, and underscore the importance of appropriately tuning $\lambda_f$. Additionally, the relatively small difference in performance between $\lambda_f = 0.1$ and $\lambda_f = 0.05$ suggests that the model is not overly sensitive to the choice of this parameter.

## C.2 FURTHER RESULTS ON TRAINING STABILITY

Figure 7 illustrates the training stability of our method compared to the representative baseline MM:R in high-dimensional Gaussian experiments in Section 6.1.2. MM:R optimizes a min-max problem, which exhibits severe training instability: although the UVP error initially decreases in the early epochs, it tends to grow substantially as training continues. Following the original paper (Korotin et al., 2021a), the smallest UVP errors observed during training are reported in Table 2. However, in practical OT scenarios the true optimal transport map is unknown, making it unclear when training should be stopped or which solution is preferable. Therefore, the ability to maintain stable training over many epochs, as demonstrated by our method, is a crucial advantage.

In the original MM:R implementation (Korotin et al., 2021a), the model relies on a special identity-potential pretraining to stabilize training. Without this initialization, training becomes highly unstable and can completely fail to approximate the OT map. In contrast, our method does not require any special pretraining and maintains stable UVP reduction throughout training, highlighting a key practical strength.

In contrast, our method demonstrates robust and stable learning, as shown in Figure 7. Our method does not require any special pretraining and maintains stable UVP reduction throughout training, steadily decreasing over epochs. This stability represents a significant practical advantage, ensuring reliable convergence even over long training schedules.

## C.3 ADDITIONAL QUALITATIVE RESULTS FOR COLOR TRANSFER

Figures 8, 9, and 10 present qualitative results for bidirectional color transfer across three distinct categories of image pairs. In each figure, the leftmost columns display the source and target images, while the remaining columns show the results of various methods applied in both the forward (source $\rightarrow$ target) and backward (target $\rightarrow$ source) directions.

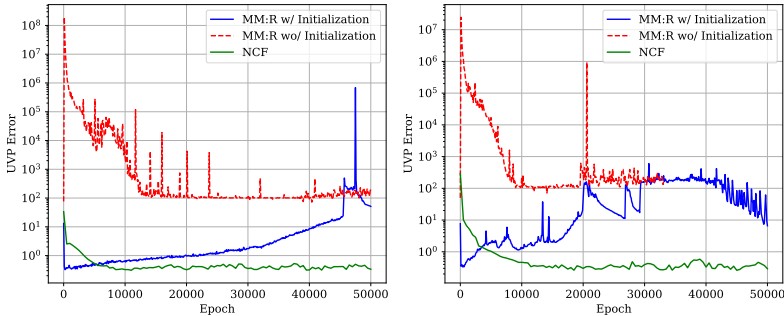

Figure 7: Training stability of our method compared to MM:R on high-dimensional Gaussian experiments.

## C.4 ADDITIONAL RESULTS FOR CLASS-CONDITIONAL OT FOR MNIST & FASHION MNIST DATASETS

In this section, we provide additional results from all the tasks conducted in our experiments. All methods are evaluated on the *testing* portions of the MNIST and Fashion MNIST datasets. Following (Asadulaev et al., 2024), we train ResNet-18 classifiers achieving $98.85\%$ accuracy for evaluation. A generated sample is deemed accurate if the trained classifier assigns it to the corresponding target class. Table 7 reports the accuracy and Fréchet Inception Distance (FID) of the generated images across the first two experimental tasks.

Table 7: Performance metrics of NCF across all tasks. For the FID row, each bracketed value represents the distance between the OT-generated distribution $(D_\omega \circ T_\mu^\nu[u_\theta])_\sharp \mu$ (resp. $(D_\omega \circ T_\nu^\mu[u_\theta])_\sharp \nu$) and the decoded distribution $D_{\omega\sharp}\nu$ (resp. $D_{\omega\sharp}\mu$), where $D_\omega$ denotes the appropriate VAE decoder corresponding to $\mu$ or $\nu$.

| Metric | Task 1 | | Task 2 | |
|---|---|---|---|---|
| | Forward | Backward | Forward | Backward |
| Accuracy(%) ↑ | 95.58 | 95.08 | 92.51 | 92.73 |
| FID ↓ | 19.93 (2.65) | 18.91 (2.45) | 18.98 (2.25) | 19.01 (2.26) |

**Task 1 (In-class transfer: Map each MNIST class $i$ to the class $i + 5$, for $i = 0, \ldots, 4$.)** We present in Figure 11 the uncurated MNIST images generated using the forward (resp. backward) mapping, $D_\omega^1 \circ T_\mu^\nu[u_\theta](\widetilde{\mathbf{y}}^{(1)})$, $\widetilde{\mathbf{y}}^{(1)} \sim \mu$ (resp. $D_\omega^1 \circ T_\nu^\mu[u_\theta](\widetilde{\mathbf{y}}^{(2)})$, $\widetilde{\mathbf{y}}^{(2)} \sim \nu$).

**Task 2 (In-class shift: Map each MNIST class $i$ to the class $(i + 1) \mod 10$, for $i = 0, \ldots, 9$.)** Similar to Task 1, Figure 12 shows the uncurated MNIST images generated by the computed mappings. In this experiment, the forward (resp. backward) maps are also trained without incorporating the implicit HJ loss $\mathcal{L}_{\text{HJ}}$, and are therefore not guaranteed to be optimal. By contrast, the mappings produced by our proposed method—explicitly designed to account for optimality—exhibit superior preservation of MNIST digit styles (e.g., thickness, orientation, etc.), as further illustrated in Figure 13.

**Task 3 (Inter-class transport: Map each class in Fashion MNIST to its corresponding class in MNIST.)** The uncurated MNIST and Fashion MNIST images generated by the computed maps are shown in Figure 14. While our method effectively recovers the overall profiles of Fashion MNIST images, the encoder–decoder scheme faces difficulties in capturing fine texture details. As a future research direction, we aim to enhance our approach by incorporating U-net architectures and directly performing OT in the pixel space. Furthermore, in Figure 15, we present the KDE plots of the pushforward distributions $T_\mu^\nu[u_\theta]_\sharp \mu$ (resp. $T_\nu^\mu[u_\theta]_\sharp \nu$) together with their targets $\nu$ (resp. $\mu$), which demonstrate the satisfactory generative quality of the computed OT map $T_\mu^\nu[u_\theta]$; Figure 16 presents the classification accuracy (%) of the generated images on the test dataset over training iterations. We display only the first 70000 iterations, since the accuracy no longer improves as training progresses.

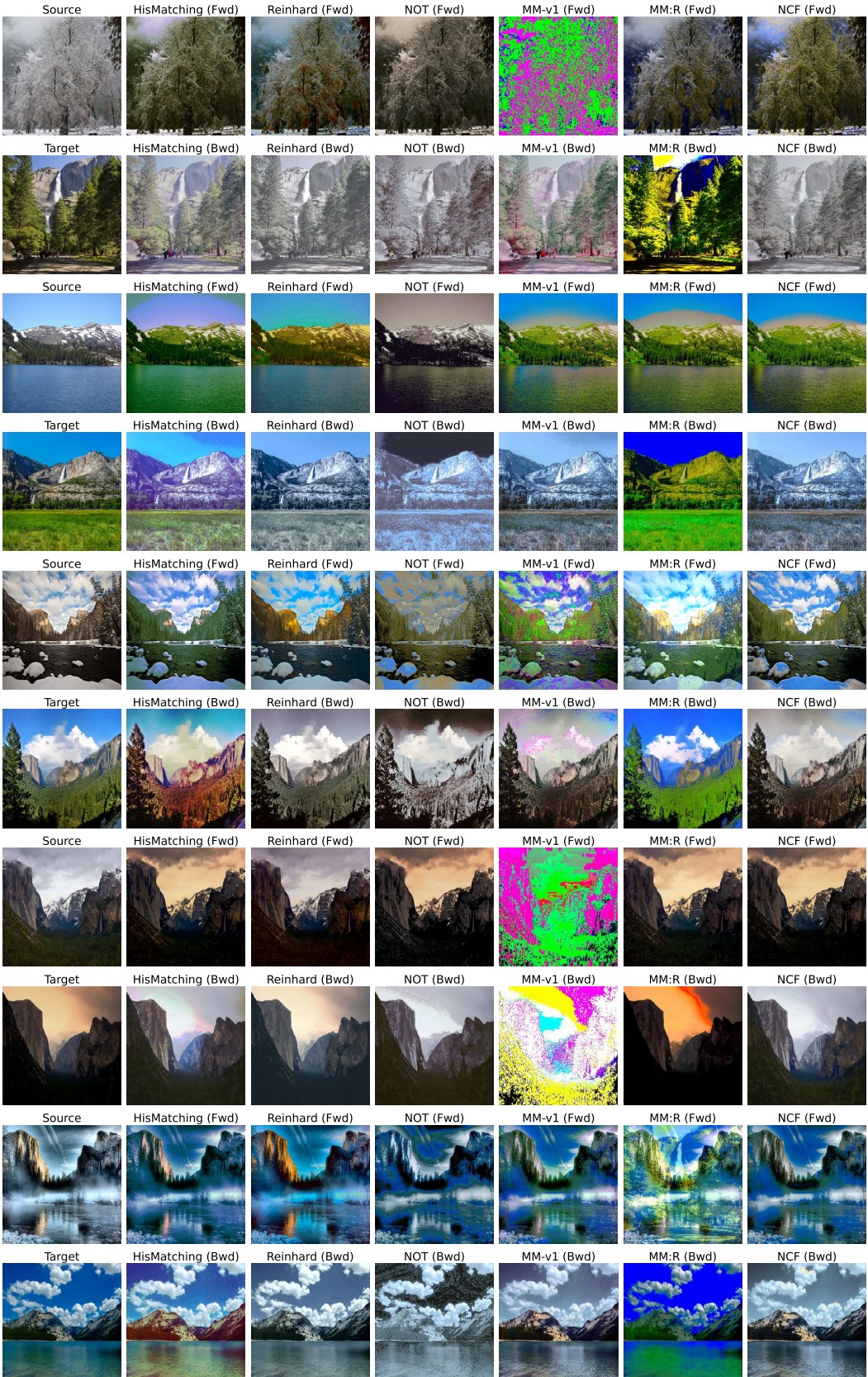

Figure 8: *Winter ↔ Summer:* Qualitative results for bidirectional color transfer between seasonal image pairs.

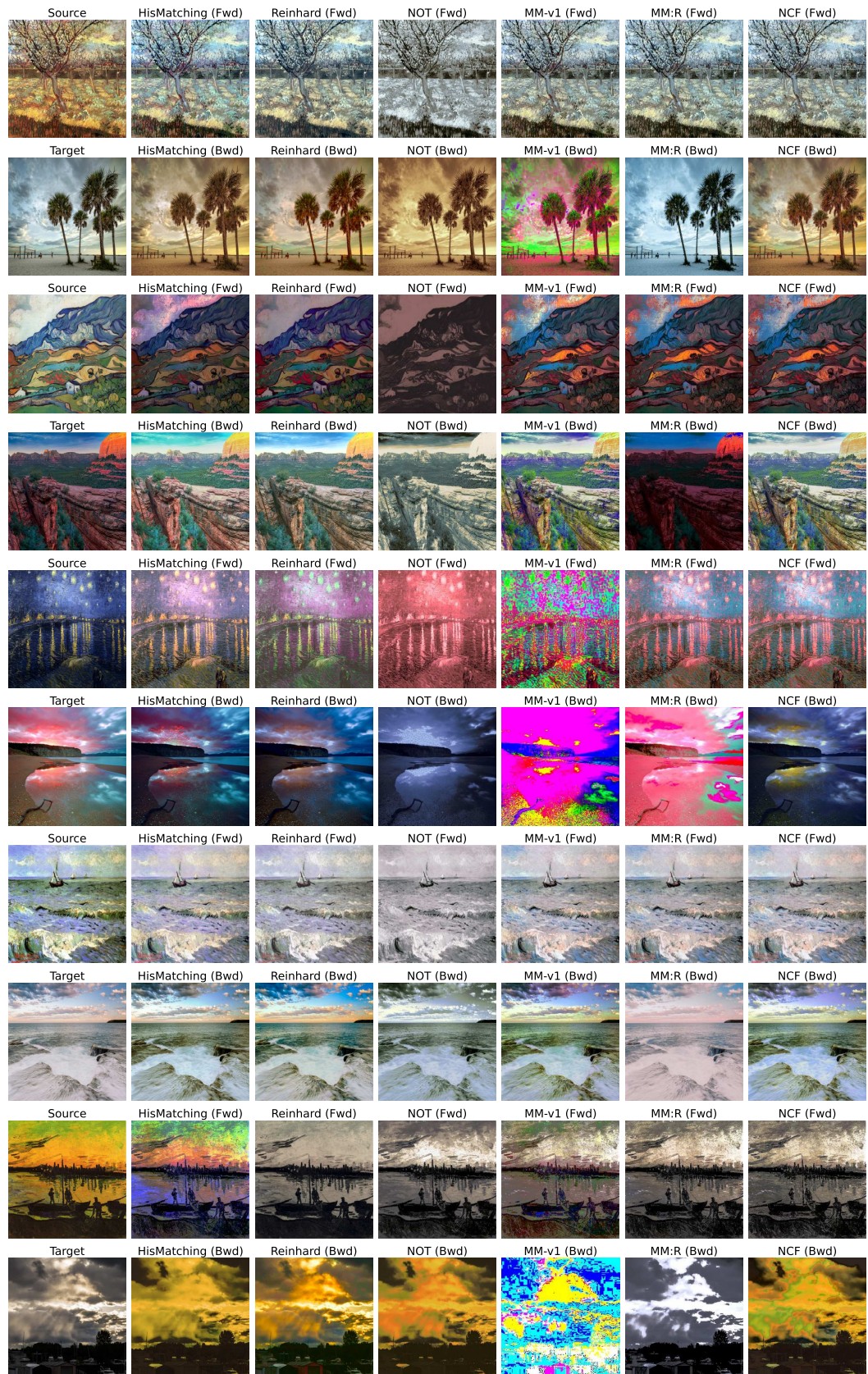

Figure 9: *Gogh painting ↔ Photograph:* Qualitative results for bidirectional color transfer between Gogh paintings and real-world photographs.

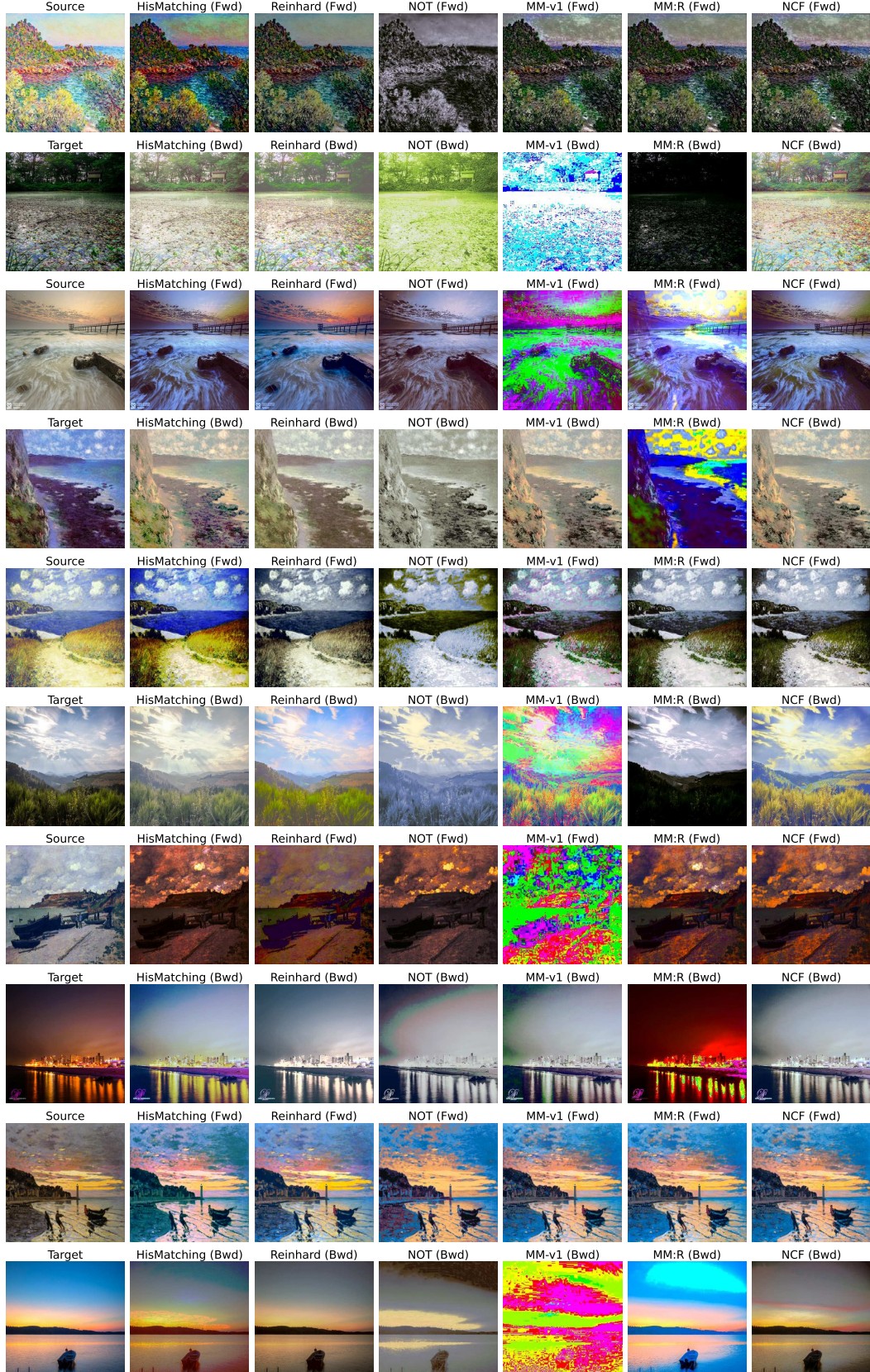

Figure 10: *Monet painting ↔ Photograph:* Qualitative results for bidirectional color transfer between Monet paintings and real-world photographs.

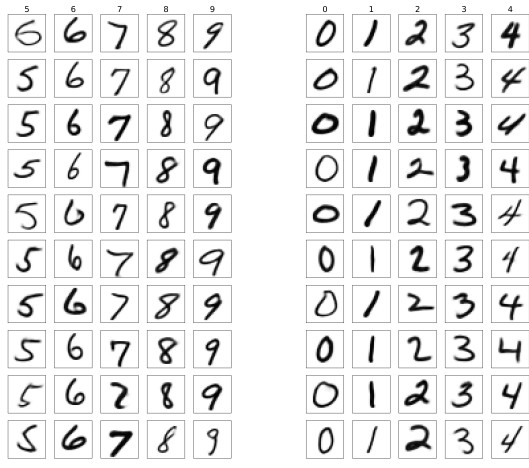

(a) Images generated @ $t_f$.   (b) Images generated @ 0.

Figure 11: Task 1: Uncurated images generated using the computed forward (**Left**) & backward (**Right**) OT maps.

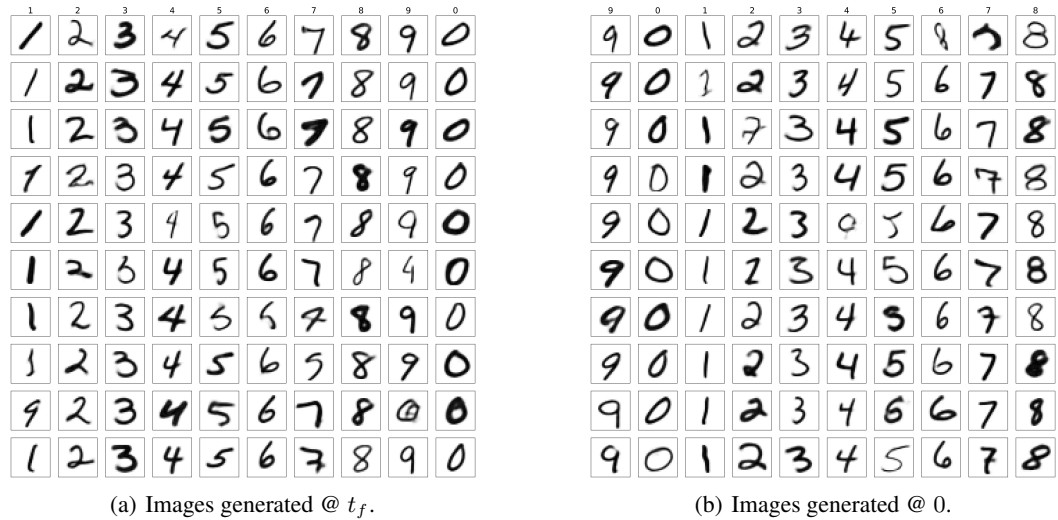

(a) Images generated @ $t_f$.                    (b) Images generated @ 0.

Figure 12: Task 2: Uncurated images generated using the computed forward (**Left**) & backward (**Right**) OT maps.

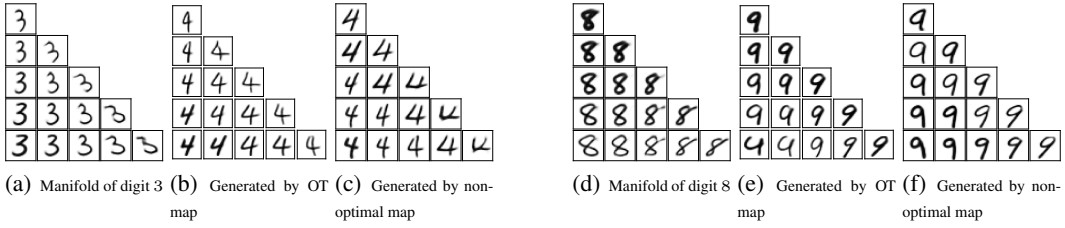

(a) Manifold of digit 3  (b) Generated by OT (c) Generated by non-    (d) Manifold of digit 8 (e) Generated by OT (f) Generated by non-
                          map                optimal map                                     map                optimal map

Figure 13: The style of each MNIST digit is better preserved by the computed optimal transport map. The triangular table is produced using linear interpolation in VAE latent space.

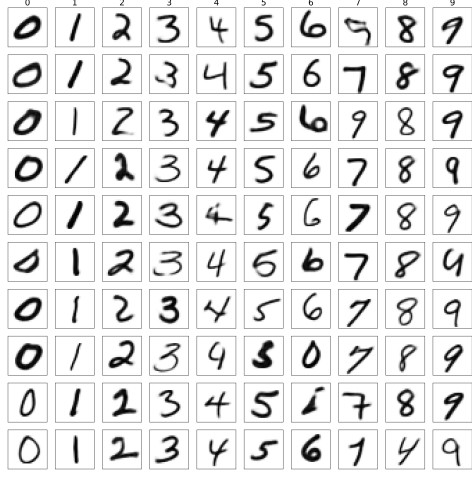

(a) MNIST images generated @ $t_f$.

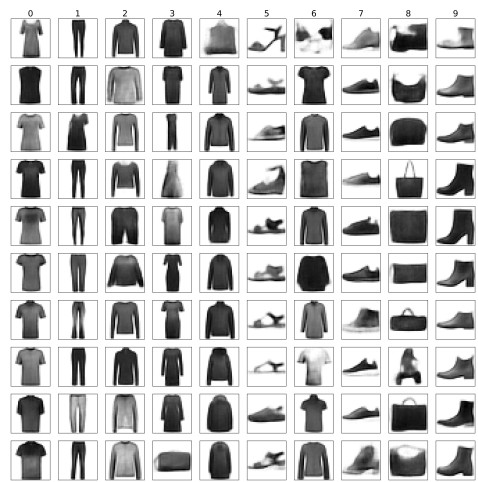

(b) Fashion MNIST images generated @ 0.

Figure 14: Task 3: Uncurated images generated using the computed forward (**Left**) & backward (**Right**) OT maps.

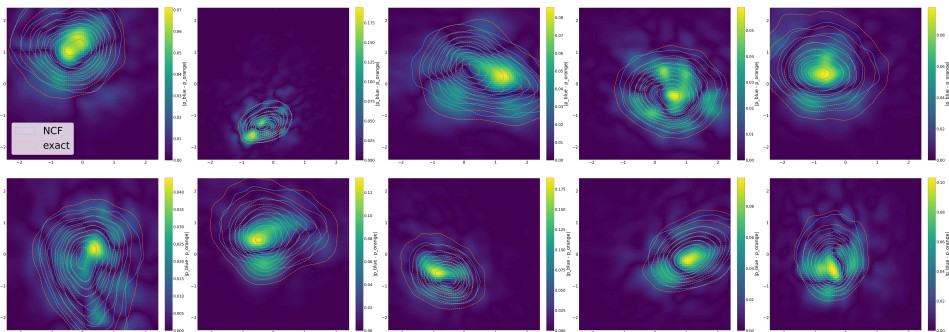

Figure 15: KDE contours of the MNIST latent samples generated using the computed OT map (blue) and the target samples (orange), conditioned on each MNIST class (0–9, arranged left to right and top to bottom). The samples are projected onto the first two PCA dimensions. The heat maps illustrate the discrepancies between the two distributions.

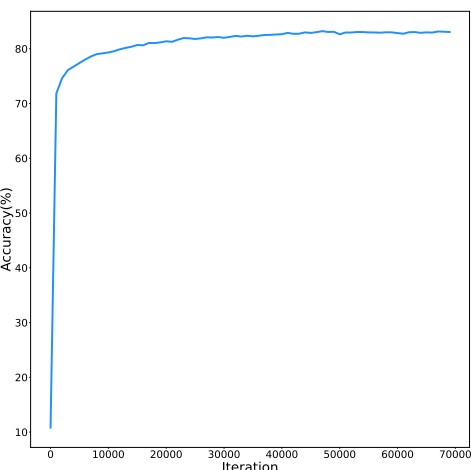

Figure 16: Accuracy(%) of trained forward class-conditional transport map versus training iterations. Results are displayed for the first 70000 iterations, beyond which the accuracy exhibits no significant improvement.

