# OpenReview forum: "Neural Hamilton--Jacobi Characteristic Flows for Optimal Transport"
_ICLR.cc/2026/Conference — ICLR 2026 Poster_

### Official Review · Reviewer_r22u · 2025-10-28

**Soundness:** 2
**Presentation:** 3
**Contribution:** 2
**Rating:** 4
**Confidence:** 4

**Summary:**

The paper proposes a Neural Characteristic Flow (NCF) method for static optimal transport (OT) with translation-invariant costs. Despite solving static OT, the method’s theory begins with the equivalent dynamical formulation, whose optimality condition is governed by the Hamilton–Jacobi (HJ) equation.
The core idea is to enforce an implicit solution formula (obtained via the method of characteristics) for the HJ PDE, whose viscosity solution uniquely determines the OT map.
To tie the PDE solution to the data (i.e., to satisfy boundary conditions), the authors add an MMD loss between the transported source and the target, combined with a data-driven HJ implicit-solution loss.
This design has several advantages: (1) it is a pure minimization approach that avoids the min–max optimization typical of dual methods and trains a single network; (2) although grounded in a dynamical viewpoint, it does not require ODE/SDE simulation; and (3) it jointly learns forward and backward maps from the same potential.
The authors also show that the framework extends to class-conditional OT by applying class-wise MMD within the same HJ formulation.
Empirically, with the standard quadratic cost, the method is shown to work in low-dimensional settings.

**Strengths:**

1. Implementation of a novel approach via an implicit HJ formulation (from paper Park & Osher, 2025).
2. Efficient training (no minimax optimization; no ODE/SDE simulation).
3. Theoretical analyses and guarantees (primarily for the quadratic-cost case).

**Weaknesses:**

1. **Lack of scalability evidence or discussion.**
Authors evaluate the proposed NCF only on low-dimensional datasets; even for MNIST they work in a low-dimensional VAE latent space. The method raises subtle potential scalability issues. First, it requires backpropagation through Jacobian–vector product, which could lead to instability and high memory consumption in higher dimensions. Second, the assumptions required by the theory—global bijectivity, a positive-definite Jacobian everywhere, and a strictly positive sampling distribution—may not hold in practical settings.

2. **Missed state-of-the-art baselines.**
The experimental evaluation would be strengthened by including comparisons with recent and relevant static Optimal Transport methods, specifically DIOTM [1] and ENOT [2]. These approaches have demonstrated superior performance and rapid convergence, and a direct comparison is crucial for a comprehensive assessment of the proposed method's contributions.

[1] Choi et al. Improving Neural Optimal Transport via Displacement Interpolation, ICLR 2025

[2] Buzun et al. ENOT: Expectile Regularization for Fast and Accurate Training of Neural Optimal Transport, NeurIPS 2024

**Questions:**

1. Why use a VAE latent space for MNIST experiments? Did you try solving the task in the original pixel space? If so, how do performance, stability, and runtime compare? If not, what specifically fails in pixel space?
2. Why does your method perform best in dimensions d≤16 but lose to MM-v1 at d=32 and d=64? Could you, please, discuss scalabilty of your method taking into account concerns from the weaknesses?
3. Could you provide an experimental comparison between NCF and DIOTM [1] and ENOT [2] ?
4. What are some real-world applications for the class-conditional OT method?
5. What is the advantage of using the Implicit Solution Formula (9) over the original HJB equation (5) (like in DIOTM [1] or HOTA [3])? Were any ablation studies conducted on this point?

[3] Buzun et al. HOTA: Hamiltonian framework for Optimal Transport Advection. https://arxiv.org/abs/2507.17513

---

> ### Author Response · Authors · 2025-11-21
> **Authors' Response to Reviewer r22u**
>
> We sincerely thank the reviewer for their thoughtful and constructive feedback. We address each point below.
>
> **Weakness 1**
> * Scalability:
>
> Our method, similar to other approaches that learn a potential function, represents the OT map via the gradient of a learned potential (HJ solution), requiring Jacobian–vector products. While this may raise concerns in very high dimensions, modern autograd frameworks handle it efficiently, with memory scaling linearly in batch size. No instability was observed. Similar computations are standard in scientific machine learning (Gao et al., 2023; Song et al., 2024). Nevertheless, extremely high-dimensional scalability is limited, as noted in the conclusion. Our focus was on demonstrating correctness and stability of the proposed HJ-based formulation rather than maximal scalability.
>
>
> Methods like GNOT and NOT scale more easily but do not enforce the OT form $T(x)=x+\nabla\phi(x)$ or the associated convexity and optimality conditions. As our experiments indicate, such methods may fail to learn a truly cost-minimizing OT map.
> Our method explicitly imposes the structure and HJ-based optimality, producing maps that more faithfully reflect true OT solutions, at higher computational cost, but allow learning forward and backward maps simultaneously. These trade-offs highlight complementary design priorities across methods: computational efficiency versus strict adherence to OT optimality.
>
> Importantly, many practical OT applications do not require extremely high-dimensional settings. For moderate-dimensional problems, our approach remains efficient and yields more accurate and structured transport maps than baseline methods. Our contribution lies in demonstrating how HJ-based optimality conditions can be effectively integrated into a deep learning framework, providing both practical benefits and new insights. While scalability in very high dimensions remains an inherent challenge, we believe the combination of HJ-based optimality and bidirectional mapping represents a promising direction for future research.
>
> * Assumptions:
>
> Regarding the theory assumptions in Theorem 5.1, first, these conditions are standard in the analysis of OT maps and related HJ equations. They are not required by the algorithm itself, but rather serve to establish that, in idealized continuous setting, the learned solution indeed converges to the desired OT map.
>
> Second, although the properties such as bijectivity are idealized, in practice, neural networks trained with our implicit HJ formulation have empirically demonstrated these behaviors, even without enforcing such constraints explicitly (Please refer to the synthetic examples in Figure 1 \& 5 for references, the majority of the transport trajectories generally do not intersect with each other.). Indeed, as shown in our experiments (particularly in Sections 6.2.1 and 6.2.2), the algorithm remains stable and produces semantically meaningful OT maps even when these assumptions are only partially satisfied.
>
> Third, the requirement that the sampling distribution be strictly positive is only a mild technical condition used to ensure that the implicit solution formula holds over the entire domain. In practice, typically in high dimension case, we do not need to enforce such a strong condition. Instead, an effective alternative is to choose $\varrho$ as a uniform/Gaussian distribution whose support adapts to the region covered by the available data samples (see details in Supplementary Material, Section B). This approach concentrates computational effort on the domain that is most relevant to the transport dynamics and scalable well as the dimension increases. The strategy has proved sufficient and stable in our experiments.
>
> In practice, many of the theoretical assumptions can be relaxed while still yielding stable and accurate performance, even as the problem scale increases.
>
> **Weakness 2**:
> We thank the reviewer for suggesting additional baselines. The official code for both DIOTM and ENOT is not publicly available (ENOT’s GitHub currently returns a 404), so we could not perform direct experiments. Nonetheless, we highlight key methodological differences. DIOTM relies on displacement interpolation between intermediate distributions, which does not strongly guarantee exact OT optimality. In contrast, our approach enforces optimality directly via the HJ equation and its characteristic-based closed formulation. ENOT uses expectile-based regularization within a max-min and c-transform framework, which can introduce numerical instability and longer runtimes. Moreover, unlike these methods that require training two networks, our approach learns bidirectional OT maps with a single network, providing a more stable, exact, and efficient solution.
>
> **Reference**
>
> Gao et al., Gradient descent finds the global optima of two-layer physics-informed neural networks, ICML, 2023.
>
> Song et al., How does PDE order affect the convergence of PINNs?, NeurIPS, 2024.

---

> ### Author Response · Authors · 2025-11-21
> **Authors' Response to Reviewer r22u**
>
> **Q1**:  We thank the reviewer for raising this important question.
> Methods learning a potential function, like ours, benefit from high accuracy and structured OT maps, but direct pixel-space computation is costly due to gradient calculations. While incorporating pixel-space generative architectures (e.g., U-Net–style decoders) could potentially help, integrating them into our HJ-based framework is nontrivial. Nonetheless, our approach provides important advantages—bidirectional OT representations and improved transport map accuracy—that are often difficult to achieve with fully scalable pixel-space methods.
>
> Working in a VAE latent space provides both theoretical and practical advantages. Mapping images to a lower-dimensional latent manifold reduces the OT problem’s dimensionality, improving computational efficiency and stability. MNIST and FMNIST lie on low-dimensional submanifolds (Pope, 2021), where OT maps may not exist or be well-defined in pixel space. So, computing pixel space OT can be ill-posed or numerically unstable. In contrast, a VAE latent space provides a smooth manifold for reliable OT learning.
>
> The goal of the latent-space experiments is semantic correctness rather than pixel-level synthesis.
> While FID may be lower than pixel-space models, our approach excels in applications where correct semantic transport matters more than photorealistic decoding, such as domain adaptation, representation alignment, and scientific machine learning. This is reflected in the strong class-conditional accuracy achieved by NCF.
>
> For these reasons, our experiments focus on the latent-space formulation. Extending NCF to operate directly in pixel space is an interesting and important future direction, which we expect to become feasible once suitable generative architectures can be integrated into our framework.
>
> **Q2**: To ensure a fair comparison, we used the same network architecture as prior OT baselines, and kept all training configurations fixed across dimensions.
> Unlike MM-v1, which parameterizes two neural networks for forward and backward maps, our method learns a single network. Our method learns both forward and backward OT maps, using roughly half the parameters of MM-v1.
>
> In higher dimensions ($d=32,64$), reduced capacity can limit performance; increasing network size recovers it while still using fewer parameters than MM-v1. Results in table show that HJ-based OT enforcement remains effective while improving speed, memory, and parameter efficiency.
>
> | Method                  | $d = 32$ | $d = 64$ |
> |-------------------------|----------|----------|
> | **MMv1**                |          |          |
> | UVP | 0.374    | 0.415    |
> | Training Time (s)| 0.542    | 0.630    |
> | Max Memory Allocation (MB)| 73.97    | 132.65   |
> | Bidirectional OT Map Storage (MB) | 0.125    | 0.508    |
> | **Ours (larger version)** |        |          |
> | UVP  | 0.307    | 0.407    |
> | Training Time (s) | 0.024    | 0.025    |
> | Max Memory Allocation (MB) | 65.88    | 114.88    |
> | Bidirectional OT Map Storage (MB) | 0.087    | 0.371    |
>
>
> **Q3**: Refer to our response to Weakness 2.
>
> **Q4**:  Class-conditional OT is useful when data has known class or cluster structure, enabling label-preserving transport. Nguyen et al. (2024) formulate class-aware OT with higher-order moment matching for unsupervised domain adaptation. Chuang et al. (2023) use conditional projection–based OT for representation alignment, improving cross-domain retrieval and image alignment. Manupriya et al. (2024) generate cell population responses conditioned on drug dosage in biomedical applications.
>
> For fair comparison with existing methods, we focused on standard benchmarks like MNIST and FMNIST. We acknowledge that applying to more practical real-world scenarios is important, and have highlighted this in the future work section of the conclusion in the revised manuscript.
>
> **Q5**:
> OT requires the viscosity solution of the HJB equation (Eq. 5), but it can admit multiple solutions. L2 residual loss used in DIOTM does not guarantee convergence to the viscosity solution. In contrast, the implicit solution formula explicitly characterizes the viscosity solution and aligns with HJ characteristics, making it more suitable for OT.
> This issue is directly illustrated by our HJ-PINN baseline, whose inferior performance shows the superiority of the implicit solution formula.
> Finally, HOTA deals with stochastic dynamics, so their HJB regularizer involves second-order derivatives and addresses a different OT problem. Hence, a direct comparison with our method may not be entirely appropriate.
>
> **Reference**
>
> Nguyen et al., A Class-Aware Optimal Transport Approach with Higher-Order Moment Matching for Unsupervised Domain Adaptation, arXiv:2401.15952, 2024.
>
> Chuang et al., Information Maximizing Optimal Transport, ICML 2023.
>
> Manupriya et al., Consistent Optimal Transport with Empirical Conditional Measures, ICML 2024.

---

### Official Review · Reviewer_nTXZ · 2025-10-30

**Soundness:** 3
**Presentation:** 3
**Contribution:** 2
**Rating:** 4
**Confidence:** 3

**Summary:**

This paper proposes a new algorithm for computing optimal transport maps between the distributions based on the usage of Hamilton-Jacobi (HJ) equation. For this purpose, the authors, first, represent the OT maps using the characteristics of the HJ equations; second, consider an implicit formula for the viscosity solution of HJ equation; third, parametrize the viscosity solution using neural networks and learn it by optimizing the  loss derived from the implicit solution formula and additional MMD regularizer. As a result, this allows for deriving the approximations of forward and backward OT maps.

**Strengths:**

In general, the paper is quite well-written and suggests some theoretical results supporting the constructed algorithm. The proposed algorithm allows for computation of OT maps without the need of min-max optimization or numerical integration of ODEs.

**Weaknesses:**

Some of the mathematical concepts introduced in the paper lack necessary details. First, the definition of Hamilton-Jacobi (HJ) equation given in section 2.2 lacks reference to the literature where it was introduced. The continuity equation mentioned in line 102 is not written directly in the text. Besides, the characterization of the viscosity solution by the system of characteristic ordinary differential equations (CODE) should be supported by the relevant reference.

Meanwhile, the intuition behind some of the ideas which appear in the derivation of the main objective is not sufficiently clear. In lines 204-209, the authors write that since the initial condition in the HJ equation is not known analytically, they introduce an additional loss term to ensure that this condition is appropriate. However, this loss term does not deal with the initial condition directly but rather enforce the solutions of the OT problem derived from the HJ (viscosity) solution to satisfy the marginal condition, i.e., to map the source distribution to the target. Thus, it seems that the addition of this loss term to the main objective serves another goal than the initially stated one, see questions section for my additional concerns on this topic. Moreover, the intuition behind the weighting function $\rho$ introduced  in the equation (13) is not clarified.

Still, while the authors provide some theoretical results justifying their approach, my major concerns correspond to the **error in the experimental comparisons** provided by the authors in section 6.1. The authors perform testing of their approach on the benchmark providing the ground-truth OT maps between the specific constructed pairs of distributions (Korotin, 2021a) and claim that their approach leads to the best performance. However, one of the competitive approaches (NOT) seems to be not tested correctly. Indeed, NOT approach actually corresponds to [MM:R] which was tested in the benchmark paper (Korotin, 2021a) and outperformed all other methods considered there. However, according to Table 2 in the paper under review, the NOT ([MM:R]) approach leads to awful results in comparison to another [MM-v1] method from the benchmark. This, actually, contradicts the results from the benchmark where the NOT method leads to the best results as I already said. Thus, I kindly suggest the authors check their results and implementation of the NOT approach which probably contains errors. These errors might also have affected the experimental results on the considered color transfer task. Meanwhile, the experimental results in section 6.2.2 show that the proposed approach did not beat the competitor GNOT according to FID metric. Thus, overall practical validity of the approach is unclear until the stated errors are resolved.

**In summary**, I am mostly confused by the revealed errors in the experimental comparisons provided in the paper. These errors raise doubts regarding the results reported for the experiments with high-dimensional Gaussians and the color transfer task. Thus, I could not assign a positive score to the paper until the issues are resolved.

**Questions:**

- Does your method have connections with the unbalanced optimal transport? This question appears since your loss (14) uses a regularization which motivates the learned OT maps to push input distribution to the target one. Such a strategy of softening the marginal constraint of the OT problem is usually exploited in unbalanced OT field. And the related question is – is it necessary to use MMD loss in this  regularization? Or maybe it can be replaced by some f-divergences?
- Is it possible to generalize your theoretical results in section 5.1 for general types of costs (other than quadratic)?
- Does the result of your algorithm depend on the choice of the parameter $\lambda$?
- Please explain the intuition behind the weighting function $\rho$ introduced in equation (13).

---

> ### Author Response · Authors · 2025-11-21
> **Authors' Response to Reviewer nTXZ**
>
> We sincerely thank the reviewer for their thoughtful and constructive feedback. We address each point below.
>
> **Weakness 1** Mathematical details missing:
>
> We thank the reviewer for the insightful comments. All the suggested clarifications and references have been added in the revised manuscript, and the continuity equation is now stated explicitly in the text.
>
> **Weakness 2** Intuition behind objective and weighting:
> From a PDE perspective, learning the OT map corresponds to an inverse problem---recovering the unknown initial function \( g \) from data. Minimizing the implicit HJ loss drives the network \( u \) to a viscosity solution of the HJ equation; however, without knowledge of \( g \), this solution does not necessarily correspond to the desired OT map.
>
> To address this, we introduce the MMD loss, which---as the reviewer correctly points out---enforces the marginal condition, ensuring that the map \( T[u] \) induced by \( u \) transports the source distribution to the target. By the uniqueness of viscosity solutions, satisfying both the HJ equation and the marginal condition implies that the network effectively learns the correct initial condition \( g \), and thus the desired OT map.
>
> Theorem 5.1 formalizes this result: minimizing \( L_{\text{HJ}} + \lambda L_{\text{MMD}} \) is sufficient to recover the OT map, providing a principled and theoretically grounded learning objective.
>
> The distributional weighting function $\varrho$ in Eq. (13) is introduced to enforce the implicit solution formula in Eq. (9) across the entire spatial domain. We choose $\varrho$ to be a probability distribution so that $\mathcal{L}_{\mathrm{HJ}}$ can be efficiently approximated using Monte Carlo sampling. A straightforward choice, when $\Omega$ is bounded, is to take $\varrho$ as the uniform distribution on $\Omega$. A more ad hoc but often effective alternative is to define $\varrho$ as a uniform distribution whose support adapts to the region covered by the available samples, thereby concentrating computational effort on the portions of the domain most relevant to the transport dynamics.
> We have clarified this in the revised manuscript to avoid any potential confusion.
>
> **Weakness 3** Experimental concerns:
>
> We thank the reviewer for pointing out the potential discrepancy regarding the NOT ([MM:R]) results. We would like to clarify a few important points:
>
> First, the NOT approach we evaluated corresponds to the official NOT implementation. In contrast, the [MM:R] mentioned in Korotin, 2021a is not included in the official GitHub repository, and the exact implementation details and hyperparameters are not publicly available. Therefore, the NOT implementation we use is not strictly identical to Korotin’s [MM:R].
>
> Second, as explicitly noted in Section 4.3 of Korotin, 2021a, ''the maximin-based solvers such as [MM:R] are also hard to optimize: they either diverge from the start or diverge after converging to nearly-optimal saddle point. This behavior is typical for maximin optimization and possibly can be avoided by a more careful choice of hyperparameters.
> '' This instability is typical of maximin optimization and complicates direct comparison with the [MM:R] results reported in Korotin, 2021a.
>
> In light of these points, we believe that our evaluation using the official NOT implementation provides a fair and reproducible baseline. Any apparent discrepancy with Korotin’s [MM:R] results can be attributed to the instability of the maximin solver and the lack of publicly available code to exactly reproduce [MM:R].
>
> Regarding the GNOT comparison, we agree that our latent-space approach does not outperform GNOT in pixel-level FID. However, that FID is not the primary metric for evaluating the practical validity of our method, since our framework is designed for semantic OT in latent space rather than pixel-level image synthesis. As discussed in Section 6.2.2 and the supplementary material, the proposed method achieves superior class-conditional transport accuracy, which serves as a central objective in many downstream tasks such as domain adaptation, representation alignment, and scientific machine learning.
>
> In addition, the FID scores in our experiments are influenced primarily by the VAE decoder rather than by the OT map itself. As reported in Section 6.2.2, when FID is recomputed between NCF-generated and VAE-decoded images, the score improves substantially to $2.73$, indicating that the transport map is accurate even though the decoder limits pixel-level fidelity. Thus, while our method is not intended to compete with pixel-space generative models in producing high-resolution images, it remains practically effective for applications where semantic correctness and computational efficiency are central.
>
> Overall, we believe that the experimental results demonstrate that the stated issues do not undermine the validity of our approach, which continues to offer a useful and efficient solution for latent-space OT problems.

---

> ### Author Response · Authors · 2025-11-21
> **Authors' Response to Reviewer nTXZ**
>
> **Q1**: While the MMD loss is flexible for the unbalanced OT, the corresponding Hamilton–Jacobi and characteristic formulations for unbalanced OT are fundamentally different from the classical setting used in our work. So we believe it is better to leave this extension for future research. In addition, Most $f$-divergences require access to probability densities or density ratios, which are unavailable in typical OT settings. MMD, in contrast, is directly computable from samples, stable in high dimensions, and integrates smoothly with the HJ implicit formulation.
>
> **Q2**: We believe that the theoretical result in Section 5.1 can be extended to a broader class of OT costs of the form $h(x-y)$, where $h(\cdot)$ is strictly convex. The key step is to differentiate the relation
> $$
> u_0\left(x - t_f \nabla h^{-1}(\nabla u_1(x))\right)
> = u_1(x) - t_f h(\nabla u_1(x)), \qquad x \in \mathbb{R}^d
> $$
> with respect to $x$, and use the resulting expression to verify the $c$-concavity properties of the associated Kantorovich potential. Once this structure is established, one may apply Theorem 1.47 in Filippo (2015) to obtain an analogue of our consistency result.
>
> Since this extension requires additional technical development beyond the scope of the present manuscript—where we focus on the classical quadratic cost—we plan to pursue this direction in future work.
>
> Regarding Theorem 5.4 (stability), we remark that the proof relies essentially on the quadratic structure. Extending this stability analysis to general convex costs would require a fundamentally different approach, and we leave this as an open problem for future investigation.
>
>
> **Q3**: The effect of the parameter $\lambda$ has been examined in Appendix C.1 of the original manuscript, with the corresponding ablation results shown in Figure 6. The results show that the model performance is not highly sensitive to $\lambda$. When the HJ loss is overly emphasized, the model struggles to transport between distributions properly (as the initial condition is not well learned). Conversely, when the MMD loss dominates, the transport becomes feasible but the optimality of the transport map is not preserved. This demonstrates that both losses play complementary and essential roles in achieving accurate and OT.
>
> **Q4**: We thank the reviewer for raising the question. The distributional weighting function $\varrho$ in Eq. (13) is introduced to enforce the implicit solution formula (9) across the entire spatial domain. We choose $\varrho$ to be a probability distribution so that $\mathcal{L}_{\mathrm{HJ}}$ can be efficiently approximated using Monte Carlo sampling. A straightforward choice, when $\Omega$ is bounded, is to take $\varrho$ as the uniform distribution on $\Omega$. A more ad hoc but often effective alternative is to define $\varrho$ as a uniform distribution whose support adapts to the region covered by the available samples, thereby concentrating computational effort on the portions of the domain most relevant to the transport dynamics.
>
> **Reference**
>
> Filippo Santambrogio. Optimal transport for applied mathematicians, 2015.

---

> > ### Comment · Reviewer_nTXZ · 2025-11-26
> > **Answer to the Authors**
> >
> > I thank the authors for their answers. However, my **main concern** regarding the probably incorrect implementation of NOT (Korotin, 2023) approach remains valid.
> >
> > > "[MM:R] mentioned in (Korotin, 2021a) is not included in the official GitHub repository, and the exact implementation details and hyperparameters are not publicly available. Therefore, the NOT implementation we use is not strictly identical to [MM:R]."
> >
> > I kindly disagree with you. The official GitHub repository the benchmark includes direct implementation of [MM:R] approach. You should consider the notebooks which implement MM solver and set parameter 'REVERSED = True' there.
> >
> > Using this official implementation, it should be easy to reproduce the baseline.
> > I feel that there is enough time till the end of the rebuttal and kindly suggest the authors to run the experiments with official implementation of [MM:R].
> >
> > This experiment should help to reveal the actual roots of contradiction between the poor results of NOT solver reported by the authors and the results of the [MM:R] solver (the same as NOT) from the benchmark paper (Korotin, 2021a).
> >
> > **References.**
> >
> > (Korotin, 2021a) Alexander Korotin, Lingxiao Li, Aude Genevay, Justin M Solomon, Alexander Filippov, and Evgeny Burnaev. Do neural optimal transport solvers work? a continuous wasserstein-2 benchmark. Advances in neural information processing systems, 34:14593–14605, 2021a.
> >
> > (Korotin, 2023) Alexander Korotin, Daniil Selikhanovych, and Evgeny Burnaev. Neural optimal transport. International conference on learning representations, 2023.

---

> > > ### Author Response · Authors · 2025-11-28
> > >
> > > We sincerely thank the reviewer for the clarification. Following your suggestion, we have now conducted experiments using the **official MM:R implementation** from the benchmark GitHub repository. The results are summarized in Table 2 of Section 6.1.2 in the revised manuscript. In line with the original paper, the reported UVP error values correspond to the minimum errors observed during training. However, we observed **training instability** and a **sensitivity on a special initialization** (see Figure 7 in Appendix C.2 of the revised manuscript). Even when running the full training schedule specified in the official code, the error increases dramatically over epochs, as shown in Figure 7.
> > >
> > > In practical OT scenarios, the true OT map is unknown, making it unclear when training should be stopped or which solution is preferable. Therefore, the ability of **our method to maintain stable training over many epochs**, as demonstrated in Figure 7, represents a crucial practical advantage.
> > >
> > > We also note that in the original MM:R implementation, the model relies on a **special identity-potential pretraining** to stabilize training. Without this initialization, training becomes highly unstable and can completely fail to approximate the OT map. In contrast, our method does not require any such special initialization and still achieves **stable and accurate OT map approximation**.
> > >
> > > Finally, we would like to clarify that the official NOT implementation used in our original experiments is not strictly identical to MM:R. Although MM:R and NOT share some conceptual similarities, tthey differ in how the OT map is parameterized and in their optimization strategies. Moreover, the official NOT implementation does not rely on any special initialization. Considering that MM:R becomes unstable without this initialization, the results of NOT observed in our experiments appears consistent and understandable. We therefore believe that the behavior we reported for NOT is reasonable within the context of these distinctions, rather than a discrepancy arising from incorrect experimentation.
> > >
> > > Additionally, we are currently running color transfer experiments using the official MM:R implementation. Once these experiments are completed, we will update the revised manuscript accordingly.

---

> > > > ### Author Response · Authors · 2025-11-28
> > > >
> > > > We have conducted additional experiments using MM:R for the color transfer task, and we have now included these results in the revised manuscript. We believe these additional experiments further demonstrate the effectiveness and robustness of our approach.
> > > >
> > > > Once again, we sincerely appreciate your time and feedback.

---

### Official Review · Reviewer_9wfo · 2025-11-01

**Soundness:** 2
**Presentation:** 3
**Contribution:** 2
**Rating:** 4
**Confidence:** 4

**Summary:**

Single-network and pure minimization training for conditional OT maps. The proposed method eliminates the need for adversarial training or saddle-point optimization that many "OT-GAN" or dual potential methods use. NCF uses a single neural network and a single loss function to compute bidirectional maps. This is a significant simplification: prior approaches often involve two networks (e.g. a generator and a discriminator or dual potentials) and tricky min–max optimization, which can be unstable and require extensive hyper-parameter tuning.

**Strengths:**

The paper is very well written and easy to follow. The theoretical motivation, from the dynamical formulation of OT and the Hamilton-Jacobi (HJ) equation to the method of characteristics, is presented clearly. The core idea of leveraging this to derive closed-form, bidirectional maps is elegant.

The proposed Neural Characteristic Flow (NCF) framework is compelling. Its main advantages using a single neural network and, crucially, avoiding adversarial min-max optimization  are significant practical contributions over common dual-formulation approaches.

The algorithm is supported by theoretical arguments. The authors provide a consistency analysis (Theorem 5.1) showing that a zero loss recovers the true OT map and a stability analysis (Theorem 5.4), but demonstrating that a small loss guarantees convergence in the Gaussian setting

**Weaknesses:**

The HJ formulation for OT is well-established. The "implicit solution formula" (Equation 9) , which is the foundation for the proposed loss , is explicitly cited from a very recent work (Park & Osher, 2025). The main contribution appears to be the application of this new formula to the OT problem, combined with a standard MMD loss, which was already considered in (Asadulaev, et. al. 2024).

Similar to other papers in this area, the primary class-conditional experiment is transporting Fashion MNIST to MNIST. This setup (e.g., mapping a "Trouser" to a "1") feels artificial and does not provide a compelling, practical use case for the class-conditional framework. The colour transfer setup is also very simple and is often easier than mapping between mixtures of Gaussians.

The experimental validation, while broad, has a significant weakness in the class-conditional setting. In Table 4 (Fashion MNIST → MNIST) , while NCF achieves the highest accuracy (83.42%), its FID score of 18.27 is substantially worse than GNOT (5.26), NOT (7.51), and MUNIT (7.91). The paper attempts to explain the poor FID by blaming the VAE decoder. The authors then provide a "re-calculated" FID of 2.73, computed between NCF outputs and VAE-decoded images, but no comparison with other methods are done.

**Questions:**

**Q1**: The Theorem 5.1 assumes regularity conditions. How does the use of an infinitely-smooth neural network reconcile with approximating a potentially non-smooth solution? Does this implicit regularization bias the solution away from the true, non-smooth solution?

**Q2**: Could you please clarify, is "Optimality" in Table 1 means guaranteed convergence to the optimum? Why Dual Formulation and Dynamical Models are mentioned as "No"?

**Q3**: The GNOT paper considered an image-to-image paired cost setup. Is your method applicable to high-dimensional paired data, forward and backward? Regarding the class-conditional results in Table 4: Given the poor FID score (18.27) compared to baselines like GNOT (5.26), can the method truly be considered competitive for high-fidelity generative tasks without using any decoders?


**Minor**:

*Q4*: For Figure 4, it would be interesting to see how the model maps the generated forward samples backward.

*Q5*: The 'integration-free' nature of the model is a key feature. By deriving a closed-form map from the HJ characteristics, the method avoids the need to solve ODEs. However, are  ODEs a real bottleneck for dynamical OT models? There is a well-known trilemma between coverage, sampling and 'accuracy' of generative models (https://arxiv.org/pdf/2112.07804). It would be interested to see a dynamic formulation of the proposed method. Can be obtain bidirectional maps in dynamical OT formulation? Maybe the authors have conducted some experiments using an ODE-like formulation, as done by (https://arxiv.org/pdf/2507.17513) using the HJB framework? The comparison would be very interesting.

---

> ### Author Response · Authors · 2025-11-21
> **Authors' Response to Reviewer 9wfo**
>
> We sincerely thank the reviewer for their thoughtful and constructive feedback. We address each point below.
>
> **Weakness 1** Contribution relative to prior HJ/OT formulations:
>
> We thank the reviewer for their comments.
>
> We agree that the HJ formulation for OT is well-established, and that the implicit solution formula in Equation 9 has been recently discussed in Park \& Osher (2025). We would like to carefully note that many deep learning approaches for OT are built upon different well-established mathematical formulations, each of which has distinct implications when integrated into neural network frameworks. How a particular mathematical formulation is integrated into a deep learning framework, and how it affects trainability, stability, and computational efficiency, are central concerns when designing OT-based networks.
>
> Our contribution lies not in deriving a new formula, but in the careful integration of the HJ implicit formula with deep learning OT methods and an MMD-based loss. While the HJ formulation for OT is well-known, how to leverage its structure effectively within deep learning training remained an open question. We focused on the fact that the HJ-based OT formulation represents solutions along characteristics, and thus the implicit solution formula, which naturally encodes these characteristics, is particularly suitable for learning OT maps. As shown in the paper, this characteristic-based HJ approach is highly flexible, enabling applications to standard OT, class-conditional OT, and various cost functionals. This combination provides practical advantages in terms of training efficiency, stability, and accuracy, as supported by our empirical results.
>
> **Weakness 2** Artificiality of the class-conditional (Fashion→MNIST) experiment:
>
> We appreciate this comment and agree that more realistic datasets would strengthen the work. We were aware of this limitation during our study and explored alternative, more realistic datasets. However, nearly all existing baselines rely on MNIST ↔ Fashion-MNIST or 2D toy settings, and using a new dataset would require extensive re-tuning that risks unfair comparisons. For consistency and reproducibility, we adopted the widely used benchmarks.
>
> We fully agree that extending our method to more realistic domains, such as medical imaging or other specialized datasets, is an important and promising direction. Our main goal with the current experiments was to demonstrate that the proposed HJ characteristic-based approach can be naturally and effectively applied to class-conditional OT, highlighting its generality. We believe that even within the current benchmarks, our experiments provide clear insights into the fundamental behavior and advantages of the proposed approach. We have highlighted extending our approach to more realistic applications as a direction for future work in the revised manuscript.
>
> **Weakness 3** FID discrepancy in class-conditional experiments:
>
> We thank the reviewer for raising this important point. We fully agree that the discrepancy between classification accuracy and the FID score in Table 4 deserves careful clarification.
>
> Our method performs OT in the VAE latent space.
> The reported FID of $2.73$, calculated between NCF outputs (decoded from transported latent samples) and VAE-decoded target images, is intended \textit{solely} to demonstrate the accuracy of the learned transport map—i.e., how well the source distribution $\mu$ is pushed forward to the target $\nu$ in the latent space where the OT map is actually learned. This metric is not intended for direct comparison with pixel-space generative models.
>
> We acknowledge that our current framework does not perform direct density transport in pixel space. While incorporating a VAE with structures such as U-Net could potentially reduce the FID discrepancy in pixel space, exploring this integration is beyond the current scope of our study. Nevertheless, performing transport in the latent space has clear motivations and advantages:
>
> 1. Mapping images into a VAE latent space drastically lowers the problem dimension, making OT computation more tractable and efficient. Moreover, datasets like MNIST and Fashion-MNIST have very low intrinsic dimension relative to the ambient pixel space (Pope, 2021), so direct OT in pixel space is often ill-posed, may not exist, or is computationally challenging.
>
> 2. Although latent-space transport does not achieve the pixel-level FID of image-to-image translation models, it is well-suited for tasks where semantic correctness outweighs visual fidelity, such as domain adaptation, representation alignment, and scientific learning. In such settings, performing OT in a well-learned latent space is not only sufficient but often preferable.
>
> We believe that the latent-space approach enables the efficient learning of accurate and semantically meaningful transport maps, even though pixel-level image fidelity is somewhat lower.

---

> ### Author Response · Authors · 2025-11-21
> **Authors' Response to Reviewer 9wfo**
>
> **Q1**: There already exists substantial numerical evidence demonstrating the capability of smooth neural networks to approximate non-smooth solutions. Park & Osher (2025) show MLPs with softplus activations capture sharp kinks and non-differentiable structures. In class-conditioned OT, $\nabla u(\cdot, t)$ can be non-smooth where characteristics intersect. Our results show that the algorithm remains stable and accurately recovers transport maps under such non-smooth behavior, highlighting the effectiveness of our formula.
>
> **Q2**: We apologize for the ambiguity. In Table 1, “Optimality” indicates whether a method can recover the unbiased OT map assuming sufficient network capacity. While many primal–dual formulations can match the exact OT map, the dual formulation in Asadulaev (2022) does not guarantee this, as its cost function (Eq. 7) penalizes only the distribution discrepancy, not the transport cost. Thus, the learned map may match distributions but deviate from the true OT map. We clarified this in the revised table caption.
>
> **Q3**: Potential-based methods, including ours, parameterize OT maps via gradients of a learned potential. Consequently, handling extremely high-dimensional data directly can be computationally challenging, and this limitation is acknowledged in the conclusion.
>
> In contrast, methods such as GNOT and NOT directly parameterize the transport map, allowing them to scale more easily to high dimensions since no structural constraints are enforced during training. However, these approaches do not guarantee that the learned map satisfies the OT form $T(x)=x+\nabla\phi(x)$ or the associated convexity and optimality conditions. As our experiments indicate, such methods may fail to minimize the transport cost accurately.
>
> Potential-based methods—including ours—explicitly impose this structure, producing maps that more faithfully reflect true OT solutions, though at higher computational cost. By incorporating the HJ equation, our method further enforces the optimality conditions. An additional advantage of our framework is that a single network can simultaneously learn both the forward and backward OT maps, providing a bidirectional transport solution that is useful in practical applications. These trade-offs highlight complementary design priorities across methods: computational efficiency versus strict adherence to OT optimality.
>
> Importantly, many practical OT applications do not require extremely high-dimensional settings. For moderate-dimensional problems, our approach remains efficient and yields more accurate and structured transport maps than baseline methods. Our contribution lies in demonstrating how HJ-based optimality conditions can be effectively integrated into a deep learning framework, providing both practical benefits and new insights. While scalability in very high dimensions remains an inherent challenge, we believe the combination of HJ-based optimality and bidirectional mapping represents a promising direction for future research.
>
> **Q4**:  Thanks for the nice advice. We have added examples of transporting generated forward samples backward in Figure 4 in our revised manuscript.
>
> **Q5**:  We would like to clarify that the key feature of our method is that the OT map corresponds exactly to the characteristics of the deterministic HJ equation, and these characteristics are straight lines. This allows us to obtain a closed-form solution for the OT map without the need for any numerical ODE integration—hence the “integration-free” nature of our approach.
>
> The ODE-based dynamic formulation suggested by the reviewer, while mathematically equivalent to our closed-form formulation, does not offer practical advantages in our setting. Numerical ODE integration is unnecessary and can introduce errors.
> To verify this, we compared our integration-free map (equivalent to $K=0$ Euler steps) with numerical Euler integration using steps $K=20,30,40,50$. Results in the table confirms that integration adds time and slightly increases error without practical benefit.
>
> | **K** |  **UVP** | **Eval Time (s)** |
> |----------------:|----------------:|---------------------:|
> | **0 (ours)**  | **0.1461**      | **0.0060**           |
> | 20             | 0.1517          | 0.0936               |
> | 30            |0.1522          | 0.1454               |
> | 40            | 0.1525          | 0.2040               |
> | 50            | 0.1527          | 0.2543               |
>
> Regarding the generative modeling trilemma, our integration-free formulation improves accuracy and sampling efficiency, while coverage is mainly determined by the MMD loss. The HJ framework ensures OT map accuracy, while numerical integration has little influence on coverage.
>
> Furthermore, the reference provided by the reviewer considers stochastic HJB formulations, which correspond to a different class of OT problems. Therefore, the stochastic approach would address a different problem and is not directly comparable to our method.

---

### Author Response · Authors · 2025-11-21
**Authors' response to all reviewers**

We sincerely thank the reviewers for their thoughtful and constructive feedback. We greatly appreciate the time and effort they devoted to evaluating our work. In response, we have carefully revised the manuscript to address all comments. Changes in the revised manuscript are highlighted in blue for clarity.

Below, we provide detailed responses to each reviewer’s questions and comments. We believe these revisions have further improved the quality and presentation of our paper.

---

### Meta-Review · Area_Chair_MPE2 · 2026-01-08

**Summary:**

The reviewers are concerns that the main contributions are overlapping with known prior results, some baselines are not using the correct implementations, mathematical details are missing, and intuition behind algorithms choices are lacking. However, the authors have done a great job at addressing the concerns and I believe that paper is of high quality.

**Reviewer Concerns:**

Correct implementation of the baseline and intuition behind algorithms choices have not been fully addressed.

**Reviewer Scores:**

I believe the reviewers would have changed their scores to
- 9wfo : 6
- nTXZ : 6
- r22u : 6

---

### Decision · Program_Chairs · 2026-01-26

Accept (Poster)